# Estrogen receptor coregulator binding modulators (ERXs) effectively target estrogen receptor positive human breast cancers

Ganesh V Raj[1]*, Gangadhara Reddy Sareddy[2,3†], Shihong Ma[1†], Tae-Kyung Lee[4†], Suryavathi Viswanadhapalli[2†], Rui Li[1], Xihui Liu[1], Shino Murakami[5,6], Chien-Cheng Chen[1], Wan-Ru Lee[1], Monica Mann[2], Samaya Rajeshwari Krishnan[2], Bikash Manandhar[4], Vijay K Gonugunta[2], Douglas Strand[1], Rajeshwar Rao Tekmal[2,3], Jung-Mo Ahn[4], Ratna K Vadlamudi[2,3*]

[1]Departments of Urology and Pharmacology, University of Texas Southwestern Medical Center at Dallas, Dallas, United States; [2]Department of Obstetrics and Gynecology, University of Texas Health Science Center, San Antonio, United States; [3]CDP program, University of Texas Health Cancer Center, San Antonio, United States; [4]Department of Chemistry and Biochemistry, University of Texas at Dallas, Richardson, United States; [5]Department of Obstetrics and Gynecology, University of Texas Southwestern Medical Center, Dallas, United States; [6]Laboratory of Signaling and Gene Regulation, Cecil H and Ida Green Center for Reproductive Biology Sciences and Division of Basic Reproductive Biology Research, University of Texas Southwestern Medical Center, Dallas, United States

*For correspondence: ganesh.raj@utsouthwestern.edu (GVR); vadlamudi@uthscsa.edu (RKV)

†These authors contributed equally to this work

Competing interests: The authors declare that no competing interests exist.

**Abstract** The majority of human breast cancer is estrogen receptor alpha (ER) positive. While anti-estrogens/aromatase inhibitors are initially effective, resistance to these drugs commonly develops. Therapy-resistant tumors often retain ER signaling, via interaction with critical oncogenic coregulator proteins. To address these mechanisms of resistance, we have developed a novel ER coregulator binding modulator, ERX-11. ERX-11 interacts directly with ER and blocks the interaction between a subset of coregulators with both native and mutant forms of ER. ERX-11 effectively blocks ER-mediated oncogenic signaling and has potent anti-proliferative activity against therapy-sensitive and therapy-resistant human breast cancer cells. ERX-11 is orally bioavailable, with no overt signs of toxicity and potent activity in both murine xenograft and patient-derived breast tumor explant models. This first-in-class agent, with its novel mechanism of action of disrupting critical protein-protein interactions, overcomes the limitations of current therapies and may be clinically translatable for patients with therapy-sensitive and therapy-resistant breast cancers.

## Introduction

Endocrine therapies for estrogen receptor alpha (ER)-positive breast cancer involve modulation of ER signaling using either antiestrogens (AE) or aromatase inhibitors (AI). However, most patients develop resistance to these drugs, and disease progression is common, with progression to metastases (*Musgrove and Sutherland, 2009*; *Ma et al., 2015*). ER signaling is complex and involves coregulators (*McDonnell and Norris, 2002*; *O'Malley and Kumar, 2009*). Therapy-resistant tumors

**eLife digest** Around 70% of breast cancers in women need one or both of the female hormones (estrogen and progesterone) to grow. To treat these 'hormone-dependent' cancers, patients receive drugs that either block the production of estrogen or directly target a receptor protein that senses estrogen in the cancer cells. Unfortunately, many breast cancers develop resistance to these drugs. This resistance is often caused by genetic mutations that alter the estrogen receptor; for example, the receptor may develop the ability to interact with other proteins in the cell known as coregulators to promote tumor growth.

Developing new drugs that prevent estrogen receptors from interacting with coregulators may provide more options for treating hormone-dependent breast cancers. Here, Raj et al. developed a new small molecule named ERX-11 that is able to inhibit the growth of human breast cancer cells that are sensitive to existing drugs as well as cells that have become drug-resistant.

For the experiments, hormone-dependent breast cancer cells from humans were transplanted into mice. This procedure usually causes the mice to develop tumors, but giving the mice ERX-11 by mouth stopped estrogen receptors from interacting with coregulators and blocked the growth of tumors. Furthermore, ERX-11 does not appear to have any toxic effects on the mice, indicating that it may also be safe for humans.

The findings of Raj et al. suggest that ERX-11 is a promising new drug candidate for treating some breast cancers. The next steps are to examine the effects of ERX-11 on mice and other animals in more detail before deciding whether this molecule is suitable for clinical trials. In the longer term, molecules similar to ERX-11 could also be developed into drugs to treat other types of cancer that are also caused by abnormal interactions of coregulator proteins.

often retain ER-expression and ER-signaling. While multiple mechanisms maintain ER signaling in therapy-resistant tumors, ER signaling is mediated by the interactions between activated ER and critical coregulator proteins (*Dasgupta and O'Malley, 2014*; *Lonard et al., 2007*).

Alterations in the concentration or activity of selective coregulators enable ER-signaling from AE-ER complexes, effectively converting the antagonist to an agonist (*O'Hara et al., 2012*; *Kurebayashi, 2003*). Over a third (38%) of ER coregulators identified in breast cancer are over-expressed (*Lonard et al., 2007*; *Lonard and O'Malley, 2012*; *Cortez et al., 2014*), such as SRC3 (AIB1) (*List et al., 2001*; *Azorsa et al., 2001*), SRC2 (*Kurebayashi et al., 2000*), and PELP1 (*Habashy et al., 2010*). These deregulated coregulators contribute to mammary tumorigenesis (*Cortez et al., 2014*), therapy resistance and metastases (*Kumar et al., 2009*; *Shou et al., 2004*; *Burandt et al., 2013*; *Girard et al., 2014*).

Recent studies revealed that breast tumors acquire mutations in the ER ligand binding domains (L536N, Y537S, Y537N and D538G) that facilitate constitutive activity of these mutant ER (MT-ER) in the absence of ligand (*Toy et al., 2013*; *Robinson et al., 2013*; *Jeselsohn et al., 2014*; *Merenbakh-Lamin et al., 2013*). Tumors with MT-ER interact with oncogenic coregulators to drive ER-dependent transcriptional programs and proliferation and are poorly responsive to AEs and AIs (*Toy et al., 2013*; *Robinson et al., 2013*; *Jeselsohn et al., 2014*; *Merenbakh-Lamin et al., 2013*; *Toy et al., 2017*).

Thus, there is a strategic need for drugs that disrupt interactions between ER and critical coregulators to block ER signaling. In this study, we have synthesized a series of small organic molecules to emulate the nuclear receptor (NR) box motif, important for ER coregulator interactions. We have identified a small molecule named as ER coregulator binding modulator-11 (**ERX-11**), with potent anti-proliferative activity against ER-driven breast tumors. ERX-11 interacts with ER and blocks the interaction between ER and coregulators. In ER-expressing breast cancer cells, ERX-11 blocks the proliferation and induces apoptosis. ERX-11 has no activity against ER-negative breast cancer cells.

## Results

### Screening and identification of ER coregulator binding modulator 11 (ERX-11)

The peptidomimetic D2 blocks the interaction between the androgen receptor (AR) and NR-box containing coregulators, such as PELP1, with an IC50 of 40 nM (*Ravindranathan et al., 2013*). String analyses of the PELP1 interactome suggested an equally robust interaction between PELP1 and ER as that between PELP1 and AR (*Figure 1—figure supplement 1A*). However, D2 was unable to block the interaction between PELP1 and ER (data not shown) and required much higher concentrations (µM range) to block the proliferation of ER-driven MCF-7 breast cancer cells (*Figure 1—figure supplement 1B*).

Since sequences flanking the NR-box may influence the affinity and selectivity of coregulator interactions (*McInerney et al., 1998*), we hypothesized that a longer oligo-benzamide scaffold may more effectively target the interaction between ER and coregulators. This strategy generated a series of tris-benzamides (see *Appendix 1—chemical structures 1—22*) (*Figure 1A*) that added a functional group (R) to the D2 bis-benzamide, corresponding to the amino acid side chain groups found at the i-3/4 or i + 7 position surrounding the NR-box sequences (*Figure 1A*, *Figure 1—figure supplement 2G*). Importantly, each of these small molecules were named as ER coregulator binding modulators (ERXs, where X refers to their multiple potential and unknown targets) had differential activities in ER-positive breast cancer cells (*Figure 1A*, *Figure 1—figure supplement 1C*), confirming our hypothesis that the sequences flanking the core LXXLL motif could determine specificity and activity. The ERXs maintained the structural requirement for mimicking helices (confirmed by molecular modeling (MacroModel, Schrodinger, NY) (*Figure 1—figure supplement 2A*, shown for ERX-11).

Within this series of tris-benzamides, one compound (**ERX-11**) with a hydroxyethyl functional group mimicking a serine residue (*Figure 1A*) was consistently able to block 17-$\beta$-estradiol (E2)-induced proliferation in 8/8 ER-positive breast cancer cell lines (*Figure 1B*, *Figure 1—figure supplements 1C, D* and *2B*), with an IC50, ranging from 250 to 500 nM. In contrast, no effect of ERX-11 was noted in 11/12 ER-negative cells, with modest activity in the MDA-MB-157 cell line (*Figure 1C*). ERX-11 was as effective as tamoxifen or ICI in reducing the growth of ZR-75 and MCF-7 cells (*Figure 1—figure supplements 1E* and *2C,D,E*), and the combination of ERX-11 and tamoxifen was not additive (*Figure 1—figure supplement 2F*). While several compounds, including selective estrogen receptor modulators, have been shown to have similar activity against ER-positive breast cancers, the novel chemical structure, potential for a unique mechanism of action led to the designation of ERX-11 as a lead compound. We then established a protocol for large-scale batch synthesis (*Figure 1—figure supplement 3A–C*).

### ERX-11 interacts with ER

Since ERX-11 mimics the NR-box, we expected its direct binding to ER. Modeling studies using Autodock (The Scripps Research Institute, La Jolla, CA) indicated that ERX-11 could bind to the AF-2 domain of ER (*Figure 1D*). Using biotinylated ERX-11 (synthesis described in *Figure 1—figure supplement 3D*), we showed that ERX-11 interacts in vitro with purified ER protein (*Figure 1E*). Addition of 'cold' ERX-11 efficiently competed for the binding of purified ER protein to biotinylated ERX-11 (*Figure 1F*). Further, short 15mer peptides, corresponding to NR box sequences within the SRC1, SRC2, AIB1 and PELP1 proteins, efficiently disrupted the interaction between biotinylated ERX-11 and purified ER (*Figure 1G*). However, not all LXXLL peptides interfere ERX-11 interaction with ER, as only peptide surrounding the third PELP1 LXXLL motif, but not the first PELP1 LXXLL motif blocked ERX-11 interaction suggesting ERX-11 can only block some LXXLL interactions (*Figure 1G*, *last two panels*). Further, pre-incubation of the purified ER protein with selective estrogen receptor degraders (SERDs) GDC-0810 or AZD-9496, or fulvestrant (ICI) was unable to block the interaction between ER and ERX-11 (*Figure 1H*). In contrast, tamoxifen was able to block the interaction between purified ER and ERX-11 (*Figure 1H*), suggesting similarities in the ER-binding pockets of ERX-11 and tamoxifen.

We then demonstrated that biotinylated ERX-11 could pull down endogenous ER in ZR-75 nuclear extracts (*Figure 2A*). These data indicate that ERX-11 directly interact with ER, both as purified protein and within a cellular context. Unbiased evaluation, using immunoprecipitation mass

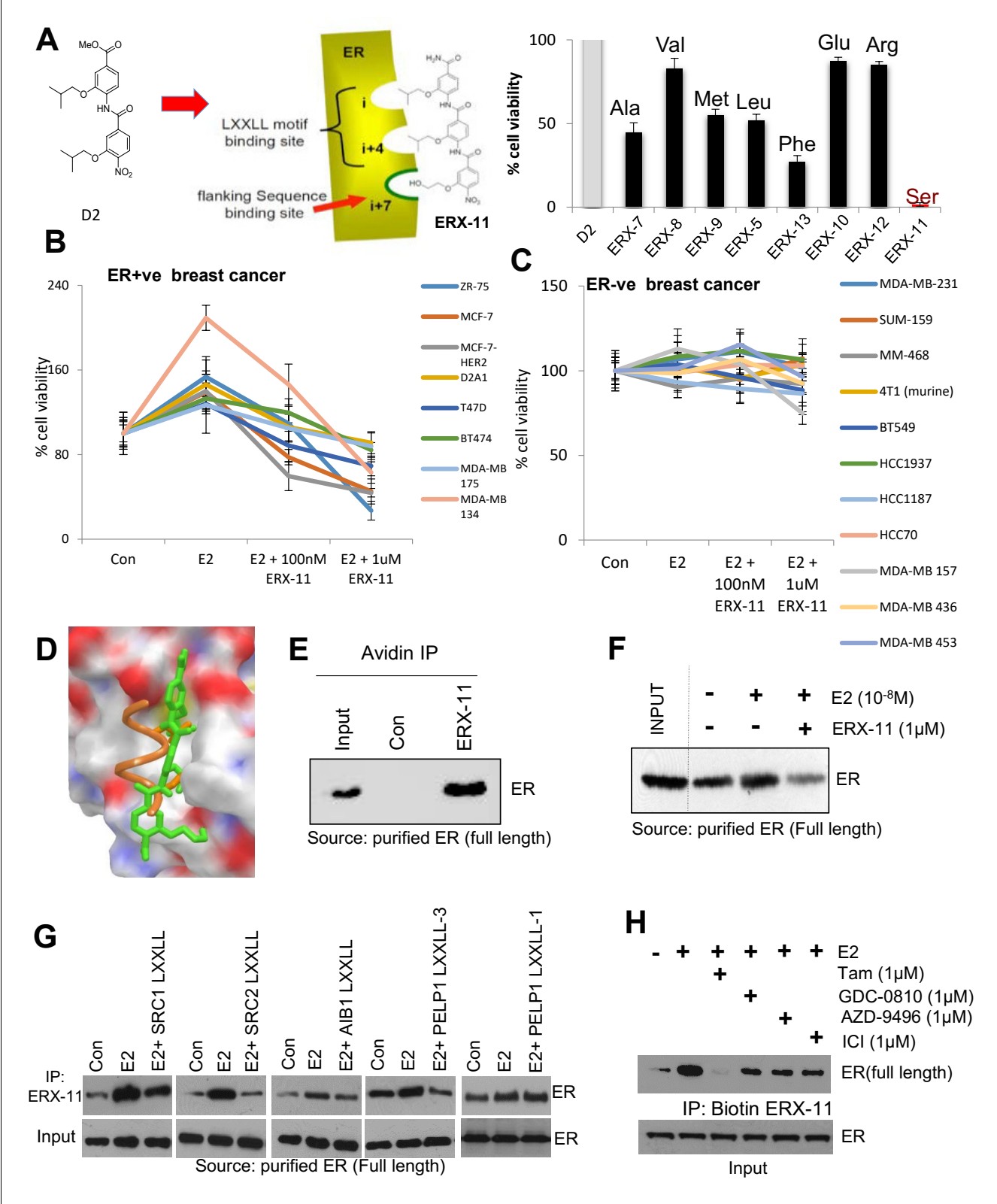

**Figure 1.** Derivation and characterization of ERX-11. Structure of ERX-11, as a derivative of the D2 peptidomimetic with a hydroxyethyl moiety in the flanking position to mimic a Serine (A, left panel). Effect of 500 nM of each peptidomimetic on the growth of MCF-7 cells using MTT cell viability assay is shown as percentage inhibition of the growth of E2-treated control cells (A, right panel). Effect of increasing doses of ERX-11 on the cell viability of ER-positive (B) and ER-negative (C) breast cancer cells using the MTT cell viability assay. Molecular docking studies on the interactions between ERX-11

*Figure 1 continued on next page*

*Figure 1 continued*

and ER using AutoDock Vina. Superimposition of the docked ERX-11 (green) on the crystal structure (PDB code 1L2I) of the LXXLL motif (orange) (**D**). Purified full-length ER was incubated with biotin-control or biotin-ERX-11 in the presence of E2. ERX-11 interaction with purified ER was analyzed using avidin bead pulldown and western blotting (**E**). Purified full-length ER was incubated with biotin-ERX-11 in the presence of E2 ± free ERX-11 (1 μM). ERX-11 ability to compete with the binding of biotin ERX-11 with ER was analyzed using avidin pulldown assay (**F**). Purified full-length ER was incubated with biotin-ERX-11 in the presence of E2 ±LXXLL peptides (1 μM) from various coregulators SRC1, SRC2, AIB1, and PELP1. LXXLL peptides ability to compete with the binding of biotin ERX-11 with ER was analyzed using avidin pulldown assay (**G**). Purified full-length ER was incubated with biotin-ERX-11 in the presence of E2 ± GDC0810, AZD-9496, ICI, and Tam (1 μM) and their ability to compete with the binding of biotin ERX-11 with ER was analyzed using avidin pulldown assay (**H**).

The following figure supplements are available for figure 1:

**Figure supplement 1.** Derivation of the α-helix mimetic ERX-11 and structural design, synthesis and activity of the α-helix mimetic.

**Figure supplement 2.** Characterization of ERX-11 activity.

**Figure supplement 3.** Derivation of the α-helix mimetic ERX-11 and synthesis of tris-benzamide peptidomimetics.

spectrometric (IPMS) analyses of the biotinylated ERX-11 pulldown in MCF-7 cells, identified ER as one of the top ERX-11 interactors (*Figure 2B*, *Table 1*). Pathways analysis revealed that ERX-11-binding proteins were involved in the activation of transcriptional regulation (*Figure 2—figure supplement 1*). Importantly, ERX-11 pulldown included a number of proteins other than ER, including a weak affinity for the progesterone receptor (PGR) and several ER-associated proteins (*Table 1*). Immunoprecipitation analyses in MCF-7 cells validated the strong affinity of ERX-11 for ER, and weak affinity for the PR-A isoform but not GR, AR or PR-B isoforms (*Figure 2C*).

The interaction between ER and ERX-11 within the cells was partially disrupted by high doses of tamoxifen (*Figure 2D*). Further, in the tamoxifen-resistant cell line, MCF-7-TamR, even high doses of tamoxifen could not disrupt the interaction between ERX-11 and ER (*Figure 1—figure supplement 3E*). The differences between these results and the in vitro results may be attributed to the context in which ER is presented within the cell.

Using GST-fused ER domain constructs, we validated that ERX-11 interact with the GST-AF2 domain of ER but not with the GST-AF1 or GST-DNA-binding domain of ER (*Figure 2E*). Further, ER-AF2 interaction with ERX-11 was disrupted by tamoxifen but not ICI (*Figure 2F*). These data clearly establish the interaction between ER and ERX-11 through the AF-2 domain.

## ERX-11 blocks ER interactions with coregulators

Using an unbiased approach with IPMS, we showed that ERX-11 significantly disrupted the interactions of 91 nuclear ER-binding proteins with ER in MCF-7 cells (*Figure 2—figure supplement 2A*), including well-characterized ER coregulators, such as SRC1, SRC3, and PELP1. Global analyses revealed that these proteins may be involved in a number of critical cellular pathways including transcription, cell cycle and regulation of cell death (*Table 2*). These findings were validated by IPMS studies in ZR-75 cells, which showed a significant overlap with MCF-7 cells in the coregulators disrupted by ERX-11 (*Figure 2—figure supplement 2B*). Of the top 10 coregulators, whose interactions with ER were negatively influenced by ERX-11, five contained LXXLL motifs with serine at i-3/4 and i+7/8 flanking position of the LXXLL motifs *Table 3*. Interestingly in the MDA-MB-231 TNBC model cells, we found that biotinylated ERX-11 was able to stringently interact only with a small number of proteins (n = 8) (*Figure 2—figure supplement 2C*).

In MCF-7 cells, a majority of ER-binding proteins disrupted by ERX-11 were also blocked by tamoxifen (55/88 proteins or 62.5%) (*Figure 2G* and *Figure 2—figure supplement 3*). Importantly, a significant number of ER-binding proteins were disrupted by ERX-11 but not tamoxifen (33/88 or 37.5%) (*Figure 2G* and *Figure 2—figure supplement 3*). The combination of tamoxifen and ERX-11 had significant overlap with ERX-11 in its ability to block ER-binding proteins (*Figure 2—figure supplement 3*).

Co-immunoprecipitation studies validated that endogenous complexes containing ER and coregulator PELP1 do form in MCF-7 cells and that ERX-11 disrupts the formation of these complexes

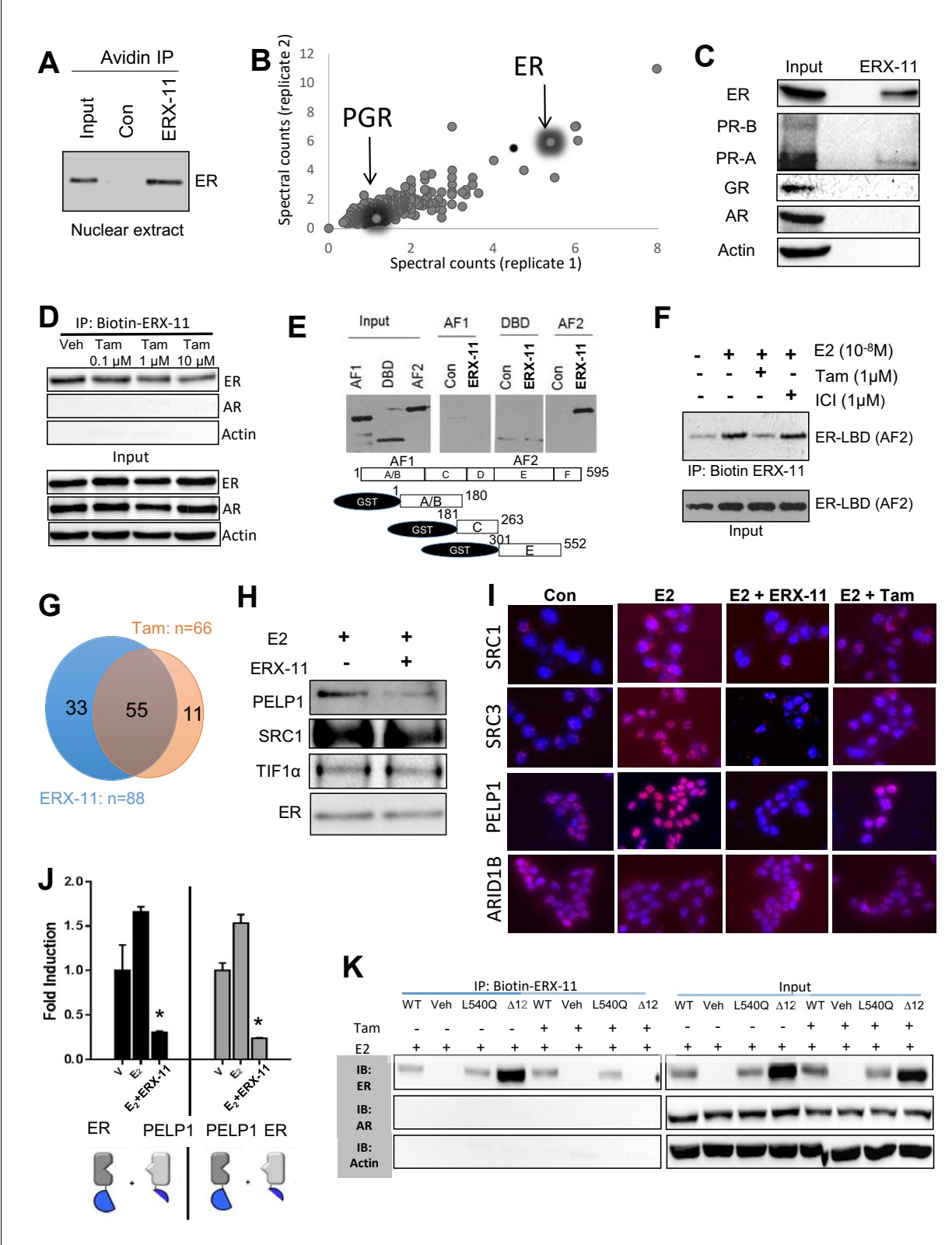

**Figure 2.** ERX-11 interacts with ER and blocks its interactome. Interaction with endogenous ER was evaluated in nuclear lysates prepared from ZR-75 cells stimulated with E2, incubated with biotin-control or biotin-ERX-11 and analyzed by avidin pull-down assay (**A**). Nuclear lysates from MCF-7 cells were incubated with biotinylated ERX-11 for 2 hr and then subject to a streptavidin column. The bound proteins were eluted and subjected to analyses by mass spectroscopy. The fold-enrichment in binding over basal is depicted for two independent replicates (x axis = replicate one and y

*Figure 2 continued on next page*

*Figure 2 continued*

axis = replicate 2). Relative binding of ER and PGR are shown (**B**). MCF-7 nuclear lysates were incubated with biotinylated ERX-11 and then subject to a streptavidin column. The bound proteins were eluted and evaluated by western blotting compared to equivalent amount of input (**C**). ZR-75 cells were incubated with tamoxifen and its ability to interfere ERX-11 binding to ER was analyzed by immunoprecipitation followed by western blotting (**D**). To confirm ERX-11 binding to ER-ligand-binding domain (AF2), a GST pull-down assay was performed. Biotinylated ERX-11 interacted with the GST-AF2 domain of ER but not with the GST-AF1- or GST- DNA-binding domain of ER (**E**). Purified ER-AF2 domain was incubated with biotin-ERX-11 in the presence of E2 ±ICI or tamoxifen (1 μM). ICI and tamoxifen ability to compete with binding of biotin ERX-11 with ER-AF2 domain was analyzed using avidin pull-down and western blotting (**F**). Venn diagram shows the overlap between the ER-binding proteins immunoprecipitated from nuclear lysates from E2-stimulated MCF-7 cells following treatment with vehicle or ERX-11 or tamoxifen (**G**). Co-immunoprecipitation analyses show the effect of ERX-11 on the interaction of ER with coregulators PELP1, SRC1, TIF1α in MCF-7 cells (**H**). Proximity ligation assay validated the ability of ERX-11 and tamoxifen to disrupt the interactions between ER and coregulators such as PELP1, SRC1, SRC3/AIB1 and ARID1B in MCF-7 cells (**I**). NanoBiT assay: expression plasmids were created to express either ER or PELP1 in conjunction with the large bit or the small bit of the NanoBiT luciferase enzyme. If the proteins directly interact within the cell, the two parts of the NanoBiT luciferase enzyme come together and create a quantifiable luminescent signal. The effect of ERX-11 on the interaction between the two sets of ER and PELP1 constructs is shown (**J**). Validation of the binding of ERX-11 to the ER-AF2 domain was further explored using AF2-domain mutants of ER stably transfected in ER-negative MDA-MB-231 breast cancer cell lines. Biotinylated ERX-11 was then used to pull-down ER from these cell lines (**K**). Data shown are the means of ±SEM performed in triplicate wells. *p<0.05.

The following figure supplements are available for figure 2:

**Figure supplement 1.** Pathways analysis in terms of either biological processes or molecular functions revealed that ERX-11-binding proteins were involved in the activation of multiple pathways leading to transcriptional regulation.

**Figure supplement 2.** Characterization of ERX-11 interactions with ER.

**Figure supplement 3.** Analyses of the ER-binding proteins blocked by ERX-11 or Tamoxifen.

**Figure supplement 4.** Effect of ERX-11 on inhibition of ER -coregulators interactions.

**Figure supplement 5.** Effect of SERDs or tamoxifen on ERX-11 interactions with ER.

**Figure supplement 6.** Models showing the putative interactions of ERX-11 with residues in the ER protein.

**Figure supplement 7.** Model describing interaction between ER (purple) and ERX-11 (green) in presence of agonist (yellow) (**A**), SERD (orange) (**B**) or tamoxifen (red)(**C**).

(*Figure 2H*). Proximity ligation assays confirmed the disruption of the endogenous interaction between ER and several coregulators including PELP1, SRC1 and SRC3 (*Figure 2I*, quantitation in *Figure 2—figure supplement 4A*). In contrast, ERX-11 has no effect on ER interaction with ARID1B (*Figure 2I*, quantitation in *Figure 2—figure supplement 4B*). Using an in vivo structural complementation NanoBiT assay, we found that the direct interaction between ER and PELP1 was enhanced by E2 treatment and that ERX-11 significantly reduced the interaction between ER and PELP1 (*Figure 2J*).

To confirm the specificity of ERX-11 for the ER AF-2 domain, we demonstrated using reporter-based assays, that ERX-11 failed to reduce the ERE-Luc reporter activity driven by a ERα-VP16 chimera that does not require AF-2 (*Figure 2—figure supplement 4C*). As expected, tamoxifen, did not affect the ERα-VP16 chimera-induced reporter activity, while ICI reduced the ERE-Luc reporter activity. Evaluation with an endometrial cancer cell line Ishikawa, which exhibits agonist activity via AF1, revealed that ERX-11 lacks AF1 agonist activity (*Figure 2—figure supplement 4D*). Collectively, these results confirm that the ERX-11 block signaling driven by functional AF2 domain but not by AF1 domain.

We then specifically evaluated whether interactions through the ER LXXLL motif was responsible for ERX-11 activity. Biotinylated ERX-11 was able to pull down both the wild-type ER and the L540Q point mutant ER (which retains E2 binding and does not interact with SRC1) and these interactions were not affected by tamoxifen (*Figure 2K*). Using ER L540Q point mutant, we showed that the mutant ER still interacts with biotinylated ERX-11 (*Figure 2K*). Interestingly, the ERX-11 binds strongly to ER▲12 mutant (helix 12 deleted ER) and this binding is blocked efficiently by tamoxifen

**Table 1.** Top proteins pulled down by biotinylated ERX-11 in MCF-7 cells, as identified by IP-MS. The column marked E2 represents spectral counts for the protein bound to biotinylated control eluted from avidin column, under conditions of E2 stimulation. The column marked E2 +ERX-11 represents spectral counts for proteins bound to biotinylated ERX-11 eluted from an avidin column with E2 stimulation. The column marked E2 + ERX-11/E2 represents the ratio of binders.

| Protein | Description | Length (AA) | Mw (Da) | PSMs | Peptide seqs | % Coverage | E2 | E2 + ERX-11 | E2 + ERX-11/ E2 |
|---|---|---|---|---|---|---|---|---|---|
| P38117 | ETFB_HUMAN Electron transfer flavoprotein subunit beta OS = *Homo sapiens* GN = ETFB PE = 1 SV = 3 | 255 | 37501.10 | 12 | 7 | 34.10 | 1 | 7.00 | 5.00 |
| Q96PZ0 | PUS7_HUMAN Pseudouridylate synthase seven homolog OS = *Homo sapiens* GN = PUS7 PE = 1 SV = 2 | 661 | 75186.30 | 20 | 14 | 26.30 | 1.00 | 7.98 | 10.99 |
| O95336 | 6 PGL_HUMAN 6-phosphogluconolactonase OS = *Homo sapiens* GN = PGLS PE = 1 SV = 2 | 258 | 27601.60 | 13 | 8 | 41.50 | 0.99 | 6.00 | 6.04 |
| Q8TD06 | AGR3_HUMAN Anterior gradient protein three homolog OS = *Homo sapiens* GN = AGR3 PE = 1 SV = 1 | 166 | 19194.90 | 14 | 9 | 47.00 | 0.99 | 5.96 | 7.03 |
| P18754 | RCC1_HUMAN Regulator of chromosome condensation OS = *Homo sapiens* GN = RCC1 PE = 1 SV = 1 | 421 | 48241.20 | 27 | 13 | 55.30 | 1.99 | 11.92 | 7.00 |
| O60506 | HNRPQ_HUMAN Heterogeneous nuclear ribonucleoprotein Q OS = *Homo sapiens* GN = SYNCRIP PE = 1 SV = 2 | 623 | 69739.70 | 20 | 20 | 48.50 | 1.97 | 10.82 | 3.50 |
| P03372 | ESR1_HUMAN Estrogen receptor OS = *Homo sapiens* GN = ESR1 PE = 1 SV = 2 | 595 | 66335.20 | 36 | 15 | 30.80 | 1.83 | 9.88 | 5.93 |
| E9PCR7 | E9PCR7_HUMAN 2-oxoglutarate dehydrogenase, mitochondrial OS = *Homo sapiens* GN = OGDH PE = 2 SV = 1 | 1038 | 115728.00 | 39 | 25 | 32.80 | 3.98 | 18.88 | 4.00 |
| O43488 | ARK72_HUMAN Aflatoxin B1 aldehyde reductase member 2 OS = *Homo sapiens* GN = AKR7A2 PE = 1 SV = 3 | 359 | 39653.80 | 22 | 11 | 39.80 | 1.99 | 8.96 | 5.51 |
| O95994 | AGR2_HUMAN Anterior gradient protein two homolog OS = *Homo sapiens* GN = AGR2 PE = 1 SV = 1 | 175 | 22277.70 | 29 | 12 | 65.70 | 2.97 | 11.92 | 4.68 |
| P19338 | NUCL_HUMAN Nucleolin OS = *Homo sapiens* GN = NCL PE = 1 SV = 3 | 710 | 76766.50 | 99 | 35 | 48.00 | 14.00 | 50.98 | 2.43 |
| O43148 | MCES_HUMAN mRNA cap guanine-N7 methyltransferase OS = *Homo sapiens* GN = RNMT PE = 1 SV = 1 | 476 | 57831.90 | 16 | 9 | 29.40 | 1.99 | 6.97 | 3.50 |
| Q562R1 | ACTBL_HUMAN Beta-actin-like protein 2 OS = *Homo sapiens* GN = ACTBL2 PE = 1 SV = 2 | 376 | 42084.00 | 14 | 14 | 39.10 | 2.00 | 7.00 | 3.00 |
| Q9Y5A9 | YTHD2_HUMAN YTH domain family protein 2 OS = *Homo sapiens* GN = YTHDF2 PE = 1 SV = 2 | 579 | 62457.80 | 15 | 12 | 23.10 | 2.00 | 6.99 | 3.00 |
| P16152 | CBR1_HUMAN Carbonyl reductase [NADPH] 1 OS = *Homo sapiens* GN = CBR1 PE = 1 SV = 3 | 277 | 30427.90 | 20 | 11 | 56.70 | 2.99 | 9.98 | 2.33 |
| Q9UBS4 | DJB11_HUMAN DnaJ homolog subfamily B member 11 OS = *Homo sapiens* GN = DNAJB11 PE = 1 SV = 1 | 358 | 40578.70 | 21 | 12 | 35.20 | 3.00 | 10.00 | 2.67 |

(*Figure 2K*). These data suggest that the presence of helix 12 may regulate the conformation of the binding pocket and account for differences in the binding of ERX-11 and tamoxifen to ER. Our data would suggest that removal of helix 12 enables access of ERX-11 to the same binding pocket as tamoxifen and may reflect the in vitro data, where tamoxifen and ERX-11 compete efficiently for ER binding. In contrast, neither GDC-0810 nor AZD-9496 were able to block ERX-11 binding to ER or its mutants, suggesting that their binding to ER occurs through distinct pockets (*Figure 2—figure supplement 5A and B*).

In competition assays, tamoxifen fails to dislodge ERX-11 from ER (*Figure 2—figure supplement 5C*, *Figure 1—figure supplement 3E*). Increasing concentrations of tamoxifen is only able to dislodge ERX-11 from ER▲12 mutant at higher concentrations, suggesting that ERX-11 interaction with ER is through a second binding site (*Figure 2—figure supplement 5D*).

Further, using an ER restoration model, MTT cell viability assays revealed that introduction of either ER, ER▲12 or ER-L540Q into MDA-MB-231 cells, restored ERX-11 growth inhibitory activity in

**Table 2.** Top biological processes of coregulators, whose interactions with ER are disrupted by ERX-11 in MCF-7 cells.

| Biology processes | Genes |
|---|---|
| RNA processing | CD2BP2 CHERP CPSF1 CPSF2 CPSF3L CSTF3 DDX17 DDX20DDX23 DHX15 DHX9 DKC1 GEMIN5 HNRNPA3 HNRNPK HNRNPLL HNRNPLL HNRNPR INTS2 INTS4 INTS5 NCBP1 PCF11 POLR2A PPP2R1A PRPF31 PRPF40a PUF60 RBM10 RBM14 SART1 SF1 SF3A3 SF3B1 SF3B3 Sfrs15 SKIV2L2 SMC1A SRRM1 SSB SYNCRIP THOC2 TRNT1 U2AF2 XRN2 ZCCHC8 |
| Transcription | ADNP CCNL1 CSDA CTNND2 DIDO1 DMAP1 EIF2S2 ERCC2 FOXA1 GTF2I GTF3C1 GTF3C KDM3B KDM5B LRPPRC MCM2 MED1 MED24 NCOA3 PELP1 POLR2A POLR3C PSIP1 PUF60 RBM14 RFX1 SAP130 SF1 SMARCA2 SMARCA4 SMARCC2 SMARCD2 THRAP3 Th1l TRIM33 UHRF1 XRN2 ZBTB7A ZMYM2 ZNF217 ZNF512B |
| Protein transport, protein localization | AP2A2 CLTC COG1 COG3 COG5 COG8 COPB1 COPB2 COPG2 CSE1L EXOC2 EXOC3 EXOC4 EXOC5 EXOC8 IPO4 KPNA4 KPNB1 NUP153 NUP155 NUP93 RANBP2 SEC16A SEC23A SEC24B SEC24C SRP72 STXBP2 TNPO1 TRAM1 TRNT1 VCP VPS11 VPS18 VPS39 |
| RNA splicing | CD2BP2 CPSF1 CPSF2 CSTF3 DDX20 DDX23 DHX15 DHX9 GEMIN5 HNRNPA3 HNRNPL HNRNPR LUC7L3 NCBP1 PCF11 POLR2A PPP2R1A PRPF31 PRPF40A PUF60 RBM10 SART1 SF1 SF3A3 SF3B1 SF3B3 SKIV2L2 SMC1A SRRM1 SYNCRIP THOC2 U2AF2 ZCCHC8 |
| Macromolecular complex subunit organization | CSE1L DARS DDX20 DDX23 EPRS ERCC2 FKBP4 GEMIN5 GTF2I GTF3C4 HSP90AA1 IPO4 KPNB1 LONP1 MCM2 MED1 MED24 NCBP1 PFKL PFKM PFKP POLR2A PPP2R1A PREX1 PRPF31 SF1 SF3A3 SF3B3 THRAP3 TNPO1 TUBA1B TUBB VCP XRN2 |
| Cell cycle | CUL1 CUL2 CUL3 CUL4B Dmc1 DNM2 DYNC1H1 EIF4G2 LIG3 MCM2 MRE11A NUMA1 PDS5B PHGDH PPP3CA PSMC1 PSMC4 PSMD3 PSMD5 RAD50 SART1 SF1 SMC1A SMC3 SMC4 TUBB UHRF1 |
| Chromosome organization | PDS5B KDM5B RBM14 RAD50 MRE11A CHD1L DMAP1 DKC1 EP400 KDM3B MCM2 SAP130 SMC1A SMC3 SMC4 SMCHD1 SMARCA2 SMARCA4 SMARCC2 SMARCD2 |
| Regulation of cell death | ACTN1 ADNP CSDA CUL1 CUL2 CUL3 DDX20 DNM2 ERCC2 PPP2R1A PREX1 SART1 SCRIB TUBB UACA VCP |

non-responsive MDA-MB-231 cells. These results further underscore the importance of ER in ERX-11 mode of action (*Figure 2—figure supplement 5E*).

## Docking simulations model ERX-11 binding to ER

We have modeled ERX-11 interaction with ER using docking simulations of ERX-11 using known crystal structures (*Figure 2—figure supplement 6*). In the agonist-bound conformation (3ERD.pdb), helix 12 is relocated and forms a hydrophobic cleft (i.e. AF2-binding site) that is surrounded by helices 3, 4, 5 and 12: ERX-11 can be modeled to make hydrophobic contact with the AF2 site with its two isobutyl side chain groups (*Figure 2—figure supplement 6A(A)*). In addition, the hydroxyl

**Table 3.** Selected pathways modulated by ERX-11 treatment. Differentially expressed genes were subjected to pathway analysis using IPA software and the selected top canonical pathways modulated by ERX-11 are shown. This data is related to *Figure 3*.

| Pathway | p-Value | Ratio | Genes |
|---|---|---|---|
| Retinoic-acid-Mediated Apoptosis Signaling | 2.44E + 00 | 1.25E-01 | PARP12,ZC3HAV1,TNFSF10,PARP9,PARP14,CRABP2,RARG,CRABP1 |
| ERK/MAPK Signaling | 1.77E + 00 | 7.49E-02 | SRC,MKNK2,PLA2G4F,DUSP2,BAD,ELF5,PPM1J,PPP1R14B,STAT1,RAC3,ELF4,PPP2R1A,RRAS,RPS6KA4 |
| Cyclins and Cell Cycle Regulation | 1.46E + 00 | 8.97E-02 | HDAC5,TGFB1,PPM1J,PPP2R1A,E2F1,HDAC11,HDAC7 |
| Death Receptor Signaling | 1.53E + 00 | 8.70E-02 | PARP12,ACTG1,ZC3HAV1,LIMK1,TNFSF10,PARP9,PARP14,BIRC3 |
| Inhibition of Matrix Metalloproteases | 2.40E + 00 | 1.54E-01 | HSPG2,MMP10,TIMP1,MMP13,MMP15,SDC1 |
| Estrogen Receptor Signaling | 1.19E + 00 | 7.09E-02 | KAT2B,ERCC2,SRC,G6PC3,TAF6,MED24,TAF6L,RRAS,MED15 |
| Breast Cancer Regulation by Stathmin1 | 4.63E-01 | 4.71E-02 | ADCY1,ARHGEF19,PPM1J,PPP1R14B,LIMK1,PPP2R1A,E2F1,TUBA4A,RRAS |

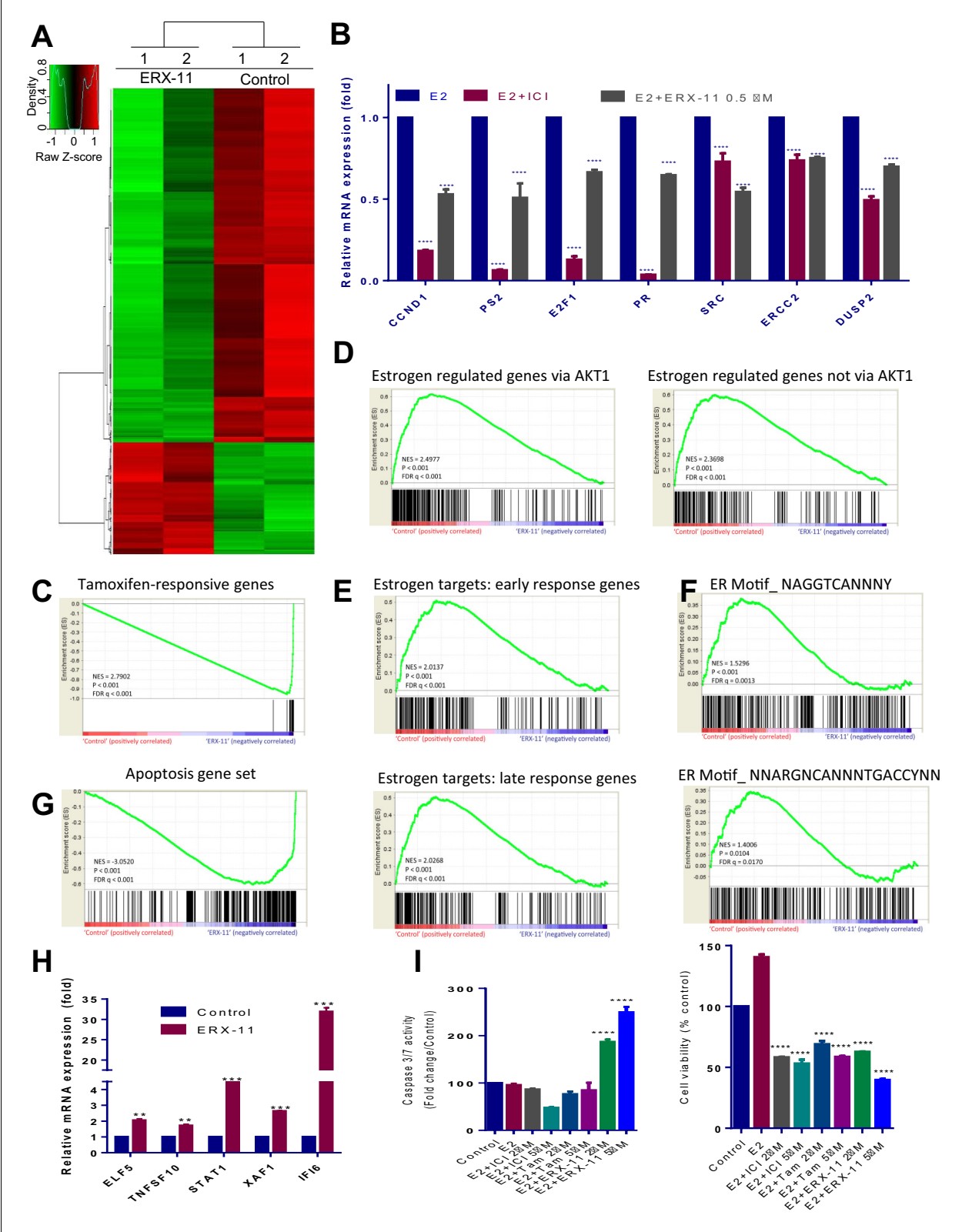

**Figure 3.** ERX-11 globally disrupts ER-mediated transcriptome. Total RNA was isolated from the ZR-75 cells that were treated with either vehicle or ERX-11 for 48 hr and subjected to RNA sequencing. The heat map of differentially expressed genes between vehicle and ERX-11 is shown (**A**). ZR-75 cells were treated with either vehicle or ERX-11 for 48 hr, and the selective genes representing each pathway were validated using RTqPCR (**B**). Gene set enrichment analysis (GSEA) testing correlation between ERX-11–regulated gene and both the tamoxifen-responsive (M3283) and estradiol-

*Figure 3 continued on next page*

*Figure 3 continued*

responsive genes set (M2234 and M2230) (**C, D**). GSEA analysis testing the correlation of ERX-11-regulated genes with signatures of early (M5906) and late (M5907) response estrogen targets as well as genes driven by consensus ER motifs (M17968 and M6101) (**E, F**). GSEA analysis of correlation of ERX-11 regulated gene set with an apoptotic gene set (M15912) (**G**). ERX-11 upregulated apoptotic genes were validated by RT-qPCR analyses (**H**). Data are represented as mean ±SEM. **p<0.01; ***p<0.001. Effect of indicated doses of Tam, ICI and ERX-11 on cell viability (CellTiter-Gloassay, Promega) and caspase3/7 activity (Caspase-Glo 3/7 Assay, Promega) in ZR-75cells (**I**). Data shown are the means of ±SEM performed in triplicate wells. ****p<0.0001.

The following figure supplement is available for figure 3:

**Figure supplement 1.** ERX-11 treatment has potential to promote apoptosis.

group of ERX-11 may interact with a residue near AF2 domain. These data suggest that ERX-11 interacts with ER LBD differently than the agonist.

In the antagonist-bound conformation (3ERT.pdb), 4-hydroxytamoxifen induces conformational change and makes helix 12 occupy the AF2-binding site, blocking both coactivator and corepressor binding: here, ERX-11 may interact with an alternate pocket formed by helices 5, 11 and 12, as shown in the *Figure 2—figure supplement 6A(B)*. These data could explain why the interaction between ERX-11 and purified ER could be blocked by tamoxifen. However, in a cellular context, tamoxifen cannot block ERX-11 binding to ER suggesting that the secondary binding site of ERX-11 on ER may be stabilized by coregulators. Docking simulation of ERX-11 on human ERα with affinity-tagged corepressor peptide (*Figure 2—figure supplement 6A(C)*)(2JFA.pdb) or rat ERβ crystal structure with ICI bound (*Figure 2—figure supplement 6A(D)*)(1HJ1.pdb) showed that ERX-11 can still bind to the AF2 domain, in a similar manner as it does when the ligand is bound. These data further support our biochemical findings that ICI does not block ERX-11 interaction with ER.

Detailed evaluation showed that (1L2I.pdb) two isobutyl groups of ERX-11 may dock into the AF2 binding site (black dashed box) (*Figure 2—figure supplement 6B*). The hydroxyl group of ERX-11 may interact with Gln375 of the helix five through a hydrogen bond. The carboxamide group of ERX-11 was docked into a pocket formed with Gln542, Asp545 and Ala546 on the helix 12. Additional docking experiment on ER crystal structures without helix 12 (3ERT.pdb and 5ACC.pdb) showed that ERX-11 was found to bind to the tamoxifen-binding site in the ER▲12 (*Figure 2—figure supplement 6C(A)*). The superimposition of tamoxifen (red) and ERX-11 (green) clearly shows the overlap of their binding sites (3ERT.pdb) (*Figure 2—figure supplement 6C(B)*). This may explain our experimental results showing the competition of tamoxifen with ERX-11 on ER▲12. In the presence of SERDs, ERX-11 can bind to the AF2 domain (*Figure 2—figure supplement 6C(C)*), which do not overlap with the binding site of SERDs (*Figure 2—figure supplement 6C(D)*) (5ACC.pdb). A model to explain the potential interactions between ER and ERX-11 or between ER▲12 mutant and ERX-11 in the absence or presence of agonist, SERDs or tamoxifen is included (*Figure 2—figure supplement 7*).

## ERX-11 blocks ER-driven breast cancer signaling pathways

RNA-seq analyses revealed that ERX-11 altered the expression of 880 E2-regulated genes (p<0.01) in ZR-75 cells compared to vehicle control (*Figure 3A*). Using stringent cutoffs (p<0.01 and RPKM FC > 1.5), ERX-11 down-regulated more genes (669) than upregulated (211) as evidenced by the volcano plot (*Figure 3—figure supplement 1A and B*) (complete list at GEO database, accession number GSE75664). RT-qPCR analyses validated the expression of the top down-regulated (*Figure 3B* and *Figure 3—figure supplement 1C*) and up-regulated genes (*Figure 3H*). Gene set enrichment analysis (GSEA) confirmed the correlation between ERX-11–regulated genes and both the tamoxifen-responsive and E2-responsive genes set (*Figure 3C and D*). In addition, ERX-11-regulated genes correlated well with signatures of early and late response E2 targets as well as genes driven by consensus ER motifs (*Figure 3E and F*).

Ingenuity pathway analysis (IPA) revealed that ERX-11 significantly down-regulated genes involved in ER-signaling, breast cancer, cell cycle, and MAPK signaling (*Table 3*). ERX-11 upregulated gene set positively correlated with apoptotic genes, on GSEA analyses (*Figure 3G*). Importantly, ERX-11 but not tamoxifen or ICI-treatment-induced apoptosis in ZR-75 (*Figure 3I*) and in T-47D cells (*Figure 3—figure supplement 1D*) as shown by induction of caspase 3/7 activity.

Apoptosis can be seen as early as 24 hr, however, the effect is more pronounced at 72 hr (*Figure 3—figure supplement 1E*). These results suggest that ERX-11 both reduces the expression of genes involved in proliferation and enhances expression of genes that promote apoptosis.

## ERX-11 inhibits ER-mediated transcription

Using ERE-Luc reporter based transcription assays, we found that ERX-11 significantly reduced the E2-induced ERE-Luc reporter gene activity in ZR-75 cells in a similar fashion as tamoxifen (*Figure 4A*). In HEK-293T cells, expression of AIB1 and SRC1 enhanced ER-driven ERE-Luc reporter activity and was blocked by both tamoxifen and ERX-11 in both a ligand-dependent (*Figure 4B*) and ligand-independent manner (*Figure 4C*). Further, ZR-75 cells stably overexpressing either PELP1 or AIB-1 were responsive to ERX-11 (*Figure 4D*). These results indicate that ERX-11 interferes with both ligand-dependent and ligand-independent transcriptional function of ER.

Using chromatin immunoprecipitation studies, we found that ERX-11 treatment significantly blocked ER recruitment to canonical ER target gene promoters following E2 treatment (*Figure 4E*). In contrast, ERX-11 did not affect AR recruitment to AR target genes (*Figure 4—figure supplement 1A*). Since ERX-11 binds to ER, we hypothesized that the effect of ERX-11 on ER DNA binding may be mediated via disruption of ER dimerization. Using the NanoBiT assay, we demonstrated that ERX-11 efficiently blocks the dimerization of ER (*Figure 4F*). In contrast, ERX-11 did not affect AR dimerization (*Figure 4—figure supplement 1B*).

Using E2 dendrimer conjugates (EDC), that are potent in activating ER non-genomic signaling but not ER genomic signaling (*Chakravarty et al., 2010*), we showed that ERX-11 was unable to influence the EDC-mediated non-genomic activation of the Src, AKT and MAPK pathways (*Figure 4—figure supplement 1C*). A detailed time course evaluation showed that ERX-11 treatment only modestly altered the stability of ER within 24 hr in ZR-75 and T-47D cells (*Figure 4G*, *Figure 4—figure supplement 1D*). However, similar to its inhibitory activity on ER transcription, after several days of ERX-11 treatment, decreased ER levels were detected (*Figure 4H*). Accordingly, RTqPCR results showed that ERX-11 reduce the ER transcript levels under conditions of long-term treatment (7 days). These results reflect the inhibition of ER signaling indirectly affected autoregulation of ER transcript by E2-ER signaling (*Figure 4—figure supplement 1E*).

## ERX-11 suppresses ER-driven breast tumor growth in vivo

Our prior studies indicated that our peptidomimetics are orally bioavailable (*Ravindranathan et al., 2013*). We detected no overt signs of toxicity after 14 days of treatment of C57BL/6 mice (n = 3) with 10, 50 or 100 mg/kg/day of ERX-11 via oral gavage. ERX-11 treatment neither caused weight loss nor have uterotrophic effects or any observable hematologic, liver and kidney abnormalities (data not shown). We designated our highest dose (100 mg/kg) as the maximum tolerated dose and used 10% of this dose (10 mg/kg/day) for testing as therapeutic dose, so that we would have at least a 10:1 therapeutic to toxicity ratio.

Established ZR-75 xenografts (n = 8 tumors/group) in the mammary fat pad of nude mice were randomized to feed via oral gavage 5 days/week with either 10 mg/kg ERX-11 or vehicle (30% Captisol). ERX-11 treatment resulted in significantly smaller tumors (63% reduction compared to control) (*Figure 5A*). ERX-11–treated tumors exhibited less proliferation (Ki67 staining), and more apoptosis (TUNEL and caspase-3 staining) than controls (*Figure 5A,B*). Further, ERX-11 treatment group had lower ER but similar PELP1 protein expression levels within the tumor, compared to control (*Figure 5B*). The mice body weights in the control and ERX-11 treated groups were similar (*Figure 5—figure supplement 1A*). Mice treated with ERX-11 exhibited no uterotrophic effects, no changes in ovary, liver and kidney gross morphology on H and E staining, or acute phase injury to liver and kidney (*Figure 5—figure supplement 2A–D*). These data indicate that ERX-11 is a potent inhibitor of the growth of ER-positive breast tumors in vivo with no overt signs of toxicity in mice.

To address the potential immunogenicity of ERX-11, D2A1 ER-positive breast cancer xenografts were established in a syngeneic BALB/c model system with an intact immune system. In D2A1 model cells, cellular gene int-5/aromatase in BALB/c mammary alveolar hyperplastic nodule (D2 HAN/D2 tumor cells) is activated as a result of mouse mammary tumor virus integration within the 3' untranslated region of the aromatase gene. Thus, these models also have ability to synthesize local estrogen via aromatase induction. Further, this model express ER, and represent a model of intra-tumoral

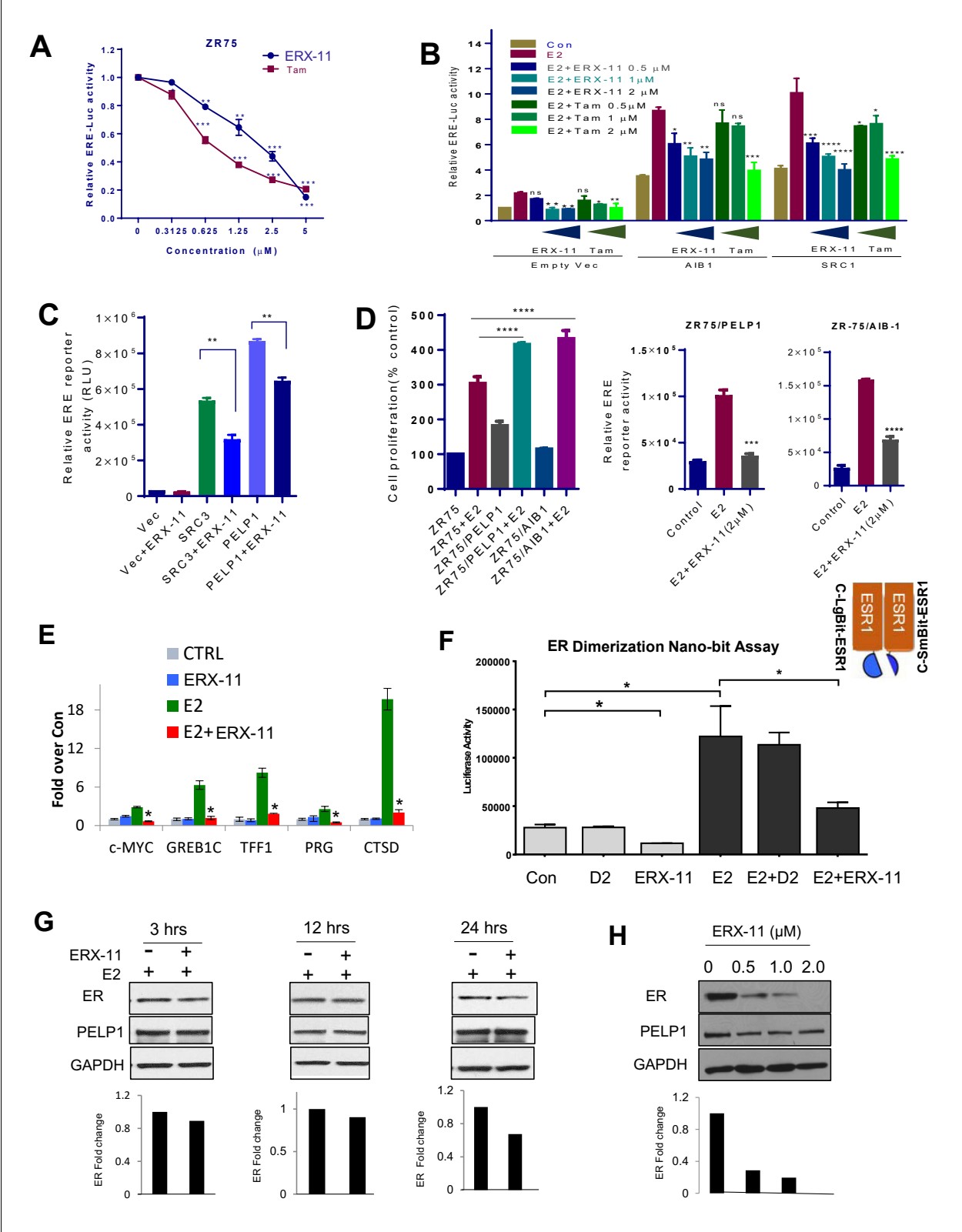

**Figure 4.** ERX-11 affects ER ligand-dependent and independent transcriptional activity. ZR-75 cells stably transfected with ER and ERE-Luc vectors were treated with E2 ($10^{-8}$M) in the presence of indicated concentrations of ERX-11 or tamoxifen. After 24 hr, the reporter gene activity was measured (**A**). HEK-293T cells stably transfected with ERE-Luc vector were transiently transfected with control or coregulator-expressing vector along with ER expression vector and after 24 hr treated with indicated doses of ERX-11 or tamoxifen along with E2 ($10^{-8}$M). After 24 hr, the reporter gene activity was

*Figure 4 continued on next page*

*Figure 4 continued*

measured (**B**). HEK-293T cells stably transfected with ERE-Luc vector were transiently transfected with control or coregulator-expressing vectors along with ER expression vector and after 24 hr treated with ERX-11. After 24 hr, the reporter gene activity was measured (**C**). Cell proliferation of ZR-75 model cells stably expressing PELP1 or AIB1 was measured using Cell Titer Glo assay (D, left panel). ZR-75 model cells stably expressing PELP1 or AIB1 were transfected with ERE-Luc reporter vector. After 48 hr, the cells were treated with ERX-11 and the reporter activity was measured 24 hr later (D right panel). The effects of ERX-11 on ER recruitment at ER target genes were examined using a ChIP assay in MCF-7 cells (**E**). The effect of ERX-11 on ER dimerization as evaluated by the NanoBiT luciferase assay is shown (**F**). ZR-75 cells were treated with E2 with or without ERX-11 (1 µM) for the indicated time, and the stability of ER was determined using western blotting. Quantitation of ER levels compared to control (E2-treated cells) was shown after normalizing to the levels of GAPDH (**G**). ZR-75 cells were treated with ERX-11 for 5 days, and the status of ER was determined using Western blotting (**H**). Data shown are the means of ±SEM performed in triplicate wells. *p<0.05; **p<0.01; ***p<0.001; ****p<0.0001.

The following figure supplement is available for figure 4:

**Figure supplement 1.** Effect of ERX-11 on AR functions, E2 mediated non-genomic actions and ER stability.

estrogen-driven mammary cancer (*Figure 5—figure supplement 1B(A,C)*). D2A1 cells are responsive to antiestrogen treatment (*Figure 5—figure supplement 1B(B)*). Oral administration of ERX-11 dramatically limited the proliferation of these rapidly progressing tumors (*Figure 5C*). The proliferative indices of ERX-11-treated tumors were significantly lower than controls (*Figure 5C*, *Figure 5—figure supplement 1C*), while the apoptotic indices were higher than control (*Figure 5—figure supplement 1C*). Again, no overt signs of toxicity was noted in these mice; specifically, no enlargement of the spleen or evidence of immune complex deposition within the kidneys was detected (data not shown). These data further support the potential clinical translatability of ERX-11.

## ERX-11 reduces growth of therapy-resistant breast cancer cells

To evaluate the effect of ERX-11 on coregulator-driven proliferation, we used ZR-75 cells stably overexpressing AIB-1 and PELP1. While these modified ZR-75 cells are highly proliferative (*Figure 4D*), ERX-11 was potent in blocking their proliferation (*Figure 5D*). ERX-11 was potent (73% reduction in tumor volume compared to control) against the growth of MCF-7-PELP1 xenografts, which overexpress PELP1 (3-fold higher than parental MCF-7) (*Figure 5E*). IHC analysis of ERX-11 treated tumors showed decreased Ki-67 staining (*Figure 5E*, *Figure 5—figure supplement 1D*).

Importantly, ERX-11 had activity against ER-driven breast cancer cell lines that were either resistant to tamoxifen (*MCF-7-TamR, Figure 6A, or MCF-7-HER2, Figure 6B*) or to letrozole (*MCF-7-LTLT, Figure 6C*). In these cell lines, ERX-11 was still able to interact with the ER, both in the absence and presence of tamoxifen (*Figure 6D*). In contrast to SERD (ICI), which had limited activity on the tamoxifen resistant cell lines, ERX-11 had potent activity (*Figure 6A*) (*Figure 6—figure supplement 1A*). ERX-11 was potent against the growth of MCF-7-LTLT xenografts (*Figure 6E*). IHC analysis of ERX-11-treated tumors showed decreased Ki-67 staining (*Figure 6—figure supplement 1B*).

We then evaluated the effect of ERX-11 against two prevalent ER mutants (*MT-ESR1-Y537S, MT-ESR1-D538G*) (*Toy et al., 2013*; *Robinson et al., 2013*; *Jeselsohn et al., 2014*; *Merenbakh-Lamin et al., 2013*). Using biotinylated ERX-11, we showed that ERX-11 interacts directly with *ESR1*-MT (*ESR1-MT-D538G, ESR1-MT-Y537S*), with high affinity comparable to the affinity to WT-ER (*Figure 6F*). Using CRISPR/Cas9, we knocked down ER in ZR-75 cells and then stably transfected with *WT-ESR1* or *MT-ESR1* (Y537S, and D538G). While *ESR1-MT* expressing cells showed higher rates of proliferation than *WT-ESR1*-expressing cells (*Figure 6—figure supplement 1C*), they were still inhibited by ERX-11 (*Figure 6G*). Further the ability of these *ESR1-MT* to drive ligand-independent transcription from an ERE-Luc reporter was also efficiently blocked by ERX-11 (*Figure 6H*). Further, these *ESR1-MT* expressing cells were resistant to tamoxifen, however, were sensitive to ERX-11-mediated growth inhibition (*Figure 6I*). Further, oral ERX-11 administration had significant activity against the growth of ZR-75-*ESR1MT-Y537S* xenografts in vivo (*Figure 6J*), with significant reduction in proliferative indices (*Figure 6K*, *Figure 6—figure supplement 1D*). These data support the efficacy of ERX-11 against breast tumors driven by mutant *ESR1*.

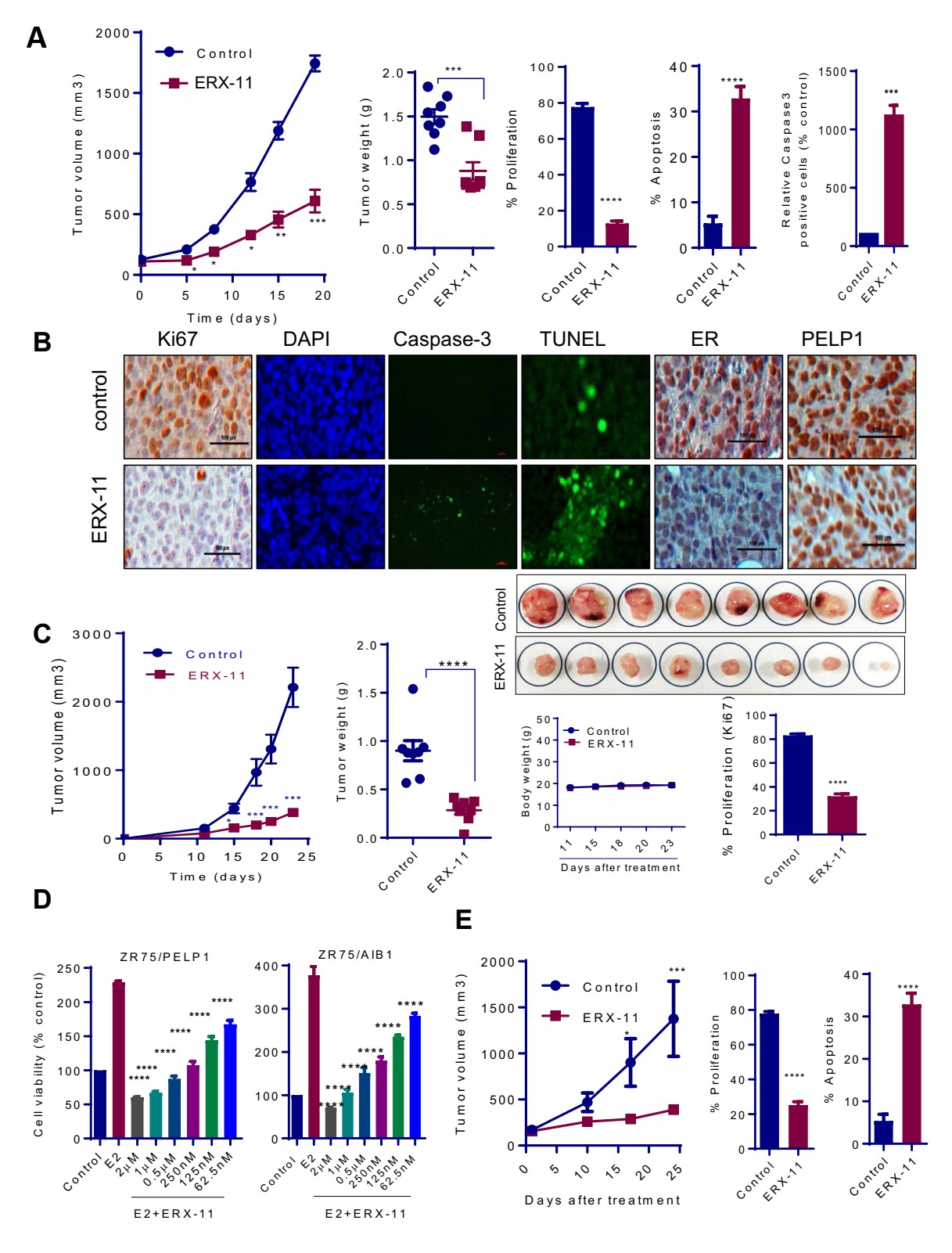

**Figure 5.** ERX-11 inhibits the growth of ER-positive, syngeneic and coregulator-driven breast tumors in vivo. ER-positive ZR-75 cells were injected into the mammary fat pads of nude mice implanted subcutaneously with E2 pellet. After 2 weeks, mice with xenografts were treated with vehicle or 10 mg/kg/day of ERX-11 (n = 8) by oral gavage. Tumor growth was measured at the indicated time points. Tumor volume is shown in the graph (A). The weights of the control or ERX-11-treated tumors at the time of necropsy are shown. Ki-67 expression as a marker of proliferation was analyzed by IHC

*Figure 5 continued on next page*

*Figure 5 continued*

and quantitated. Apoptosis was measured using Caspase3 activation and by using TUNEL assay, and the number of TUNEL-positive and cleaved caspase 3 cells were counted in five different fields and plotted as histogram. DAPI was used to visualize the nuclei (**A, B**). Representative IHC analysis of ER and PELP1 performed on xenograft tumors that were treated with or without ERX-11 (**B**). Effect of ERX-11 on the growth of ER-positive D2A1 syngeneic tumors. Small pieces of D2A1 syngeneic tumors were implanted subcutaneously into the BALB/c mice. After 1 week, mice (n = 8) were treated with vehicle or ERX-11 (20 mg/kg/day). Tumor growth was measured at indicated time points. The body weights and extirpated tumor weights are shown. Ki-67 expression was analyzed by IHC and quantitated (**C**). The effect of ERX-11 on the coregulator-driven cell survival was measured by MTT assay using ZR-75 cells stably expressing SRC3/AIB1 or PELP1 (**D**). MCF-7-PELP1 cells were injected into the mammary fat pad of nude mice (n = 5) implanted with an estrogen pellet. After 3 weeks, mice were treated with vehicle or ERX-11 (10 mg/kg/day). Tumor volume, status of Ki-67 and apoptosis was shown (**E**). Data shown are the means of ±SEM. *p<0.05, **p<0.01, ***p<0.001, ****p<0.0001.

The following figure supplements are available for figure 5:

**Figure supplement 1.** Characterization of ERX-11 treated tumors.

**Figure supplement 2.** Normal tissues collected from mice that were treated with vehicle or ERX-11 were examined for toxicity.

## ERX-11 has activity against primary patient derived breast tumor explants

We recently developed an ex vivo culture model of primary breast and prostate tumors, which allows for the evaluation of drugs on breast tumors while maintaining their native tissue architecture (*Dean et al., 2012*; *Schiewer et al., 2012*). In brief, surgically extirpated de-identified breast tissues are sliced into small pieces and grown ex vivo for a short term on a gelatin sponge in the absence or presence of ERX-11 (*Figure 7A*). Incubation of ERX-11 with ER-positive breast tumor samples (*patient characteristics detailed in Table 4*) dramatically decreased their proliferation in 11/12 patients (Ki67 staining) compared to untreated controls (*Figure 7B*). Further, ERX-11 treatment significantly reduced the ER staining (*Figure 7C*), but not PELP1 staining (*Figure 7—figure supplement 1*) in 12/12 ER-positive tumors. Importantly, ERX-11 treatment had no effect on the proliferation on 6/6 triple negative breast cancer (TNBC) tumors (*Figure 7D*). These results suggest that ERX-11 has the potential to selectively influence the growth of human breast tumors expressing ER.

## Discussion

The majority of breast cancer is ER positive. Therapeutic agents that suppress oncogenic ER activation by depletion of hormone-driven growth signaling or by blocking synthesis of hormone have become the mainstay of systemic treatment for breast cancer. However, both de novo and acquired therapy resistance are major clinical challenges. Importantly, ER signaling is intact in these therapy-resistant tumors. ER interaction with critical coregulator proteins appears to mediate ER signaling in these therapy-resistant and ER-positive metastatic tumors. While AIs and AEs may disrupt some of these ER-coregulator interactions, their ability to target these interactions is limited in therapy-resistant cells. In this study, we described the development of a novel inhibitor that targets ER interactions with coregulators. Using in vitro and in vivo assays, we demonstrated that (*Musgrove and Sutherland, 2009*) ERX-11 blocks both ligand-dependent and ligand-independent ER signaling, (*Ma et al., 2015*) ERX-11 effectively blocks ER signaling in both therapy-sensitive and therapy-resistant cells and (*McDonnell and Norris, 2002*) disruption of ER signaling by ERX-11 has biological activity against both therapy-sensitive and therapy-resistant cells, with minimal overt signs of toxicity in vitro and in vivo.

Until recently, protein–protein interactions have been viewed as undruggable. However, the recent advent of a transformative class of compounds (peptidomimetics) allowed us to rationally design and synthesize small organic molecules that structurally emulate target protein sequences in defined conformations. We have developed a novel oligo-benzamide scaffold for mimicking helical protein segments (*Ravindranathan et al., 2013*; *Ahn and Han, 2007*). The rigid framework of the oligo-benzamide scaffold can present functional groups mimicking amino acid residues in a helical conformation (e.g. ones in i, i-3/4, and i + 7 positions). Furthermore, we have established efficient synthetic routes to make bis- and tris-benzamides as alpha-helix mimetics (*Lee and Ahn, 2011*;

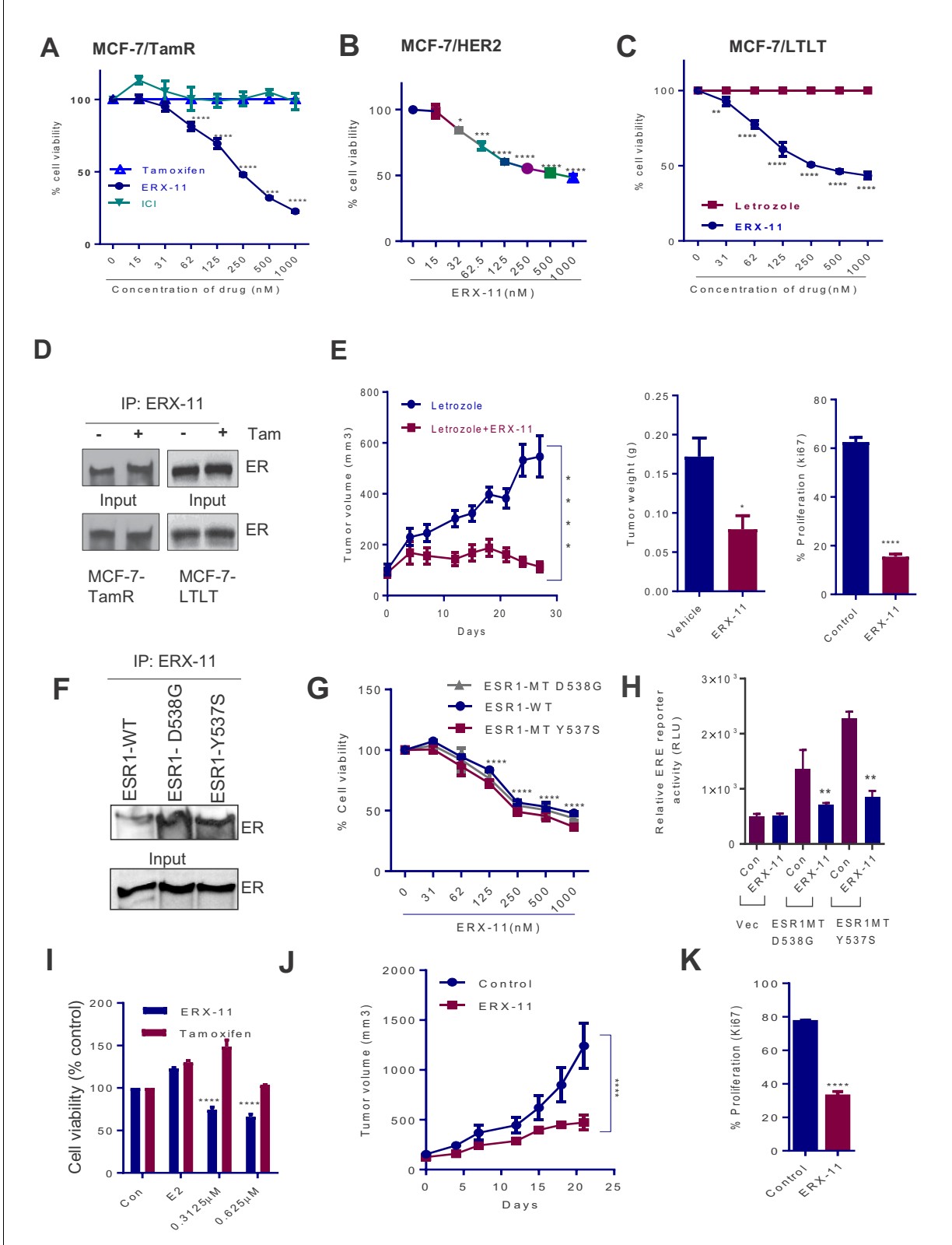

**Figure 6.** ERX-11 reduces the growth of ER positive and ER-MT endocrine-therapy-resistant tumors. Cell viability assays evaluated the effect of ERX-11 on Tamoxifen-resistant MCF-7-TamR cells (A), tamoxifen-resistant MCF-7/HER2 cells (B) and letrozole-resistant MCF-7-LTLT cells (C). ICI was used as control. Results are represented as mean ± SEM. *p<0.05; **p<0.01; ***p<0.001; ****p<0.0001. MCF-7-TamR and MCF-7-LTLT cells were cultured in 1 µM tamoxifen or 1 µM letrozole, respectively, and nuclear lysates were subject to biotin-ERX-11 pull down followed by western blotting with ER

*Figure 6 continued on next page*

*Figure 6 continued*

antibody (D). Following implantation and growth of ER-positive letrozole-resistant xenografts in nude mice (n = 8), mice were treated with control or ERX-11 (20 mg/kg/day). Tumor volume, tumor weight and Ki-67 status of control and treated tumors was shown (E). Nuclear extracts prepared from HEK-293T cells transiently transfected with WT- or MT-ER expression plasmids and analyzed for interaction between WT- and MT-ER to the biotin-ERX-11 using avidin pulldown followed by western blot analysis (F). Effect of ERX-11 on the cell viability of ZR-75 ESR1-KO cells stably expressing ESR1-WT or ESR1-Y537S mutant or ESR1-D538G mutant (G) was measured using MTT assay. ER-negative MDA-MB-231 cells were co-transfected with ERE reporter along with WT-*ESR1* and MT-*ESR1* plasmids. After 48 hr, the cells were treated with ERX-11 (500 nM) and the reporter activity was measured 24 hr later (H). Effect of ERX-11 and tamoxifen on the cell viability of ZR-75 cells stably expressing ER-Y537S mutant was measured using MTT assays (I). ZR-75 cells stably expressing ER-Y537S mutant were injected into the mammary fat pads of nude mice implanted subcutaneously with an estrogen pellet. After 2 weeks, mice with xenografts were treated with vehicle or ERX-11 (20 mg/kg/day, n = 6). Tumor growth was measured at indicated time points (J). Ki-67 expression was analyzed by IHC and quantitated (K). Data shown are the means of ± SEM. *p<0.05, **p<0.01, ***p<0.001, ****p<0.0001.

The following figure supplement is available for figure 6:

**Figure supplement 1.** ERX-11 reduces the growth of ER-positive and ER-MT endocrine-therapy-resistant tumors.

*Marimganti et al., 2009*). Based on the helix mimicking tris-benzamide scaffold, we designed ERX-11 to block the interactions between ER and a subset of its coregulator proteins containing the NR box in a helical structure.

The unbiased IPMS analyses suggest that ERX-11 interacts with ER and blocks the interactions of ER with multiple coregulators. While the broad effect of ERX-11 protein–protein interactions raises potential concerns about off-target activities, they are largely mitigated by the lack of overt signs of toxicity in cell culture models and multiple animal models tested to date. In addition, the multiplicity of targeted protein-protein interactions makes the development of resistance to the ERX-11 less likely. Thus, ERX-11 represents an exciting new mechanism to attenuate ER oncogenic functions.

Like tamoxifen, ERX-11 potently blocks the proliferation of therapy-sensitive cells. Unlike tamoxifen, ERX-11 has activity in multiple therapy-resistant models, including those driven by ER ligand-binding domain mutants. Unlike a classic SERD, ERX-11 does not cause immediate ER degradation, but appears to affect *ESR1* levels over several days, by blocking its transcription.

Our data clearly indicate that ERX-11 binds to AF2 domain of ER. However, the exact interface between ERX-11 and ER has not been established. The ability of tamoxifen to compete efficiently for ERX-11 binding to purified ER in vitro and to ER within the cell suggests a significant overlap between the tamoxifen and ER-binding site on ER. In addition, the inability of the SERDs and ICI to disrupt the interaction between ER and ERX-11 suggests that ERX-11 may also interact with a secondary tamoxifen-binding site within ER. As shown for ER beta, tamoxifen has two distinct binding sites- one in the consensus ligand-binding pocket, and another in the hydrophobic groove of the coactivator recognition surface (*Wang et al., 2006*). Published studies using LXXLL peptide probes showed ER undergo distinct conformational changes as a result of binding with different ligands and such changes expose distinct surfaces on ER facilitating interaction with various coregulators (*Paige et al., 1999*). These studies also reported that tamoxifen binding can create unique surface on ER that facilitate binding of unique LXXLL-binding peptides (*Tamrazi et al., 2002*). Further investigations of the ERX-11- ER interface with co-crystallization studies are ongoing and are likely to clarify the precise nature of the interaction.

Our studies also suggest that ERX-11 can interact with ER, even in therapy-resistant cells. While therapy resistance can be attributed to multiple mechanisms, structural changes in ER via post-translational modifications or point mutations may create new binding surfaces on ER for coregulator interactions and potentially for ERX-11 interactions. The interaction between ERX-11 and ER-MTs in therapy-resistant cells supports a model for ERX-11 interaction with ER at an alternate secondary site distinct from the site of tamoxifen binding. These findings may explain the differences between ERX-11 and tamoxifen in their activity in both therapy-sensitive and therapy-resistant breast cancer cells.

The ability of ERX-11 to block ER dimerization may be responsible for its disruption of ER DNA binding and these findings are supported by published reports that LXXLL peptides may affect ER dimerization (*Tamrazi et al., 2002*). We have noted that ERX-11 has a weak affinity for the A-isoform of PGR but not for the B-isoform. We have also found that ERX-11 does not interact with other

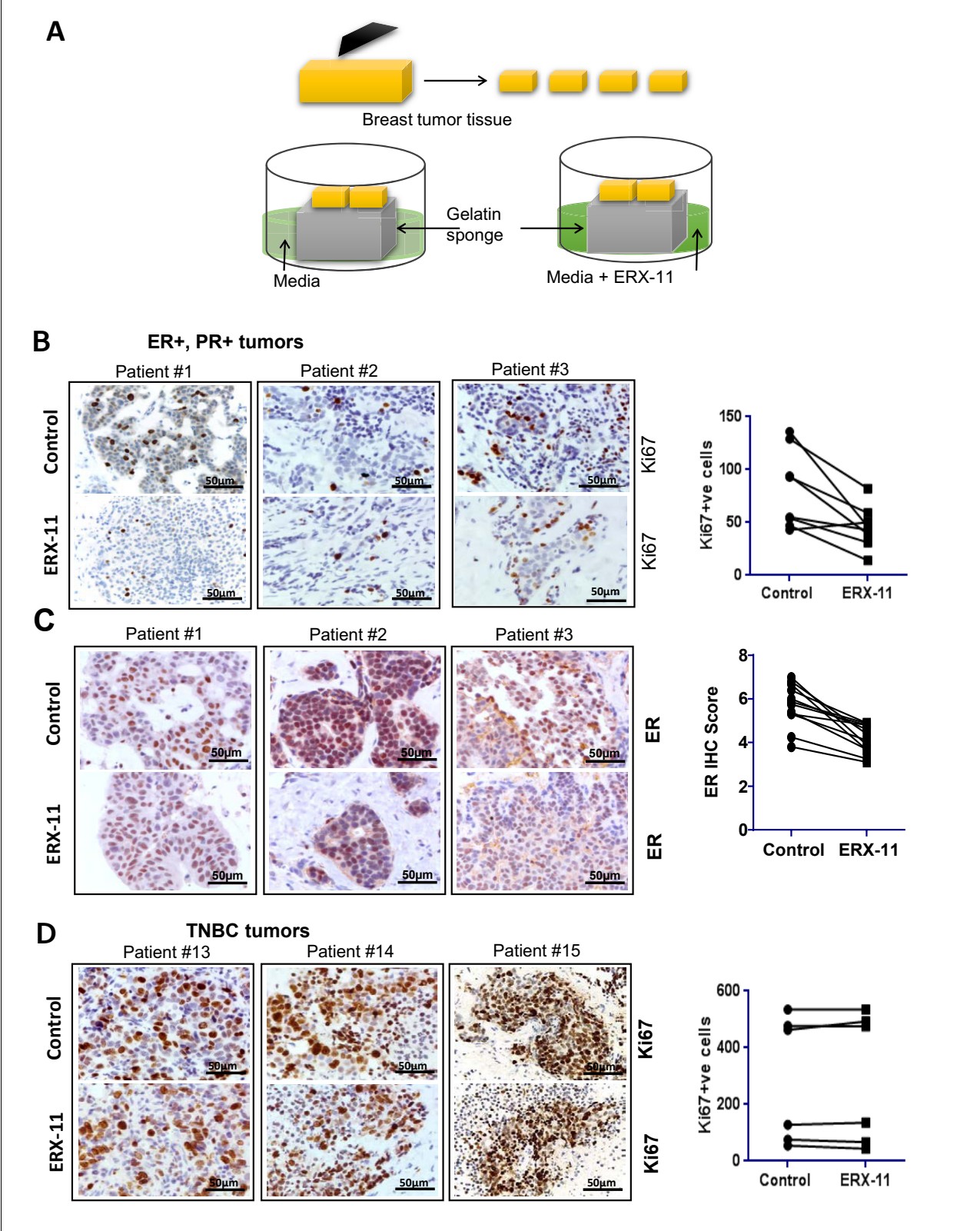

**Figure 7.** ERX-11 decreases the growth of patient-derived explants (**PDEx**): Schematic representation of ex vivo culture model is shown. (**A**) The explants were treated with ERX-11 for 48 hr. Effect of ERX-11 on Ki67 expression in ER-positive tumors with representative sections from three individual tumors and overall trend are shown (**B**). Effect of ERX-11 on ER expression in three representative ER-positive tumors is also shown (**C**). Effect of ERX-11 on Ki67 expression in three representative ER-negative tumors is shown (**D**).

*Figure 7 continued on next page*

*Figure 7 continued*

The following figure supplement is available for figure 7:

**Figure supplement 1.** Effect of ERX-11 treatment on the status of ER.

steroid receptors like GR or AR. In addition, ERX-11 failed to show activity on AR-expressing prostate cancer cells.

Disruption of multiple protein–protein interactions by ERXs may result in significant toxicity. To evaluate this, we have initially tested toxicity using a dose of 100 mg/kg in immune competent C57BL/6 mice for 14 days. We did not observe any uterotropic activity or immune effects. Further, no overt signs of toxicity was found in various E2-responsive organs including the liver, lung, heart, and kidney. Similarly, in five separate studies using a dose of 10 mg/kg in tumor-bearing mice, we did not find any overt signs of toxicity; however, ERX-11 treatment significantly limited ER expression in the tumors and reduced tumor growth. To address potential issues with immunogenicity of these tris-benzamides, we also evaluated ERX-11 treatment in syngeneic models and found no overt signs of toxicity.

While the ER coregulator protein levels are tightly regulated under normal conditions, many coregulators are over-expressed in breast cancer (*Lonard et al., 2007*), substantially contribute to ER-signaling, drive disease progression (*Singh and Kumar, 2005*) and correlate with a poor prognosis (*List et al., 2001*; *Azorsa et al., 2001*; *Tamrazi et al., 2002*). The differential coregulator milieu within breast tumors and the dependence of breast tumors on ER and coregulator-driven signaling may explain why ERX-11 has potent antiproliferative activity within the tumor but does not have any overt signs of toxicity. Since ERX-11 blocked some but not all ER coregulator interactions, ERX-11 may function as a polypharmacology agent, and its activity may depend on the concentration and repertoire of coregulators present in a tumor cell.

Importantly, using ex vivo culture of patient-derived tumor tissues, we demonstrated that ERX-11 is effective in limiting proliferation of ER-positive but not ER-negative tumors. We also discovered that ERX-11 treatment of primary tumors within their native microenvironment also reduced ER levels within the tumor. None of the primary ER-negative tumors responded to ERX-11 therapy. These explant studies represent the first evaluation of a drug effect using tissues from primary breast cancer patients and are likely to show biologically relevant outcomes.

**Table 4.** Clinicopathologic characteristics of the 12 patients, whose ER+, PR + status in breast tumors were analyzed by Ki67 and ER staining. This data is related to **Figure 7B,C**.

| Case # | Tumor | ER% | PR% | HER2 |
|---|---|---|---|---|
| 1 | IDC | 100 | 80 | Negative |
| 2 | IDC | 90 | 90 | Negative |
| 3 | Papillary | 100 | 70 | Non-amplified |
| 4 | IDC | 90 | 90 | Negative |
| 5 | IDC | 100 | 95 | Non-amplified |
| 6 | IDC | 100 | 30 | Negative |
| 7 | IDC | 100 | 10–40 | Negative |
| 8 | IDC | 80 | 50 | Negative |
| 9 | IDC | 50–60 | 80–90 | Negative |
| 10 | IDC | 100 | 100 | Non-amplified |
| 11 | IDC | 100 | 95 | Non-amplified |
| 12 | IDC | 95 | 95 | Negative |

In response to hormone binding, ER interacts with multiprotein complexes containing coregulators and transcriptional regulators to activate transcription (*McKenna et al., 1999*; *Collingwood et al., 1999*; *Tsai and O'Malley, 1994*; *Torchia et al., 1998*). Even though coregulators modulate ER functions, each coregulator protein appears to play an important but not overlapping function in vivo (*Han et al., 2006*; *Xu et al., 1998*). Accordingly, the ERX-mediated blockage of coregulator interactions with ER resulted in both inactivation and activation of unique sets of genes and pathways modulated by ER oncogenic signaling, leading to tumor suppression. The pathways/genes modulated by ERX-11 can be used to correlate with the outcome of its therapy, and they may serve as biomarkers that prognosticate response to these agents.

The biology of E2- ER signaling is complex and context dependent. Elegant studies have shown that ER, depending on the ligand and presence of unique set of coregulators can promote apoptosis (*Jordan, 2015*). In that scenario, antiestrogen and SERM are shown to inhibit apoptosis, hence, many of these drugs exhibit cytostatic response. Estrogen-induced apoptosis is shown to cause an increase in Fas receptor associated with the extrinsic pathway of apoptosis (*Lewis-Wambi and Jordan, 2009*). Similarly, a recent study found that inhibition of SRC family coregulators using small molecule inhibitor promote significant apoptosis (*Song et al., 2016*). We consistently observed activation of apoptosis by ERX-11 but failed to observe any apoptosis by tamoxifen or ICI in our assays. We believe that activation of apoptosis is not due to elimination of ER rather due to unique mechanism of action of ERX-11. Specifically, we predict that changes in the ER signaling is due to alterations in coregulator binding to ER. Accordingly, our RNAseq data showed that alterations in genes that contribute activation of apoptosis. However, further studies are needed to clearly identify the mechanisms by which ERX-11 promotes apoptosis.

Currently used drugs (AEs and AIs) are associated with an initial period of clinical response; however, most patients develop resistance with cancer progression. Recent studies suggested that selective estrogen receptor downregulators (SERDs), molecules that eliminate ER expression, may have utility for treating breast cancers that have progressed on AE and/or AIs (*McDonnell et al., 2015*). Several orally available SERDs (GDC-0810, AZD9496) were recently developed and shown to have utility in treating resistant tumors using preclinical models (*Lai et al., 2015*; *Weir et al., 2016*). However, it is important to note that each of the SERDs are only in phase I clinical trials (clinical trials. gov) and none of them are either FDA approved or have proven efficacy in patients. The only FDA-approved agent is Fulvestrant (ICI), which has known pharmacologic limitations as evidenced by its dosing regimen of intramuscular injections every 14 days. Recent studies also showed *ESR1* mutations lead to constitutive activity and reduced sensitivity to ER antagonists, and mutations such as Y537S contribute to fulvestrant resistance in vivo (*Toy et al., 2017*). Since ERXs are small, stable, orally bioavailable small molecular inhibitors, their use as an alternative therapeutic approach may decrease therapy resistance and reduce side effects, which are the current limitations of AEs or AIs.

In summary, we have developed and tested the utility of ERX-11 as a novel therapeutic agent for ER-positive, therapy-sensitive and therapy-resistant breast cancers. Since ERX-11 is orally available and is well tolerated with fewer side effects, ERX-11 can be readily extended to clinical use as a therapeutic agent, and may enhance the survival of advanced breast cancer patients.

## Materials and methods

### Cell lines

Human breast cancer cells MCF-7, ZR-75, T-47D, MDA-MB-231, BT474, BT549, BT453, SUM159, 4T1, MM468, HCC1937, HCC1187, HCC70, MDA-MB-157, MDA-MB-453, MDA-MB-468 and HEK293T cells were either obtained from American Type Culture Collection (ATCC, Manassas, VA) or a kind gift from Dr. John Minna at UT Southwestern. Ishikawa cells were purchased from Sigma (St. Louis, MO). All of these cells were passaged in the user's laboratory for fewer than 6 months after receipt or resuscitation. Validation experiments were performed using a number of luminal, basal and TNBC cells with extensive prior available molecular profiles were a kind gift of Dr. John Minna. All the model cells utilized are free of mycoplasma contamination. Additionally, STR DNA profiling of the cells was used to confirm the identity using UTHSA and UT Southwestern core facilities. MCF-7-PELP1 cells (*Vallabhaneni et al., 2011*), MCF-7-HER2 cells (*Nabha et al., 2005*), MCF-7-TamR cells (*Nabha et al., 2005*), MCF-7-LTLTca cells (*Macedo et al., 2008*) and D2A1 cells

(*Tekmal and Durgam, 1997*) were described earlier. MCF-7-LTLTca and MCF-7-TamR cells were cultured in Phenol red-free RPMI medium containing 10% dextran charcoal-treated serum supplemented with either 1 µmol/L of letrozole or 1 µmol/L of tamoxifen, respectively.

## Reagents

17-$\beta$-Estradiol (cat#E2257), and (Z)−4-Hydroxytamoxifen (cat# H7904) were purchased from Sigma (St. Louis, MO). CRISPR/Cas9 plasmids targeting *ESR1* gene were obtained from Horizon Discovery (Cambridge, MA). The anti-PELP1 (cat# 300-180A) and anti-AIB1 (cat# A300-347A) antibodies were purchased from Bethyl Laboratories (Montgomery, TX). Cleaved caspase 3 antibody was purchased from Cell signaling technology (cat# 9661S, Danvers, MA). TUNEL kit (cat# 11684795910) for apoptosis detection was purchased from Roche (Mannheim, Germany) and Ki-67 (1:150) anti-human clone MIB-1 antibody (cat#M7240) was purchased from Dako (Carpinteria, CA).

## ERX-11 synthesis

The designed tris-benzamides were constructed by iterative amide bond formation of a 3-alkoxy-4-nitrobenzoic acid with a 3-alkoxy-4-aminobenzamide (*Figure 1—figure supplement 3*). A 4-nitrobenzoic acid containing a trityl-protected hydroxyethoxy group **3** g was coupled to bis-benzamide **8** that was synthesized by following the previously reported procedure (*Ravindranathan et al., 2013*), making tris-benzamide **9** (*Figure 1—figure supplement 3C*). See the chemistry supplement for detailed synthetic procedures and characterization.

## Cell viability assays

The effects of ERX analogues on cell viability were measured using the MTT Cell Viability Assay in 96-well plates. Breast cancer cells were seeded in 96-well plates (1 $\times$ 10$^3$ cells/well) in phenol red-free RPMI medium containing 5% dextran-coated charcoal-treated fetal bovine serum (DCC-FBS) serum. After an overnight incubation, cells were treated with varying concentrations of the ERX analogues in the presence or absence of E2 (1 $\times$ 10$^{-8}$ M) for 7 days. For some experiments viability was also measured using Cell Titer-Glo Luminescent Cell Viability Assay (Promega) in 96-well, flat, clear-bottom, opaque-wall micro plates according to manufacturer's protocol. For some experiments, apoptosis was measured using Caspase-Glo 3/7 Assay (Promega) using manufacturer's protocol.

## Immunoprecipitation and western blotting

Western blotting and immunoprecipitation were performed as described previously (*Nair et al., 2010*). Biotin-ERX-11 pull-down assays were done using a previously established protocol using avidin beads (*Mann et al., 2013*). Pull-down assays using ER-AF2-GST were performed as described previously (*Nair et al., 2010*). Purified ER full-length proteins and ER LBD (AF2) protein was purchased from Thermo Fisher Scientific, Waltham, MA. The sequences of LXXLL peptides used in the competition assays. SRC1- LXXLL peptide: LTARHKILHRLLQEGSPSD; SRC2- LXXLL peptide: DSKG QTKLLQLLTTKSDQM; SRC3/AIB1- LXXLL peptide: ESKGHKKLLQLLTCSSDDR; PELP1-LXXLL-1 peptide: GLSAVSSGPRLRLLLLESVSG; PELP1-3 LXXLL peptide: SIKTRFEGLCLLSLLVGESPT

## Animal studies

All animal experiments were performed after obtaining UTHSA IACUC approval and using methods in the approved protocol. For xenograft tumor assays, 2 $\times$ 10$^6$ ZR-75, or ZR-75- ER MT-Y537S, or MCF-7- PELP1 cells were mixed with an equal volume of matrigel and implanted in the mammary fat pads of 6-week-old female athymic nude mice as described (*Cortez et al., 2012*). Based on our previous data as well as published findings, the number of mice needed were chosen to demonstrate differences in tumor incidence or treatment effect. Calculations are based on a model of unpaired data power = 0.8; p<0.05. Once tumors reached measurable size, mice were divided into control and treatment groups (n = 5–8 tumors per group). The control group received vehicle and the treatment groups received ERX-11 (10 mg/kg/day) in 30% Captisol orally. Dose were selected based on pilot MTD study of 10, 50 and 100 mg/kg of ERX-11 for 14 days using C57BL/6 mice. The mice were monitored daily for adverse toxic effects. For MCF-7-LTLT xenograft studies, MCF-7-LTLT model cells were first injected into the mammary glands of nude mice implanted with androstenedione pellets. When the tumor was established, it was dissected into small pieces and they were again

implanted subcutaneously into nude mice implanted with androstenedione pellets. Fulvestrant treatment was used as a positive control and MCF-7-LTLT xenografts were treated with 200 mg/kg/ 2 days a week/sc. For syngeneic mice studies, D2A1 cells were first injected into the mammary glands of BALB/c mice. When the tumor was established, it was dissected into small pieces and they were again implanted subcutaneously into the BALB/c mice. After three days of tumor tissue implantation, mice were randomly selected to receive control (n = 7–8) and treatment (n = 7–8) with 20 mg/kg/ day of ERX-11 orally. Tumor growth was measured with a caliper at 3–4 day intervals. At the end of each experiment, the mice were euthanized, and the tumors were removed, weighed and processed for IHC staining.

## Patient-derived explant (PDEx) studies

UTSW Patients provided written consent allowing the use of discarded surgical samples for research purposes according to an institutional board-approved protocol. De-identified patient tumors were obtained from the UTSW Tissue Repository after institutional review board approval (STU-032011–187). Excised tissue samples were processed and cultured ex vivo as previously described (*Mohammed et al., 2015*). Briefly, tissue samples were incubated on gelatin sponges for 48 hr in culture medium containing 10% FCS, followed by treatment with either vehicle or E2 (10 nM) in the absence or presence of 10 μM ERX-11 for 48 hr (see *Table 4* for clinicopathologic characteristics of these tumors). Representative tissues were fixed in 10% formalin at 4°C overnight and subsequently processed into paraffin blocks. Sections were stained with hematoxylin and eosin and examined to confirm and quantify the presence/proportion of tumor cells. Immunohistochemistry was then performed.

## Protein interaction analyses

String analyses were performed for human PELP1 using the http://string-db.org website, with the evidence view at the highest stringency for no more five interactors.

## Conformational analysis of ERX-11

A Monte Carlo conformational search was performed using the torsional sampling method (MCMM) implemented in MacroModel (version 9.0, Schrödinger, New York, NY) with automatic setup options. The calculation was done with the maximum number of steps set to 5000 using 100 steps per rotatable bond and an energy cutoff of 21 kJ/mol above the global energy minimum. The searches were done using MM3 force field (chosen for its accuracy with organic molecules) combined with the GB/SA water solvation model with standard settings and the following cut-offs: van der Waals, 8.0 Å; electrostatic, 20.0 Å; and hydrogen bond, 4.0 Å. The observed conformations were minimized by 500 iterations of Polak-Ribiere Conjugate Gradient (PRCG) algorithm (a conjugate gradient minimization scheme that uses the Polak-Ribiere first derivative method with restarts every 3N iterations) (0.05 kJ/mol).

## Molecular docking of ERX-11 to ER

AutoDock 4.2 software package, as implemented through the graphical user interface called Auto-DockTools (ADT), was used to create input PDBQT files of a receptor and a ligand. The input file of ER was prepared using the published coordinates (PDB 1L2I). Water molecules were removed from the protein structure and hydrogen was added. All other atom values were generated automatically by ADT. The docking area was assigned visually around the peptide ligand. A grid box of 24 Å x 20 Å x 24 Å was calculated around the docking area using AutoGrid. The x,y,z coordinates of the center of the grid box were set to x = −9.0, y = 14.0 and z = 26.0, respectively. The input file of ERX-11 was created from its energy-minimized conformation using ADT. Docking calculations were performed with AutoDock Vina 1.1.2. A search exhaustiveness of 16 was used and all other parameters were left as default values.

## Reporter gene assays

Briefly, cells were transiently co-transfected with 200 ng of ERE-Luc reporter with 100 ng of ER-WT, ER-MT, PELP1, SRC1, SRC2, SRC3 or control vectors using Turbofect transfection reagent (Thermo Scientific, Waltham, MA). After 24 hr, cells were treated with either vehicle or ERX-11 for an

additional 24 hr. $\beta$-galactosidase reporter (50 ng) plasmid was co-transfected and used for data normalization. Cells were lysed in Passive Lysis Buffer, and luciferase activity was measured using the luciferase assay system (Promega, Madison, WI) in a luminometer.

## RNA sequencing and RT-qPCR

RNA-seq was performed using the UTHSA core–established protocol. Briefly, ZR-75 cells were treated with either vehicle or ERX-11 for 48 hr, and total RNA was isolated using RNAesy mini kit (Qiagen) according to the manufacturer's instructions. Differential expression analysis was performed by DEseq and significant genes with at least 1.5-fold change with $p < 0.01$ were chosen for analysis. The interpretation of biological pathways using RNA-seq data was performed with IPA software using all significant and differentially expressed genes. RNA-seq data have been deposited in the GEO database under accession number GSE75664. To validate the selected genes, reverse transcription (RT) reactions were performed by using SuperScript III First Strand kit (Invitrogen,

**Table 5.** Primer sequences used for RTqPCR

| Gene name | Primer sequence |
| --- | --- |
| SRC-325F | GAGCGGCTCCAGATTGTCAA |
| SRC-410R | CTGGGGATGTAGCCTGTCTGT |
| E2F1-378F | ACGCTATGAGACCTCACTGAA |
| E2F1-626R | TCCTGGGTCAACCCCTCAAG |
| ERCC2-68F | GGAAGACAGTATCCCTGTTGGC |
| ERCC2-169R | CAATCTCTGGCACAGTTCTTGA |
| LIMK1-276F | CAAGGGACTGGTTATGGTGGC |
| LIMK1-367R | CCCCGTCACCGATAAAGGTC |
| MMP15-149F | AGGTCCATGCCGAGAACTG |
| MMP15-305R | GTCTCTTCGTCGAGCACACC |
| DUSP2-491F | GGGCTCCTGTCTACGACCA |
| DUSP2-574R | GCAGGTCTGACGAGTGACTG |
| RCOR2-118F | CACTCGCACGACAGCATGAT |
| RCOR2-285R | CATCGCAATGTACTTGTCAAGC |
| DKK1-95F | CCTTGAACTCGGTTCTCAATTCC |
| DKK1-232R | CAATGGTCTGGTACTTATTCCCG |
| PDGFB-63F | CTCGATCCGCTCCTTTGATGA |
| PDGFB-301R | CGTTGGTGCGGTCTATGAG |
| PGLYRP2-23F | TCCTACTCGGATTGCTACTGTG |
| PGLYRP2-206R | AAGTGGTAGAGGCGATTGTGG |
| ELF5-357F | TAGGGAACAAGGAATTTTTCGGG |
| ELF5-519R | GTACACTAACCTTCGGTCAACC |
| TNFSF10-46F | TGCGTGCTGATCGTGATCTTC |
| TNFSF10-126R | GCTCGTTGGTAAAGTACACGTA |
| STAT1-368F | ATCAGGCTCAGTCGGGGAATA |
| STAT1-553R | TGGTCTCGTGTTCTCTGTTCT |
| XAF1-297F | GCTCCACGAGTCCTACTGTG |
| XAF1-403R | GTTCACTGCGACAGACATCTC |
| IFI6-257F | GGTCTGCGATCCTGAATGGG |
| IFI6-401R | TCACTATCGAGATACTTGTGGGT |

Carlsbad), according to manufacturer's protocol. Real-time PCR was done using SybrGreen on an Illumina Real-Time PCR system, using primers listed in *Table 5*.

## ChIP

Chromatin immunoprecipitation (ChIP) analysis was performed using antibodies specific for the ER (Santa Cruz). Briefly, MCF-7 ($7 \times 10^6$) or T-47D ($2 \times 10^7$) cells were plated in 150 mm dishes, starved in unsupplemented phenol red-free DMEM for 24 hr and then treated for 2 hr with either ethanol or E2 after prior incubation with either DMSO or ERX-11. Relative recruitment was determined by qPCR of purified ChIP and input DNA in triplicate. The results presented are representative of two independent experiments.

## NanoBiT luciferase studies

The NanoBiT assay utilizes a structural complementation-based approach to monitor protein–protein interactions within living cells. Large BiT (LgBiT; 18 kDa) and Small BiT (SmBiT; 1 kDa) subunits of NanoLuc Luciferase were optimized for the analysis of protein interaction dynamics. When LgBiT and SmBiT subunits are separated, the Large BiT part loses the majority of luciferase activity. However, when the direct interaction between fusion proteins on LgBiT and SmBiT occurred, the interaction promotes structural complementation between LgBiT and SmBiT and results in full luciferase activity. Protein–protein interaction are then monitored in living cells following addition of the Nano-Glo Live Cell Reagent, a non-lytic detection reagent containing the cell-permeable furimazine substrate and observed luminescent signals.

To generate different NanoBiT fusion constructs, human ER and PELP1 coding sequences were amplified by PCR and separately subcloned into NB-MCS vectors (Promega). To test the protein–

**Table 6.** Analyses of the amino acids at the flanking sequences of top ER binders whose interactions are blocked by ERX-11 in MCF-7 and ZR-75 cells, as determined by unbiased IP-MS.

| Protein/GENE ID | # LXXLL motifs | LXXLL sequences | Serine at i ± 3/4 |
|---|---|---|---|
| Plectin Q15149-3 | 6 | 210 GHNLISLLEVL 220<br>213 LISLLEVLSGDS 224<br>421 YRELVLLLQWM 431<br>659 LRYLQDLLAWV 669<br>1102 YQQLLQSLEQG 1112<br>4006 TGQLLLPLSDA 4016 | yes |
| FAM83H Q6ZRV2 | 2 | 816 AAQLLDTLGRS 826<br>966 SLRLRQLLSPK 976 | yes |
| AHNAK Q09666 | 0 | | |
| CLTC: cQ00610 | 5 | 563 TAFLLDALKNN 573<br>854 RNRLKLLLPWL 864<br>1001 PNELIELLEKIV 1011<br>1021 HRNLQNLLILT 1031<br>1418 PLLLNDLLMVLS 1429 | yes |
| FAS P49327 | 10 | 76 DPQLRLLLEVT 86<br>418 HATLPRLLRAS 428<br>560 QIGLIDLLSCM 570<br>691 APPLLQELKKV 701<br>1175 QQELPRLLSAA 1185<br>1211 EDPLLSGLLDSP 1221<br>1346 GFLLLHTLLRGH 1358<br>1470 RCVLLSNLSST 1480<br>2216 QLNLRSLLVNP 2226<br>2381 NRVLEALLPLKG 2391 | yes |
| TRG P27635 | 1 | 116 VNRLLDSLEPP 126 | no |
| ACTN4 O43707 | 1 | 81 GLKLMLLLEVIS 92 | yes |
| TFG Q92734 | 0 | | |
| Coatomer subunit alpha P53621 | 1 | 83 RRCLFTLLGHLDYI 96 | no |
| Q9NVI7-2 | 1 | 99 ALSLLHTLVWA 109 | no |

protein interaction between ER and PELP1 by using the NanoBiT assay, C-LgBit-*ESR1* paired with C-SmBit-*PELP1* or C-LgBit-*PELP1* with C-SmBit-*ESR1* constructs were transiently transfected into HEK-293T cells by using Fugene HD transfection reagent (Promega). To test the ER dimerization, C-SmBit-*ESR1* were cotransfected with either N-LgBit-*ESR1* or C-LgBit-*ESR1* constructs. On the day after transfection, the medium for the HEK-293T cells was changed to phenol red-free DMEM containing 1% charcoal-stripped FBS. After a 24 hr incubation, the cells were treated with DMSO or ERX-11 (10 µM) for 2 hr and then treated cells with EtOH or E2 (10 nM) for 30 mins. After treatment, Nano-Glo live cell reagents were added into cells and luminscence was measured after 10 mins.

## Proximity ligation assays

MCF-7 cells were cultured on collagen-coated cover slips. After treated with vehicle or 10 µM ERX-11 for 30 min, cells are treated with vehicle or 10 nM E2 for 30 min. After the treatment, cells were washed with PBS and then fixed with 10% buffered formalin for 20 min and permeabilized with ice cold methanol at −20°C for 5 min. The cells were then blocked with blocking solution (provided with Duolink In Situ PLA probe, Sigma) for 30 min at 37°C followed by incubating with primary antibodies for ER from different species at 37°C for 2 hr. After washing twice with Wash Buffer A (Duolink In Situ Reagents) at room temperature, cells were then incubated with the appropriate anti-species secondary antibodies to which oligonucleotides had been conjugated (anti-rabbit PLA probe PLUS and anti-mouse PLA probe MINUS) for 1 hr at 37°C, followed by treatment with Duolink ligation-ligase solution for 30 min at 37°C. Finally, the cells were incubated with the Duolink amplification-polymerase solution for 60 min at 37°C, followed by washing and mounting on slides with Duolink Mounting Medium with DAPI. Images were taken using a Nikon Fluorescence Microscope.

## IPMS

MCF-7 or ZR-75 cells were grown in RPMI-1640 medium supplemented with 10% FBS. After cells reach 90% confluence, the medium was changed to phenol red-free RPMI1640 containing 1.5% charcoal stripped serum for 48 hr to starve the cells. After starvation, the cells were treated with vehicle or 10 µM ERX-11 for 2 hr followed by a 2 hr treatment with vehicle or 10 nM E2. Then, the cells were washed using PBS and incubated with Pierce IP Lysis Buffer (Thermo Fisher Scientific Inc.) at 4°C for 20 min. The cell lysates were centrifuged at 14,000 rpm at 4°C for 10 min, and the supernatants were used for IPMS analysis.

Dynabeads Protein G (Invitrogen) are pre-coupled with anti- ER antibody (Santa Cruz, sc-8002) and then incubated with 1.5 mg cell lysate over night at 4°C. Then the Dynabeads were washed using PBS and PBST (0.01% Tween 20) and eluted in 30 µL NuPAGE LDS sample buffer at 95°C for 15 min. The final eluents containing ER and associated proteins were separated by using SDS-PAGE. The obtained proteins were proteolytically digested and subjected to mass spectrometry (MS) analysis.

Gel band samples were digested overnight with trypsin (Promega) following reduction and alkylation with DTT and iodoacetamide (Sigma). The samples then underwent solid-phase extraction cleanup with Oasis HLB plates (Waters) and the resulting samples were analyzed by LC/MS/MS, using an Orbitrap Fusion Lumos (Thermo Electron) coupled to an Ultimate 3000 RSLC-Nano liquid chromatography systems (Dionex). Samples were injected onto a 75 µm i.d., 50 cm long Easy Spray column (Thermo) and eluted with a gradient from 0% to 28% buffer B over 60 min. Buffer A contained 2% (v/v) ACN and 0.1% formic acid in water, and buffer B contained 80% (v/v) ACN, 10% (v/v) trifluoroethanol, and 0.08% formic acid in water. The mass spectrometer operated in positive ion mode with a source voltage of 2.2 kV and capillary temperature of 275°C. MS scans were acquired at 120,000 resolution and up to 10 MS/MS spectra were obtained in the ion trap for each full spectrum acquired using higher-energy collisional dissociation (HCD) for ions with charge 2–7. Dynamic exclusion was set for 25 s.

Raw MS data files were converted to a peak list format and analyzed using the central proteomics facilities pipeline (CPFP), version 2.0.3. Peptide identification was performed using the X!Tandem and open MS search algorithm (OMSSA) search engines against the human protein database from Uniprot, with common contaminants and reversed decoy sequences appended. Fragment and precursor tolerances of 20 ppm and 0.1 Da were specified, and three missed cleavages were allowed. Carbamidomethylation of Cys was set as a fixed modification and oxidation of Met was set as a

variable modification. Label-free quantitation of proteins across samples was performed using SINQ normalized spectral index Software.

## Immunohistochemistry

For immunohistochemical analysis sections were incubated overnight with the ER (1:50) or PELP1 (1:200) or Ki67 (1:100) antibody in conjunction with proper controls. The sections were then washed three times with 0.05% Tween in PBS for 10 min, incubated with secondary antibody for 1 hr, washed three times with 0.05% Tween in PBS for 10 min, visualized by DAB substrate and counter-stained with hematoxylin QS (Vector Lab, Burlingame, CA). A proliferative index was calculated as the percentage of Ki-67-positive cells in five randomly selected microscopic fields at 40X per slide. TUNEL analysis was done using the in situ Cell Death Detection Kit (Roche, Indianapolis, IN) as per the manufacturer's protocol, and five randomly selected microscopic fields in each group were used to calculate the relative ratio of TUNEL-positive cells.

For the PDEx samples, 5-μm sections were de-waxed, rehydrated and endogenous peroxidases were blocked with hydrogen peroxide. Sections were then boiled in citrate and blocked in 5% serum for 1 hr. Primary antibodies were incubated overnight at 4°C at 1:200 for Ki67 (Vector Laboratories Burlingame, CA) and at 1:400 for ER (Santa Cruz Biotechnology, CA). Biotinylated anti-rabbit secondary antibodies (DAKO Carpentaria, CA) were incubated for 60 min at room temperature after slides were washed for 1 hr in PBS. Slides were incubated in ABC-HRP complex (Vector Laboratories Burlingame, CA) for 30 min. Bound antibodies were then visualized by incubation with 3,3' diamino-benzidine tetrahydrochloride (liquid DAB, DAKO). Slides were then rinsed in tap water and counter-stained with hematoxylin, and then cover slide were mounted. Tumor cells with nuclear staining were recorded as positive manually per tissue core, by a reviewer who was blinded to the clinical data. The numbers of Ki67-positive tumor cells were counted in three high-power fields (x40). The ER immunostaining was registered semi-quantitatively in two ways. Staining intensity (0, no staining; 1, weak staining; 2, moderate staining; and 3, intense staining) and the proportion of stained cells (0, no staining; 1, 1–25% staining; 2, 26–50%; 3, 51–75%; and 4, if more than 75% of the tumor cells were positive.

## Statistics

GraphPad Prism 6 software (GraphPad Software, SanDiego, CA) was used to analyze all data. Data represented in the bar graphs is shown as mean ± SEM. *t*-test was performed for all pairwise comparisons. A value of $p<0.05$ was considered as statistically significant. The multiple groups' statistical data were analyzed with one-way ANOVA. RNA-seq data were analyzed using IPA software.

## Acknowledgements

This study is supported by the CPRIT grant # DP150096 (RV, JA, GVR), DOD grant # W81XWH-12-1-0288 and W81XWH-13-2-0093 (GVR) funding from the Dorothy and James Cleo Thompson foundation (GVR), the John A Cole foundation (GVR), the Welch Foundation (AT-1595, JA), UTH Cancer Center grant P30 CA-54174 (RV) and NIH grant 1R01CA179120-01 (RV). We acknowledge the invaluable insights into ER structural modeling provided by Dr. Geoff Greene of the University of Chicago.

## Additional information

### Funding

| Funder | Grant reference number | Author |
| --- | --- | --- |
| Cancer Prevention and Research Institute of Texas | DP150096 | Ganesh V Raj<br>Ratna K Vadlamudi |
| Congressionally Directed Medical Research Programs | W81XWH-12-1-0288 | Ganesh V Raj |
| Congressionally Directed Medical Research Programs | W81XWH-13-2-0093 | Ganesh V Raj |
| Congressionally Directed Medical Research Programs | W81XWH-16-1-0294 | Rajeshwar Rao Tekmal |

| Welch Foundation | AT-1595 | Jung-Mo Ahn |
|---|---|---|
| National Institutes of Health | CA179120-01 | Ratna K Vadlamudi |
| National Institutes of Health | P30 CA-54174 | Ratna K Vadlamudi |
| UTHSA , School of Medicine | Briscoe Women Health Scholar | Ratna K Vadlamudi |

The funders had no role in study design, data collection and interpretation, or the decision to submit the work for publication.

### Author contributions

GVR, Conceptualization, Data curation, Software, Formal analysis, Supervision, Funding acquisition, Investigation, Writing—original draft, Project administration, Writing—review and editing; GRS, Data curation, Software, Formal analysis, Validation, Investigation, Methodology, Writing—review and editing, Co Second author; SMa, SV, Formal analysis, Validation, Investigation, Methodology, Writing—review and editing, Co Second author; T-KL, Software, Investigation, Methodology, Co Second author; RL, C-CC, W-RL, MM, SRK, BM, VKG, DS, Investigation, Methodology; XL, Validation, Investigation, Methodology; SMu, Data curation, Investigation; RRT, Resources, Supervision, Investigation; J-MA, Conceptualization, Data curation, Software, Formal analysis, Funding acquisition, Investigation, Methodology, Writing—original draft, Project administration, Writing—review and editing; RKV, Conceptualization, Resources, Data curation, Formal analysis, Supervision, Funding acquisition, Investigation, Methodology, Writing—original draft, Project administration, Writing—review and editing

### Author ORCIDs
Ratna K Vadlamudi, http://orcid.org/0000-0003-2849-4076

### Ethics

Animal experimentation: All animal experiments were performed after obtaining UTHSCSA IACUC approval and using methods in the approved protocol (15039X).

## Additional files

### Major datasets

The following dataset was generated:

| Author(s) | Year | Dataset title | Dataset URL | Database, license, and accessibility information |
|---|---|---|---|---|
| Sareddy GR, Chen Y, Vadlamudi RK | 2016 | Estrogenreceptor coregulator binding modulators (ERXs) effectively target estrogenreceptor positive human breast cancers | http://www.ncbi.nlm.nih.gov/geo/query/acc.cgi?&acc=GSE75664 | Publicly available at the NCBI Gene Expression Omnibus (accession no. GSE75664) |

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

## Appendix 1

# Supplementary information on synthetic procedures and characterization of compounds

## Chemicals and general synthetic procedures

$N^\alpha$-Fmoc-protected amino acids, aminomethylated polystyrene resin (100–200 mesh), Rink amide MBHA resin (100–200 mesh), and Rink amide linker were purchased from EMD Chemicals (Gibbstown, NJ). All amino acids used were of the L-configuration. Other chemicals and solvents were purchased from the following sources: benzotriazol-1-yloxytri (pyrrolidino)phosphonium hexafluorophosphate (PyBOP), and 6-chloro-1-hydroxybenzotriazole (Cl-HOBt), 1-[Bis(dimethylamino)methylene]−1 H-1,2,3-triazolo[4,5-b] pyridinium 3-oxide hexafluorophosphate (HATU), N,N-diisopropylethylamine (DIEA), and trifluoroacetic acid (TFA) (Oakwood Products, West Columbia. SC); acetonitrile (CAN), ethyl acetate (EtOAc), hexanes, N,N-dimethylformamide (DMF), dichloromethane (DCM), tetrahydrofuran (THF), and methanol (Fisher Scientific, Pittsburgh, PA); 4-amino-3-hydroxybenzoic acid, tin(II) chloride dehydrate, tetrakis(triphenyphosphine)palladium, triphenylsilane, and triisopropylsilane (TIS), N,N-Di-Boc-1H-pyrazole-1-carboxamide, biotin, α-cyano-4-hydroxycinnamic acid (CHCA), piperidine, anisole, acetic anhydride, ninhydrin, dimethyl sulfide (DMS), 1,2-ethanedithiol, iodomethane, 1-bromobutane, 2-bromopropane, 1-bromo-2-methylpropane, benzyl bromide, allyl bromide, 2-bromoethanol, 4-bromobutyric acid, and 4-amino-1-propanol (Aldrich, Milwaukee, WI). DMF, DCM, THF, and chloroform were dried over activated 4 Å molecular sieves before use. Other solvents were used without further purification. Thin-layer chromatography (TLC) was performed on silica gel plates (250 mm, Sorbent Technologies, Atlanta, GA) and the plates were visualized under UV at 254 nm. Standard grade silica gel (230–400 mesh, Sorbent Technologies, Atlanta, GA) was used for flash column chromatography. HPLC analyses were carried out on Agilent 1100 series HPLC system (Foster City, CA) equipped with a diode-array UV detector and a C18-bounded HPLC column (Vydac 218TP104, 4.6 mm ×250 mm, 10 μm) by using a 40 min-gradient elution from 10% to 90% acetonitrile in water (0.1% TFA) and a flow rate of 1.0 mL/min. Eluents were monitored at 220 nm. $^1$H and $^{13}$C NMR spectra were recorded on Bruker AVANCE III 500 (500 MHz) NMR spectrometer. Chemical shifts were reported in ppm from tetramethylsilane (TMS) as an internal standard. Data were expressed as follows: chemical shift (d), multiplicity (s, singlet; d, doublet; dd, doublet of doublet; t, triplet; dt, doublet of triplet, q, quartet; brs, broad singlet; m, multiplet), coupling constants (Hz). Solid-phase reactions were carried out in 12 mL polypropylene cartridges with 20 m PE frit (Applied Separations, Allentown, PA) and a labquake tube shaker (Fisher Scientific, Pittsburgh, PA) was used for mixing. MALDI-TOF MS was measured on Shimadzu AXIMA Confidence MALDI-TOF mass spectrometer (nitrogen UV laser, 50 Hz, 337 nm) by using α-cyano-4-hydroxycinnamic acid (CHCA) as matrix.

## General procedure for the alkylation of phenol

A 500 mL round-bottomed flask was charged with methyl 3-hydroxy-4-nitrobenzoate 1 (31.4 mmol), $K_2CO_3$ (62.8 mmol), DMF (300 mL), and alkyl halide (94.2 mmol). The reaction mixture was heated at 90°C for 12 hr. Alternatively, the reaction mixture was stirred at room temperature for the synthesis of 2a. The reaction mixture was cooled to room temperature, and then partitioned between EtOAc (400 mL) and brine (300 mL). The organic layer was separated and the aqueous layer was extracted with EtOAc (200 mL). The combined organic layers were washed brine (200 mL ×3), dried over $Na_2SO_4$, filtered, and concentrated under reduced pressure to give the compound two as yellow solid. The product was used in the next reaction without further purification.

## General procedure for the hydrolysis of esters

A 500 mL round-bottomed flask was charged with methyl 3-alkoxy-4-nitrobenzoate 2 (31 mmol), MeOH (200 mL), THF (200 mL) and 10% NaOH (30 mL). Alternatively, 1N LiOH (60 mL) was used for the synthesis of 3 f. The reaction mixture was stirred at room temperature for 12 hr. The volatile was removed under reduced pressure, and the resulting residue was acidified with 1 N HCl (200 mL) and then extracted with EtOAc (300 mL). The organic layer was separated and washed with 1 N HCl (100 mL) and brine (100 mL). The organic layer was then dried over $Na_2SO_4$, filtered, and concentrated under reduced pressure to give the compound three as yellow solid.

## General procedure for amide bond formation

Fmoc-Rink amide MBHA resin (0.40 g, 0.20 mmol, 0.5 mmol/g) was swollen in DMF for 2 hr and washed with DMF (3 × 2 min). The Fmoc protecting group was removed by treating with piperidine (20% in DMF, 1 × 5 min and 1 × 30 min), and washed with DMF (3 × 2 min). 3-Alkoxy-4-nitrobenzoic acid three was introduced by using a preactivated HOAt ester which was prepared by mixing a 3-alkoxy-4-nitrobenzoic acid 3 (4 equiv), HATU (four equiv), and DIEA (eight equiv) in DMF (8 mL) for 1 hr. The solution was added to the resin and shaken at room temperature for 24 hr. The resin was then filtered and washed with DMF (3 × 2 min).

## General procedure for reduction of nitro group

Resin-bound nitro group (0.20 mmol) was swollen in 50% AcOH/0.5N HCl/THF (1:1:6, 8 mL) for 20 min and treated with $SnCl_2 \cdot 2H_2O$ (five equiv). The reaction mixture was shaken at room temperature for 24 hr. The resin was then filtered, and washed with 0.5N HCl/DMF (1:6) (3 × 5 min), $H_2O$/DMF (1:6) (3 × 5 min), and DMF (3 × 5 min).

## Procedure for guanidinylation

Resin-bound N-Fmoc group 7i was swollen in DMF for 2 hr and washed with DMF (3 × 2 min). The Fmoc protecting group was removed by treating with piperidine (20% in DMF, 1 × 5 min and 1 × 30 min), and washed with DMF (3 × 2 min). Then, the resulting resin was swollen in DMF and treated with N,N'-di-Boc-1H-pyrazole-1-carboxamidine (three equiv.) and DIEA (six equiv.). The reaction mixture was shaken at room temperature for 24 hr. The resin was then filtered, and washed with DMF (3 × 2 min) giving resin-bound N,N'-di-Boc-guanidine 7 j.

## General procedure for cleavage of tris-benzamide

Resin-bound tris-benzamide seven was washed with DMF (3 × 2 min) and DCM (3 × 2 min), and dried in vacuo. The dried resin was treated with a cleavage mixture of TFA/TIS/$H_2O$ (95:2.5:2.5, 8 mL) for 90 min. The TFA solution was then filtered, and the resin was washed with TFA (2 mL). The combined TFA solution was concentrated to a volume of approximately 0.5 mL with a gentle stream of nitrogen, and the tris-benzamide was precipitated with cold diethyl ether (10 mL). The resulting precipitate was collected by centrifugation and the ether solution was decanted. Washing with cold diethyl ether was repeated and the tris-benzamide was dried in vacuo.

**Appendix 1—chemical structure 1.** Compound 3a.

The title compound was obtained as a yellow solid (5.1 g, 84% yield over two steps) using iodomethane (13.4 g, 94.2 mmol). $^1$H NMR (DMSO-$d_6$, 500 MHz): δ 13.66 (br s, 1 hr), 7.98 (d, $J$ = 8.2 Hz, 1 hr), 7.78 (d, $J$ = 1.5 Hz, 1 hr), 7.66 (dd, $J$ = 8.2, 1.5 Hz, 1 hr), 4.00 (s, 3 hr). $^{13}$C NMR (DMSO-$d_6$, 125 MHz): δ 166.3, 152.0, 142.4, 136.2, 125.5, 121.8, 15.0, 57.3.

**Appendix 1—chemical structure 2.** Compound 3b.

The title compound was obtained as a yellow solid (6.4 g, 91% yield over two steps) using 2-bromopropane (11.8 g, 94.2 mmol). $^1$H NMR (DMSO-$d_6$, 500 MHz): δ 13.62 (br s, 1 hr), 7.93 (d, $J$ = 8.3 Hz, 1 hr), 7.78 (d, $J$ = 1.5 Hz, 1 hr), 7.63 (dd, $J$ = 8.3, 1.5 Hz, 1 hr), 4.92 (septet, $J$ = 6.1 Hz, 1 hr), 1.32 (d, $J$ = 6.1 Hz, 6 hr). $^{13}$C NMR (DMSO-$d_6$, 125 MHz): δ 166.3, 150.1, 143.7, 135.9, 125.3, 121.7, 117.0, 72.9, 22.0.

**Appendix 1—chemical structure 3.** Compound 3c.

The title compound was obtained as a yellow solid (7.7 g, 93% yield over two steps) using 1-bromobutane (12.9 g, 94.2 mmol). $^1$H NMR (DMSO-$d_6$, 500 MHz): δ 13.63 (br s, 1 hr), 7.96 (d, $J$ = 8.2 Hz, 1 hr), 7.77 (d, $J$ = 1.5 Hz, 1 hr), 7.64 (dd, $J$ = 8.2, 1.5 Hz, 1 hr), 4.24 (t, $J$ = 6.4 Hz, 2 hr), 1.75–1.69 (m, 2 hr), 1.48–1.41 (m, 2 hr), 0.94 (t, $J$ = 7.3 Hz, 3 hr). $^{13}$C NMR (DMSO-$d_6$, 125 MHz): δ 166.3, 151.3, 142.6, 136.1, 125.4, 121.7, 115.7, 69.6, 30.8, 19.0, 14.0.

**Appendix 1—chemical structure 4.** Compound 3d.

The title compound was prepared according to the published procedure.

**Appendix 1—chemical structure 5.** Compound 3e.

The title compound was obtained as a yellow solid (7.5 g, 89% yield over two steps) using benzyl bromide (8.2 g, 47.1 mmol). [1]H NMR (DMSO-$d_6$, 500 MHz): $\delta$ 13.66 (br s, 1 hr), 8.01 (d, $J$ = 8.2 Hz, 1 hr), 7.89 (d, $J$ = 1.5 Hz, 1 hr), 7.68 (dd, $J$ = 8.2, 1.5 Hz, 1 hr), 7.48–7.35 (m, 5 hr), 5.41 (s, 2 hr). [13]C NMR (DMSO-$d_6$, 125 MHz): $\delta$ 166.2, 151.0, 142.8, 136.3, 136.1, 129.0, 128.6, 127.9, 125.5, 122.1, 116.3, 71.1.

**Appendix 1—chemical structure 6.** Compound 3f.

The title compound was obtained as a yellow solid (8.7 g, 85% yield over two steps) using tert-butyl 4-bromobutyrate [5] (10.5 g, 47.1 mmol). [1]H NMR (DMSO-$d_6$, 500 MHz): $\delta$ 13.65 (br s, 1 hr), 7.98 (d, $J$ = 8.2 Hz, 1 hr), 7.76 (d, $J$ = 1.5 Hz, 1 hr), 7.65 (dd, $J$ = 8.2, 1.5 Hz, 1 hr), 4.25 (t, $J$ = 6.3 Hz, 2 hr), 2.38 (t, $J$ = 7.5 Hz, 2 hr), 1.99–1.92 (m, 2 hr), 1.41 (s, 9 hr). [13]C NMR (DMSO-$d_6$, 125 MHz): $\delta$ 172.2, 166.2, 151.3, 142.6, 136.2, 125.5, 121.9, 115.8, 80.3, 69.0, 31.4, 28.2, 24.5.

**Appendix 1—chemical structure 7.** Compound 3g.

A 500 mL round-bottomed flask was charged with methyl 3-hydroxy-4-nitrobenzoate 1 (5.0 g, 25.4 mmol), $K_2CO_3$ (7.0 g, 50.6 mmol), TBAB (0.42 g, 1.3 mmol), DMF (200 mL), and 2-bromoethyl trityl ether [6] (14.0 g, 38.1 mmol). The reaction mixture was heated at 90°C for 12 hr and then cooled to room temperature. The solution was then poured into brine (200 mL) and extracted with EtOAc (200 mL ×2). The combined organic extracts were washed with brine (100 mL ×3), dried over $Na_2SO_4$, filtered, and concentrated under reduced pressure to give compound 2 g as a yellow solid. The product was used in the next reaction without further purification.

A 1 L round-bottomed flask was charged with compound 2 g, MeOH (200 mL), THF (600 mL) and 10% NaOH (40 mL). The reaction mixture was stirred at room temperature for 24 hr and then concentrated under reduced pressure. The reaction mixture was acidified with 5% citric acid (200 mL) and extracted with extracted with EtOAc (300 mL ×2). The combined organic extracts were washed with brine (200 mL ×3), dried over $Na_2SO_4$, filtered, and concentrated under reduced pressure to give the crude product. Recrystallization from EtOAc/hexanes (1:1) gave compound 3 g as a light yellow solid (11.2 g, 94% yield over two steps). $R_f$ = 0.33 (EtOAc). [1]H NMR (DMSO-$d_6$, 500 MHz): δ 13.65 (br s, 1 hr), 8.01 (d, $J$ = 8.2 Hz, 1 hr), 7.89 (d, $J$ = 1.5 Hz, 1 hr), 7.69 (dd, $J$ = 8.2, 1.5 Hz, 1 hr), 7.40–7.38 (m, 6 hr), 7.35–7.32 (m, 6 hr), 7.29–7.26 (m, 3 hr), 4.49–4.47 (m, 2 hr), 3.31–3.29 (m, 2 hr). [13]C NMR (DMSO-$d_6$, 125 MHz): δ 166.3, 151.22, 143.9, 142.9, 136.0, 128.7, 128.4, 127.5, 125.3, 122.0, 116.1, 86.5, 69.3, 62.7.

**Appendix 1—chemical structure 8.** Compound 3i.

A 250 mL round-bottomed flask was charged with methyl 3-hydroxy-4-nitrobenzoate 1 (2.7 g, 13.6 mmol), $K_2CO_3$ (3.8 g, 27.2 mmol), TBAB (0.23 g, 0.68 mmol), DMF (100 mL), and 3-(tritylamino)propyl methanesulfonate (7.0 g, 17.7 mmol). The reaction mixture was heated at 90°C for 12 hr and then cooled to room temperature. The solution was then poured into brine (200 mL) and extracted with EtOAc (200 mL ×2). The combined organic extracts were washed with brine (100 mL ×3), dried over $Na_2SO_4$, filtered, and concentrated under reduced pressure to give compound 2 hr as a yellow solid. The product was used in the next reaction without further purification.

A 500 mL round-bottomed flask was charged with compound 2 hr, MeOH (100 mL), THF (100 mL) and 1N LiOH (28 mL). The reaction mixture was stirred at room temperature for 24 hr and then concentrated under reduced pressure. The reaction mixture was acidified with

5% citric acid (50 mL) and extracted with extracted with EtOAc (100 mL ×2). The combined organic extracts were washed with brine (100 mL ×3), dried over $Na_2SO_4$, filtered, and concentrated under reduced pressure to give the crude product. Purification by flash chromatography using EtOAc gave compound 3 hr as a white solid (3.0 g, 46% yield over two steps).

The compound 3 hr (3.0 g, 6.2 mmol) was treated with a cleavage mixture of TFA/TIS/DCM (10:5:85, 50 mL) for 30 min. The excess of TFA were removed under reduced pressure and precipitation with diethyl ether gave the crude product, which was directly used for the next reaction without further purification.

A 250 mL round-bottomed flask was charged with the crude product, $NaHCO_3$ (2.7 g, 31.1 mmol), THF (100 mL), and $H_2O$ (50 mL). Then, Fmoc-OSu (3.2 g, 9.5 mmol) was added to the solution and the reaction mixture was stirred at room temperature for 12 hr. The volatile was removed under reduced pressure and the aqueous phase was washed with diethyl ether. The aqueous phase was acidified with 1 N HCl (50 mL) and then extracted with EtOAc (100 mL ×2). The combined organic layers were washed with 1 N HCl (50 mL) and brine (50 mL). The organic layer was then dried over $Na_2SO_4$, filtered, and concentrated under reduced pressure to give the compound 3i as a yellow solid (2.5 g, 87% yield over two steps). [1]H NMR (DMSO-$d_6$, 500 MHz): δ 13.67 (br s, 1 hr), 7.99 (d, J = 8.3 Hz, 1 hr), 7.89 (d, J = 7.6 Hz, 2 hr), 7.77 (d, J = 1.4 Hz, 1 hr), 7.69 (d, J = 7.6 Hz, 2 hr), 7.66 (dd, J = 8.3, 1.5 Hz, 1 hr), 7.41 (t, J = 7.6 Hz, 2 hr), 7.37 (t, J = 5.6 Hz, 1 hr), 7.33 (dt, J = 0.9, 7.5 Hz, 2 hr), 4.33 (d, J = 6.7 Hz, 2 hr), 4.24–4.20 (m, 3 hr), 3.16 (q, J = 6.4 Hz, 2 hr), 1.88 (quin, J = 6.4 Hz, 2 hr). [13]C NMR (DMSO-$d_6$, 125 MHz): δ 166.29, 156.6, 151.5, 144.4, 142.5, 141.2, 136.2, 128.1, 127.5, 125.60, 125.55, 121.7, 120.6, 115.8, 67.5, 65.6, 47.3, 37.3, 29.2.

**Appendix 1—chemical structure 9.** ERX-7.

The title compound was obtained as a yellow solid (64 mg, 53% overall yield) using compound 3a and Fmoc-Rink amide MBHA resin (0.70 mmol/g, 0.30 g, 0.21 mmol). [1]H NMR (DMSO-$d_6$, 500 MHz): δ 9.88 (br s, 1 hr), 9.46 (br s, 1 hr), 8.06 (d, J = 8.7 Hz, 1 hr), 7.99 (br s, 1 hr), 7.99 (d, J = 8.3 Hz, 1 hr), 7.95 (d, J = 8.2 Hz, 1 hr), 7.85 (d, J = 1.5 Hz, 1 hr), 7.66 (dd, J = 8.2, 1.5 Hz, 1 hr), 7.65 (d, J = 1.5 Hz, 1 hr), 7.62 (dd, J = 8.2, 1.8 Hz, 1 hr), 7.59 (d, J = 1.8 Hz, 1 hr), 7.55 (dd, J = 8.2, 1.8 Hz, 1 hr), 7.36 (br s, 1 hr), 4.04 (s, 2.2 hr), 4.03 (s, 0.8 hr), 3.93 (d, J = 6.7 Hz, 2 hr), 3.92 (d, J = 6.4 Hz, 2 hr), 2.17–2.07 (m, 2 hr), 1.05 (d, J = 6.7 Hz, 6 hr), 1.02 (d, J = 6.7 Hz, 6 hr). [13]C NMR (DMSO-$d_6$, 125 MHz): δ 167.8, 164.7, 164.1, 152.1, 151.31, 150.30, 141.5, 139.9, 132.4, 131.5, 130.29, 130.27, 125.6, 124.5, 122.8, 120.4, 120.3, 120.1, 113.8, 111.7, 111.6, 75.1, 75.0, 57.3, 28.34, 28.26, 19.6, 19.5. HRMS-ESI (m/z): [M-H]- calcd for C30H34N4O8: 577.2304, found: 577.2318.

**Appendix 1—chemical structure 10.** ERX-8.

The title compound was obtained as a yellow solid (65 mg, 49% overall yield) using compound 3b and Fmoc-Rink amide MBHA resin (0.70 mmol/g, 0.30 g, 0.21 mmol). [1]H NMR (DMSO-$d_6$, 500 MHz): $\delta$ 9.84 (br s, 1 hr), 9.46 (br s, 1 hr), 8.00 (d, $J$ = 8.2 Hz, 1 hr), 7.99 (br s, 1 hr), 7.99 (d, $J$ = 8.2 Hz, 1 hr), 7.96 (d, $J$ = 8.2 Hz, 1 hr), 7.85 (d, $J$ = 1.5 Hz, 1 hr), 7.65 (d, $J$ = 1.8 Hz, 1 hr), 7.63 (dd, $J$ = 8.2, 1.5 Hz, 1 hr), 7.61 (dd, $J$ = 8.2, 1.8 Hz, 1 hr), 7.59 (d, $J$ = 1.8 Hz, 1 hr), 7.55 (dd, $J$ = 8.2, 1.8 Hz, 1 hr), 7.36 (br s, 1 hr), 4.96 (septet, $J$ = 6.1 Hz, 1 hr), 3.93 (d, $J$ = 6.4 Hz, 2 hr), 3.92 (d, $J$ = 6.4 Hz, 2 hr), 2.17–2.08 (m, 2 hr), 1.35 (d, $J$ = 6.1 Hz, 6 hr), 1.05 (d, $J$ = 6.7 Hz, 6 hr), 1.02 (d, $J$ = 6.7 Hz, 6 hr). [13]C NMR (DMSO-$d_6$, 125 MHz): $\delta$ 167.8, 164.7, 164.1, 151.3, 150.30, 150.25, 142.9, 139.6, 132.4, 131.5, 130.3, 125.5, 124.5, 122.7, 120.4, 120.2, 120.1, 115.6, 111.7, 111.6, 75.1, 75.0, 73.0, 28.34, 28.28, 22.1, 19.6, 19.5. HRMS-ESI (m/z): [M-H]- calcd for C32H38N4O8: 605.2617, found: 605.2640.

**Appendix 1—chemical structure 11.** ERX-9.

The title compound was obtained as a yellow solid (83 mg, 48% overall yield) using compound 3 c and Fmoc-Rink amide MBHA resin (0.70 mmol/g, 0.40 g, 0.28 mmol). [1]H NMR (DMSO-$d_6$, 500 MHz): $\delta$ 9.85 (br s, 1 hr), 9.46 (br s, 1 hr), 8.04 (d, $J$ = 8.2 Hz, 1 hr), 7.99 (br s, 1 hr), 7.99 (d, $J$ = 8.2 Hz, 1 hr), 7.96 (d, $J$ = 8.2 Hz, 1 hr), 7.84 (d, $J$ = 1.5 Hz, 1 hr), 7.65 (d, $J$ = 1.8 Hz, 1 hr), 7.64 (dd, $J$ = 8.2, 1.5 Hz, 1 hr), 7.61 (dd, $J$ = 8.2, 1.8 Hz, 1 hr), 7.59 (d, $J$ = 1.8 Hz, 1 hr), 7.55 (dd, $J$ = 8.2, 1.8 Hz, 1 hr), 7.36 (br s, 1 hr), 4.28 (t, $J$ = 6.3 Hz, 2 hr), 3.93 (d, $J$ = 6.7 Hz, 2 hr), 3.92 (d, $J$ = 6.4 Hz, 2 hr), 2.16–2.10 (m, 2 hr), 1.79–1.73 (m, 2 hr), 1.51–1.43 (m, 2 hr), 1.05 (d, $J$ = 6.7 Hz, 6 hr), 1.02 (d, $J$ = 6.7 Hz, 6 hr), 0.96 (d, $J$ = 7.3 Hz, 3 hr). [13]C NMR (DMSO-$d_6$, 125 MHz): $\delta$ 167.8, 164.7, 164.1, 151.5, 151.3, 150.3, 141.8, 139.8,

132.4, 131.5, 130.30, 130.29, 125.5, 124.4, 122.7, 120.4, 120.3, 120.0, 114.4, 111.7, 111.6, 75.1, 75.0, 69.6, 30.8, 28.34, 28.28, 19.61, 19.55, 19.0, 14.0. HRMS-ESI (m/z): [M-H]- calcd for $C_{33}H_{40}N_4O_8$: 619.2773, found: 619.2791.

**Appendix 1—chemical structure 12.** ERX-5.

The title compound was obtained as a yellow solid (83 mg, 53% overall yield) using compound 3d and Fmoc-Rink amide MBHA resin (0.50 mmol/g, 0.50 g, 0.25 mmol). [1]H NMR (DMSO-$d_6$, 500 MHz): δ 9.85 (br s, 1 hr), 9.46 (br s, 1 hr), 8.05 (d, $J$ = 8.2 Hz, 1 hr), 7.99 (d, $J$ = 8.2 Hz, 1 hr), 7.99 (br s, 1 hr), 7.97 (d, $J$ = 8.2 Hz, 1 hr), 7.83 (d, $J$ = 1.5 Hz, 1 hr), 7.65 (d, $J$ = 1.8 Hz, 1 hr), 7.64 (dd, $J$ = 8.2, 1.5 Hz, 1 hr), 7.62 (dd, $J$ = 8.2, 1.8 Hz, 1 hr), 7.59 (d, $J$ = 1.8 Hz, 1 hr), 7.55 (dd, $J$ = 8.2, 1.8 Hz, 1 hr), 7.36 (br s, 1 hr), 4.05 (d, $J$ = 6.4 Hz, 2 hr), 3.94 (d, $J$ = 6.4 Hz, 2 hr), 3.92 (d, $J$ = 6.4 Hz, 2 hr), 2.17–2.07 (m, 3 hr), 1.05 (d, $J$ = 6.7 Hz, 6 hr), 1.03 (d, $J$ = 6.7 Hz, 6 hr), 1.02 (d, $J$ = 6.7 Hz, 6 hr). [13]C NMR (DMSO-$d_6$, 125 MHz): δ 167.8, 164.7, 164.1, 151.5, 151.2, 150.3, 141.7, 139.9, 132.4, 131.5, 130.29, 130.28, 125.6, 124.4, 122.7, 120.4, 120.3, 120.1, 114.3, 111.7, 111.5, 75.8, 75.1, 75.0, 28.34, 28.29, 28.1, 19.61, 19.58, 19.2. HRMS-ESI (m/z): [M-H]- calcd for $C_{33}H_{40}N_4O_8$: 619.2773, found: 619.2794.

**Appendix 1—chemical structure 13.** ERX-13.

The title compound was obtained as a yellow solid (56 mg, 57% overall yield) using compound 3e and Fmoc-Rink amide MBHA resin (0.50 mmol/g, 0.30 g, 0.15 mmol). [1]H NMR (DMSO-$d_6$, 500 MHz): δ 9.88 (br s, 1 hr), 9.46 (br s, 1 hr), 8.09 (d, $J$ = 8.3 Hz, 1 hr), 7.99 (d,

$J$ = 8.2 Hz, 1 hr), 7.99 (br s, 1 hr), 7.97 (d, $J$ = 1.8 Hz, 1 hr), 7.94 (d, $J$ = 8.2 Hz, 1 hr), 7.69 (dd, $J$ = 8.2, 1.5 Hz, 1 hr), 7.65 (d, $J$ = 1.8 Hz, 1 hr), 7.62 (dd, $J$ = 8.2, 1.8 Hz, 1 hr), 7.59 (d, $J$ = 1.8 Hz, 1 hr), 7.56 (dd, $J$ = 8.2, 1.8 Hz, 1 hr), 7.50–7.38 (m, 5 hr), 7.36 (br s, 1 hr), 5.43 (s, 2 hr), 3.93 (d, $J$ = 6.4 Hz, 2 hr), 3.92 (d, $J$ = 6.4 Hz, 2 hr), 2.17–2.09 (m, 3 hr), 1.05 (d, $J$ = 6.7 Hz, 6 hr), 1.01 (d, $J$ = 6.7 Hz, 6 hr). $^{13}$C NMR (DMSO-$d_6$, 125 MHz): δ 167.8, 164.7, 164.0, 151.3, 151.2, 150.3, 141.9, 139.8, 136.2, 132.5, 131.5, 130.30, 130.27, 129.1, 128.7, 127.9, 125.7, 124.6, 122.8, 120.4, 120.34, 120.26, 115.1, 111.7, 111.6, 75.1, 75.0, 71.3, 28.34, 28.26, 19.61, 19.56. HRMS-ESI (m/z): [M-H]- calcd for C36H38N4O8: 653.2617, found: 653.2632.

**Appendix 1—chemical structure 14.** ERX-10.

The title compound was obtained as a yellow solid (90 mg, 49% overall yield) using compound 3 f and Fmoc-Rink amide MBHA resin (0.70 mmol/g, 0.40 g, 0.28 mmol). $^1$H NMR (DMSO-$d_6$, 500 MHz): δ 12.20 (br s, 1 hr), 9.87 (br s, 1 hr), 9.46 (br s, 1 hr), 8.06 (d, $J$ = 8.2 Hz, 1 hr), 7.99 (d, $J$ = 8.2 Hz, 1 hr), 7.99 (br s, 1 hr), 7.94 (d, $J$ = 8.2 Hz, 1 hr), 7.84 (d, $J$ = 1.5 Hz, 1 hr), 7.65 (d, $J$ = 1.8 Hz, 1 hr), 7.65 (dd, $J$ = 8.2, 1.5 Hz, 1 hr), 7.61 (dd, $J$ = 8.2, 1.8 Hz, 1 hr), 7.59 (d, $J$ = 1.8 Hz, 1 hr), 7.55 (dd, $J$ = 8.2, 1.8 Hz, 1 hr), 7.36 (br s, 1 hr), 4.30 (t, $J$ = 6.3 Hz, 2 hr), 3.93 (d, $J$ = 6.7 Hz, 2 hr), 3.92 (d, $J$ = 6.7 Hz, 2 hr), 2.43 (t, $J$ = 7.3 Hz, 2 hr), 2.16–2.09 (m, 2 hr), 2.03–1.98 (m, 2 hr), 1.05 (d, $J$ = 6.7 Hz, 6 hr), 1.02 (d, $J$ = 6.7 Hz, 6 hr). $^{13}$C NMR (DMSO-$d_6$, 125 MHz): δ 174.4, 167.8, 164.7, 164.1, 151.4, 151.3, 150.3, 141.7, 139.9, 132.4, 131.5, 130.30, 130.28, 125.7, 124.6, 122.7, 120.4, 120.24, 120.16, 114.4, 111.7, 111.6, 75.1, 75.0, 69.0, 30.2, 28.34, 28.28, 24.4, 19.6, 19.5. HRMS-ESI (m/z): [M-H]- calcd for C33H38N4O10: 649.2515, found: 649.2535.

**Appendix 1—chemical structure 15.** ERX-12.

The title compound was prepared using compound 3i and Fmoc-Rink amide MBHA resin (0.50 mmol/g, 0.40 g, 0.20 mmol). Following the general procedure, the resin-bound N-Fmoc group 7i was converted to the resin-bound N, N'-di-Boc-guanindine 7 j by the removal of Fmoc group with piperidine followed by reaction with N,N'-di-Boc-1H-pyrazole-1-carboxamidine. After cleavage from the resin and HPLC purification, ERX-12 was obtained as a yellow solid (31 mg, 20% overall yield). $^1$H NMR (DMSO-$d_6$, 500 MHz): $\delta$ 9.88 (br s, 1 hr), 9.46 (br s, 1 hr), 8.09 (d, $J$ = 8.2 Hz, 1 hr), 7.99 (br s, 1 hr), 7.98 (d, $J$ = 8.2 Hz, 1 hr), 7.92 (d, $J$ = 8.2 Hz, 1 hr), 7.85 (d, $J$ = 1.5 Hz, 1 hr), 7.69 (dd, $J$ = 8.2, 1.5 Hz, 1 hr), 7.66 (d, $J$ = 1.8 Hz, 1 hr), 7.64 (br s, 1 hr), 7.62 (dd, $J$ = 8.2, 1.8 Hz, 1 hr), 7.60 (d, $J$ = 1.8 Hz, 1 hr), 7.56 (dd, $J$ = 8.2, 1.8 Hz, 1 hr), 7.36 (br s, 1 hr), 4.32 (t, $J$ = 6.1 Hz, 2 hr), 3.93 (d, $J$ = 6.7 Hz, 2 hr), 3.92 (d, $J$ = 6.6 Hz, 2 hr), 3.30 (q, $J$ = 6.5 Hz, 2 hr), 2.16–2.08 (m, 2 hr), 2.03–1.98 (m, 2 hr), 1.05 (d, $J$ = 6.7 Hz, 6 hr), 1.02 (d, $J$ = 6.7 Hz, 6 hr). $^{13}$C NMR (DMSO-$d_6$, 125 MHz): $\delta$ 167.8, 164.7, 164.0, 157.3, 151.5, 151.40, 150.35, 141.6, 140.0, 132.6, 131.6, 130.3, 130.2, 125.8, 124.7, 122.8, 120.4, 120.3, 114.6, 111.69, 111.65, 75.1, 75.0, 67.1, 38.0, 28.4, 28.34, 28.26, 19.6, 19.5. HRMS-ESI (m/z): [M + H]+calcd for C33H41N7O8: 664.3089, found: 664.3094.

**Appendix 1—chemical structure 16.** Compound 8.

This compound was prepared according to the published procedure (*Ravindranathan et al., 2013*).

**Appendix 1—chemical structure 17.** Compound 9.

To a solution of compound 3 g (2.82 g, 6.01 mmol) and DIEA (2.0 mL, 11.3 mmol) in DMF (30 mL) was added HATU (2.94 g, 7.50 mmol) and the mixture was stirred for 1 hr. Compound 8 (1.5 g, 3.75 mmol) was added and the resulting mixture was stirred at room temperature for 24 hr. The solution was then poured into 5% citric acid (50 mL) and extracted with EtOAc (50 mL x 2). The combined organic extracts were washed with

saturated NaHCO$_3$ (50 mL x 2), dried over Na$_2$SO$_4$, filtered, and concentrated under reduced pressure. Recrystallization from EtOAc gave compound nine as a light yellow solid (2.3 g, 72%). $^1$H NMR (DMSO-$d_6$, 500 MHz): δ 9.87 (br s, 1 hr), 9.41 (br s, 1 hr), 8.07 (d, $J$ = 8.2 Hz, 1 hr), 8.02 (d, $J$ = 8.2 Hz, 1 hr), 7.98 (d, $J$ = 8.2 Hz, 1 hr), 7.97 (br s, 2 hr), 7.71 (dd, $J$ = 8.2, 1.5 Hz, 1 hr), 7.61 (br s, 1 hr), 7.60–7.58 (m, 2 hr), 7.55 (dd, $J$ = 8.2, 1.8 Hz, 1 hr), 7.42–7.40 (m, 6 hr), 7.36–7.33 (m, 7 hr), 7.29–7.26 (m, 3 hr), 4.52–4.50 (m, 2 hr), 3.92 (d, $J$ = 6.4 Hz, 2 hr), 3.91 (d, $J$ = 6.4 Hz, 2 hr), 3.33–3.32 (m, 2 hr, overlap with H$_2$O peak), 2.18–2.05 (m, 2 hr), 1.05 (d, $J$ = 6.7 Hz, 6 hr), 0.99 (d, $J$ = 6.7 Hz, 6 hr). $^{13}$C NMR (DMSO-$d_6$, 125 MHz): δ 167.8, 164.7, 164.1, 151.3, 150.2, 144.0, 141.9, 140.1, 132.1, 131.4, 130.4, 128.7, 128.4, 127.6, 125.5, 124.4, 122.6, 120.4, 120.29, 120.26, 114.7, 111.7, 86.5, 75.1, 75.0, 69.3, 62.5, 28.34, 28.26, 19.61, 19.55.

**Appendix 1—chemical structure 18.** ERX-11.

A solution of compound 9 (1.0 g, 1.18 mmol) in TFA (20 mL) was stirred at room temperature for 30 min. The excess of TFA were removed under reduced pressure and precipitation with diethyl ether gave ERX-11 as a white solid (0.61 g, 85%). $^1$H NMR (DMSO-$d_6$, 500 MHz): δ 9.86 (br s, 1 hr), 9.46 (br s, 1 hr), 8.03 (d, $J$ = 8.2 Hz, 1 hr), 7.99 (br s, 1 hr), 7.99 (d, $J$ = 8.2 Hz, 1 hr), 7.94 (d, $J$ = 8.2 Hz, 1 hr), 7.87 (d, $J$ = 1.9 Hz, 1 hr), 7.65 (d, $J$ = 1.9 Hz, 1 hr), 7.64 (dd, $J$ = 8.5, 1.6 Hz, 1 hr), 7.61 (dd, $J$ = 8.2, 1.9 Hz, 1 hr), 7.59 (d, $J$ = 1.9 Hz, 1 hr), 7.55 (dd, $J$ = 8.2, 1.9 Hz, 1 hr), 7.36 (br s, 1 hr), 4.30 (t, $J$ = 5.0 Hz, 2 hr), 3.93 (d, $J$ = 6.6 Hz, 2 hr), 3.92 (d, $J$ = 6.6 Hz, 2 hr), 3.77 (t, $J$ = 5.0 Hz, 2 hr), 2.17–2.08 (m, 2 hr), 1.05 (d, $J$ = 6.6 Hz, 6 hr), 1.02 (d, $J$ = 6.6 Hz, 6 hr). $^{13}$C NMR (DMSO-$d_6$, 125 MHz): δ 167.8, 164.7, 164.1, 151.6, 151.34, 150.30, 141.9, 139.7, 132.4, 131.5, 130.30, 130.29, 125.6, 124.6, 122.7, 120.4, 120.3, 120.1, 114.73, 111.66, 111.58, 75.1, 75.0, 72.0, 59.7, 28.34, 28.27, 19.61, 19.55. HRMS-ESI (m/z): [M-H]- calcd for C31 H36 N4 O9: 607.2410, found: 607.2435.

**Appendix 1—chemical structure 19.** Compound 10.

This compound was prepared according to the published procedure (*Ravindranathan et al., 2013*).

**Appendix 1—chemical structure 20.** Compound 11.

To a solution of compound **3** g (1.5 g, 3.2 mmol) and DIEA (1.1 mL, 6.1 mmol) in DMF (30 mL) was added HATU (1.5 g, 4.1 mmol) and the mixture was stirred at room temperature for 1 hr. Compound **10** (0.90 g, 2.0 mmol) was added and the resulting mixture was stirred at room temperature for 24 hr. The solution was then poured into 5% citric acid (50 mL) and extracted with EtOAc (50 mL x 2). The combined organic extracts were washed with saturated $NaHCO_3$ (50 mL x 2), dried over $Na_2SO_4$, filtered, and concentrated under reduced pressure. Recrystallization from EtOAc/hexanes (1:1) gave compound **11** as a light yellow solid (1.5 g, 83%). $R_f$ = 0.33 (hexanes/EtOAc 4:1). [1]H NMR (DMSO-$d_6$, 500 MHz): $\delta$ 9.80 (br s, 1 hr), 9.53 (br s, 1 hr), 8.18 (d, $J$ = 8.2 Hz, 1 hr), 8.07 (d, $J$ = 8.2 Hz, 1 hr), 7.99 (d, $J$ = 8.2 Hz, 1 hr), 7.97 (br s, 1 hr), 7.71 (dd, $J$ = 8.2, 1.5 Hz, 1 hr), 7.69 (dd, $J$ = 8.2, 1.8 Hz, 1 hr), 7.62–7.59 (m, 3 hr), 7.42–7.40 (m, 6 hr), 7.36–7.32 (m, 6 hr), 7.29–7.26 (m, 3 hr), 6.12–6.04 (m, 1 hr), 5.43 (ddd, $J$ = 17.2, 3.1, 1.7 Hz, 1 hr), 5.31 (ddd, $J$ = 10.3, 2.9, 1.4 Hz, 1 hr), 4.83 (dt, $J$ = 5.4, 1.4 Hz, 2 hr), 4.52–4.50 (m, 2 hr), 3.95 (d, $J$ = 6.4 Hz, 2 hr), 3.91 (d, $J$ = 6.4 Hz, 2 hr), 3.33–3.32 (m, 2 hr, overlap with $H_2O$ peak), 2.18–2.07 (m, 2 hr), 1.06 (d, $J$ = 6.7 Hz, 6 hr), 0.99 (d, $J$ = 6.7 Hz, 6 hr). [13]C NMR (DMSO-$d_6$, 125 MHz): $\delta$ 165.5, 164.8, 164.1, 151.34, 151.29, 145.0, 144.0, 141.9, 140.1, 133.2, 132.5, 128.7, 128.4, 127.6, 126.2, 125.5, 124.4, 122.6, 122.3, 120.4, 120.3, 118.4, 114.7, 112.5, 111.6, 86.5, 75.1, 75.0, 69.3, 65.6, 62.5, 28.32, 28.26, 19.56, 19.54.

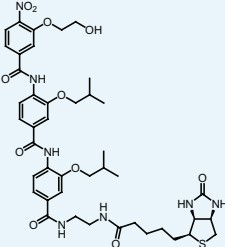

**Appendix 1—chemical structure 21.** Compound 12.

To a solution of compound **11** (1.5 g, 1.7 mmol) in DCM (50 mL) were added Pd(PPh$_3$)$_4$ (196 mg, 0.17 mmol) and phenylsilane (0.37 mL, 3.4 mmol), and the mixture was stirred at room temperature for 1 hr. The reaction mixture was then concentrated under reduced pressure and the resulting solid was washed with ether. The corresponding carboxylic acid was obtained as a white solid (1.45 g, quantitative yield) and used in the next reaction without further purification.

To a solution of the carboxylic acid (300 mg, 0.35 mmol) and DIEA (0.24 mL, 1.4 mmol) in DMF (10 mL) was added HATU (179 mg, 0.46 mmol) and the mixture was stirred at room temperature for 1 hr. N-Boc-ethylenediamine (112 mg, 0.70 mmol) was added and the resulting mixture was stirred at room temperature for 24 hr. The solution was then poured into 5% citric acid (20 mL) and extracted with EtOAc (20 mL x 2). The combined organic extracts were washed with saturated NaHCO$_3$ (20 mL x 2), dried over Na$_2$SO$_4$, filtered, and concentrated under reduced pressure. Recrystallization from EtOAc/hexanes (1:1) gave compound **12** as a light yellow solid (152 mg, 44%). [1]H NMR (DMSO-$d_6$, 500 MHz): δ 9.86 (br s, 1 hr), 9.41 (br s, 1 hr), 8.48 (t, $J$ = 5.6 Hz, 1 hr), 8.06 (d, $J$ = 8.2 Hz, 1 hr), 8.03 (d, $J$ = 8.2 Hz, 1 hr), 7.99 (d, $J$ = 8.2 Hz, 1 hr), 7.98 (br s, 1 hr), 7.71 (dd, $J$ = 8.2, 1.5 Hz, 1 hr), 7.61–7.58 (m, 2 hr), 7.55–7.51 (m, 2 hr), 7.42–7.40 (m, 6 hr), 7.36–7.32 (m, 6 hr), 7.29–7.26 (m, 3 hr), 6.95 (t, $J$ = 5.6 Hz, 1 hr), 4.51–4.50 (m, 2 hr), 3.92 (d, $J$ = 6.2 Hz, 2 hr), 3.91 (d, $J$ = 6.2 Hz, 2 hr), 3.33–3.30 (m, 4 hr), 3.15–3.11 (m, 2 hr), 2.17–2.06 (m, 2 hr), 1.05 (d, $J$ = 6.7 Hz, 6 hr), 0.99 (d, $J$ = 6.7 Hz, 6 hr).

**Appendix 1—chemical structure 22.** ERX-11-biotin.

A solution of compound **12** (50 mg, 0.050 mmol) in TFA/TIS/H$_2$O (95:2.5:2.5, 5 mL) was stirred at room temperature for 30 min. The excess of TFA were removed with a gentle stream of nitrogen, and a white solid was precipitated with cold diethyl ether. The precipitate was filtered, washed with diethyl ether and dried in vacuo. The resulting yellow solid was used in the next reaction without further purification.

To a solution of the yellow solid, biotin (24 mg, 0.10 mmol) and DIEA (0.035 mL, 0.20 mmol) in DMF (5 mL) was added HATU (43 mg, 0.11 mmol), and the resulting mixture was stirred at room temperature for 12 hr. The reaction solution was then poured into 1N HCl (20 mL) and extracted with EtOAc (20 mL x 2). The combined organic extracts were washed with saturated $NaHCO_3$ (20 mL x 2) and concentrated under reduced pressure. The resulting solid was washed with EtOAc (10 mL x 5) and dried in vacuo giving **ERX-11-biotin** as a light yellow solid (32 mg, 73%). [1]H NMR (DMSO-$d_6$, 500 MHz): δ 9.87 (br s, 1 hr), 9.47 (br s, 1 hr), 8.53 (t, J = 5.5 Hz, 1 hr), 8.03 (d, J = 8.2 Hz, 1 hr), 7.99 (d, J = 8.2 Hz, 1 hr), 7.97 (br s, 1 hr), 7.94 (d, J = 8.2 Hz, 1 hr), 7.87 (d, J = 1.5 Hz, 1 hr), 7.65–7.64 (m, 2 hr), 7.62 (dd, J = 8.2, 1.8 Hz, 1 hr), 7.56 (d, J = 1.5 Hz, 1 hr), 7.52 (dd, J = 8.2, 1.7 Hz, 1 hr), 6.43 (br s, 1 hr), 6.36 (br s, 1 hr), 4.98 (t, J = 5.6 Hz, 1 hr), 4.32–4.29 (m, 3 hr), 4.13–4.10 (m, 1 hr), 3.93 (d, J = 6.4 Hz, 2 hr), 3.92 (d, J = 6.4 Hz, 2 hr), 3.79–3.76 (m, 2 hr), 3.33–3.31 (m, 2 hr, overlap with $H_2O$ peak), 3.27–3.23 (m, 2 hr), 3.09–3.05 (m, 2 hr), 2.82 (dd, J = 12.5, 5.2 Hz, 1 hr), 2.58 (d, J = 12.2 Hz, 2 hr), 2.16–2.08 (m, 4 hr), 1.64–1.43 (m, 4 hr), 1.35–1.28 (m, 2 hr), 1.05 (d, J = 6.7 Hz, 6 hr), 1.02 (d, J = 6.7 Hz, 6 hr). [13]C NMR (DMSO-$d_6$, 125 MHz): δ 173.0, 166.2, 164.7, 163.2, 162.8, 151.6, 151.4, 150.4, 141.9, 139.7, 132.4, 131.7, 130.3, 130.2, 125.5, 124.6, 122.9, 120.3, 120.13, 120.11, 114.8, 111.6, 111.4, 75.1, 75.0, 72.0, 61.5, 59.70, 59.68, 55.9, 38.8, 36.3, 28.7, 28.5, 28.4, 28.3, 25.7, 19.63, 19.55. HRMS-ESI (m/z): [M-H][-] calcd for C43H55N7O11S: 876.3608, found: 876.3602.

## General procedure for peptide synthesis

The peptides were synthesized manually using standard $N^\alpha$-Fmoc/$t$Bu solid-phase peptide synthesis protocol. Aminomethylated polystyrene resin (0.25 mmol, 0.4 mmol/g) was swollen in DMF for 2 hr and washed with DMF (3 × 1 min). Fmoc-Rink amide linker (1.5 equiv), HBTU (1.5 equiv), Cl-HOBt (1.5 equiv), and DIEA (three equiv) were dissolved in DMF (6 mL). The resulting solution was then added to the resin and shaken for 12 hr. The coupling reaction was followed by Kaiser ninhydrin test, and unreacted amines were capped by using acetic anhydride (20 equiv) in DMF (6 mL) for 20 min. The Fmoc protecting group of the Rink amide linker was removed via treatment with 20% piperidine in DMF (6 mL, 1 × 5 min and 1 × 30 min) and washed with DMF (3 × 1 min). The first amino acid was introduced by using a preactivated Fmoc amino acid that was prepared by mixing a Fmoc amino acid (four equiv), HBTU (four equiv), Cl-HOBt (four equiv), and DIEA (eight equiv) in DMF (6 mL) for 5 min. The coupling reaction was conducted for 4 hr or until Kaiser ninhydrin test became negative. When a coupling reaction was found to be incomplete, the resin was washed with DMF (3 × 1 min) and the amino acid was coupled again with a freshly prepared preactivated Fmoc amino acid. When the second coupling reaction did not result in negative Kaiser ninhydrin test, the resin was washed with DMF (3 × 1 min) and the unreacted amines were capped by being treated with acetic anhydride (20 equiv) in DMF for 20 min. These steps (removal of a Fmoc group and coupling of a Fmoc amino acid) were repeated until all amino acids in the sequence of a peptide were coupled. The resin was then washed with DCM (5 × 1 min) and dried in vacuo.

## General procedure for cleavage and final deprotection of peptides

A cleavage mixture of trifluoroacetic acid (TFA), dimethyl sulfide, 1,2-ethanedithiol, and anisole (8 mL, 90:5:3:2) was added to a peptide on dried resin (0.25 mmol) in a disposable 50 mL polypropylene tube, and the mixture was stirred for 90 min at room temperature in the dark. Then, the TFA solution was filtered, and the resin was washed with TFA (3 mL) and DCM (3 mL). The combined TFA solution was concentrated to a volume of approximately 3 mL with a gentle stream of nitrogen, and the peptide was precipitated with cold diethyl ether (40 mL). The precipitated peptide was centrifuged, and the ether solution was decanted to remove the scavengers. Washing with cold diethyl ether was repeated, and the precipitated peptide was centrifuged, decanted, and dried under vacuum. The resulting peptide was characterized via RP-HPLC and MALDI-TOF MS.

## General procedure for the purification of peptides

A crude peptide was dissolved in 50% aqueous acetic acid, and the insoluble was removed by centrifugation. The acetic acid solution containing peptide was purified with HPLC by using a reverse phase semipreparative Vydac column (C4-bonded, 214TP1010, 10 mm × 250 mm, 10 μm) with gradient elution at a flow rate of 3.0 mL/min. A fraction containing the peptide was collected and lyophilized. The purity of all of the synthesized peptides was checked by analytical HPLC and found to be greater than 95%. The molecular mass of the purified peptides was confirmed by MALDI-TOF MS (*Appendix 1—Table 1*).

**Appendix 1—Table 1.** Characterization of the peptides

| LXXLL Peptide | Sequence | molecular mass (MALDI-TOF-MS) | |
| --- | --- | --- | --- |
| | | calculated | observed |
| SRC1 | Ac-LTARHKILHRLLQEGSPSD-NH$_2$ | $[M + H]^+$ for C96H162N32O28: 2212.2 | 2212.4 |
| SRC2 | Ac-DSKGQTKLLQLLTTKSDQM-NH$_2$ | $[M + H]^+$ for C92H162N26O32: 2176.2 | 2176.6 |
| AIB1 | Ac-ESKGHKKLLQLLTCSSDDR-NH$_2$ | $[M + H]^+$ for C92H159N29O31: 2199.2 | 2199.8 |
| PELP1 | Ac-SIKTRFEGLCLLSLLVGESPT-NH$_2$ | $[M + H]^+$ for C103H174N26O31: 2304.3 | 2304.6 |

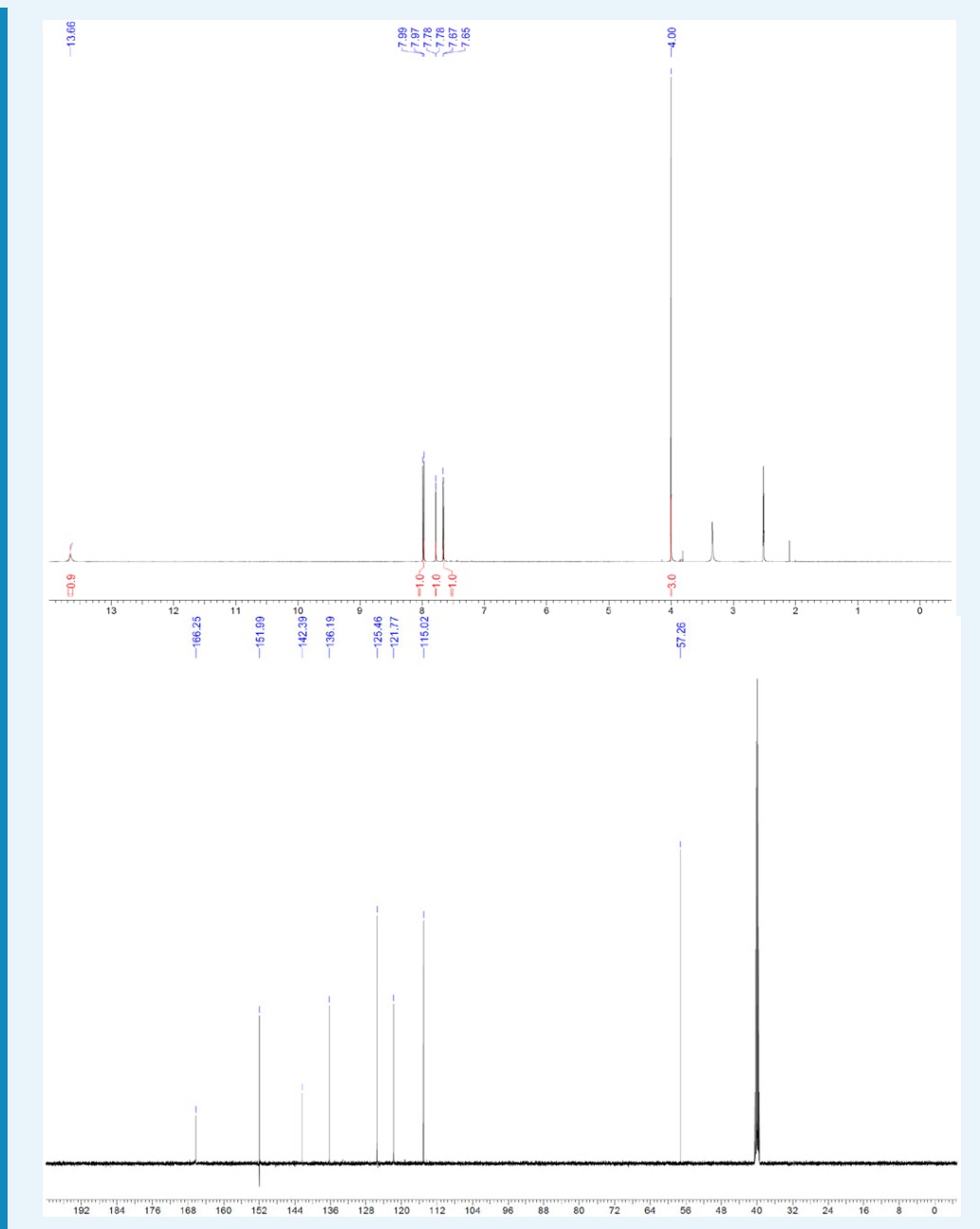

**Appendix 1—figure 1.** $^1$H and $^{13}$C NMR of compound **3a**.

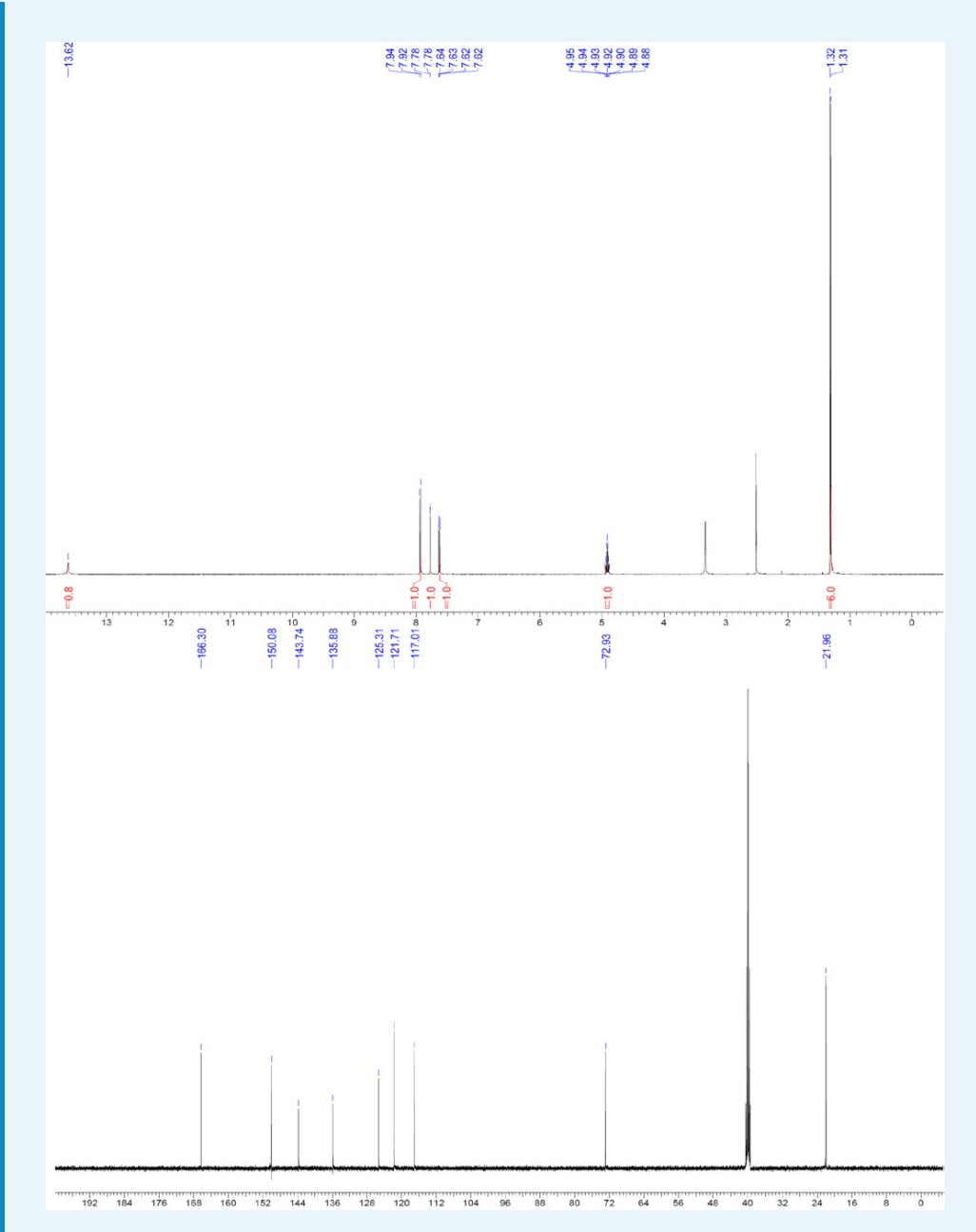

**Appendix 1—figure 2.** [1]H and [13]C NMR of compound **3b**.

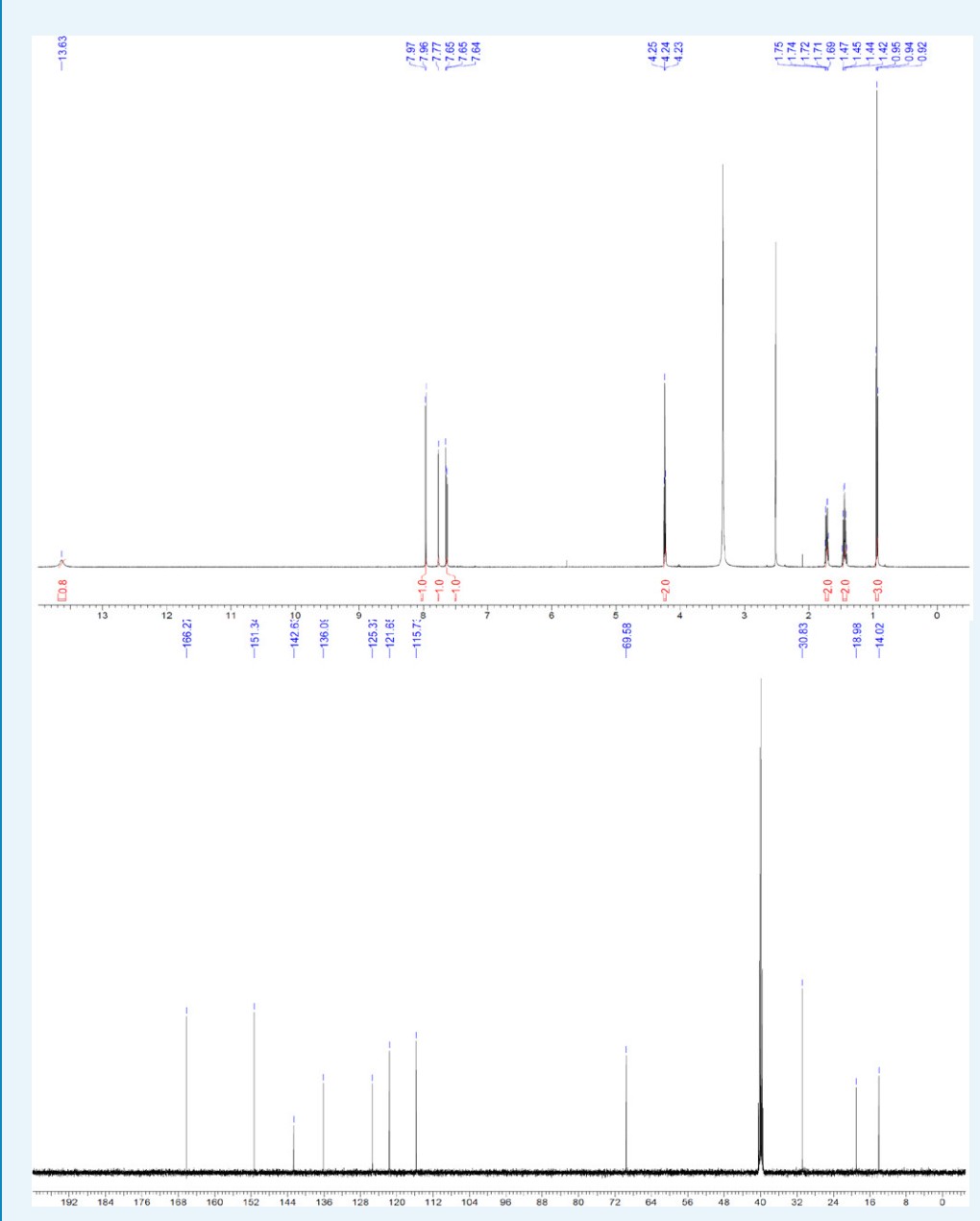

**Appendix 1—figure 3.** [1]H and [13]C NMR of compound **3c**.

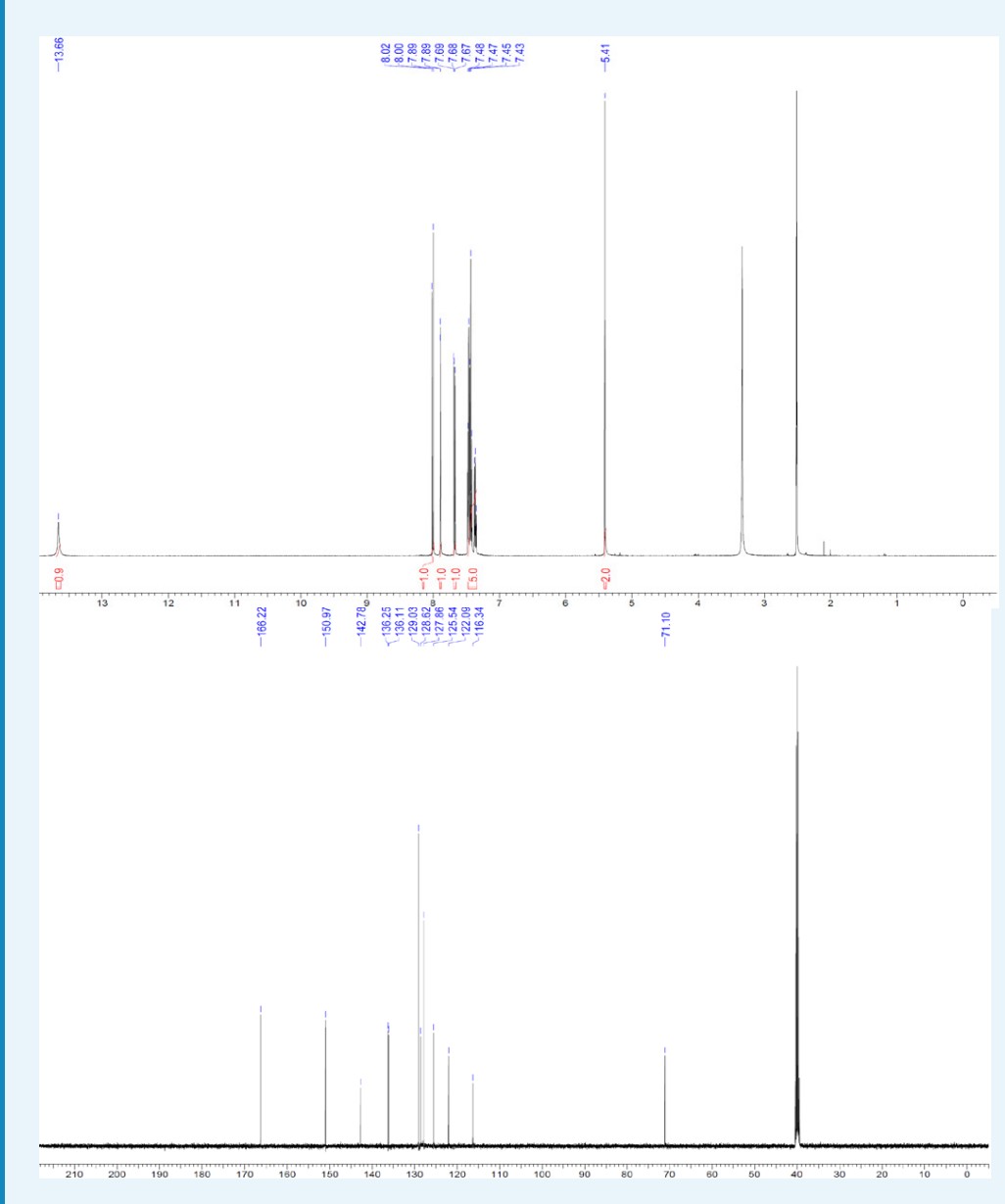

**Appendix 1—fFigure 4.** [1]H and [13]C NMR of compound **3e**.

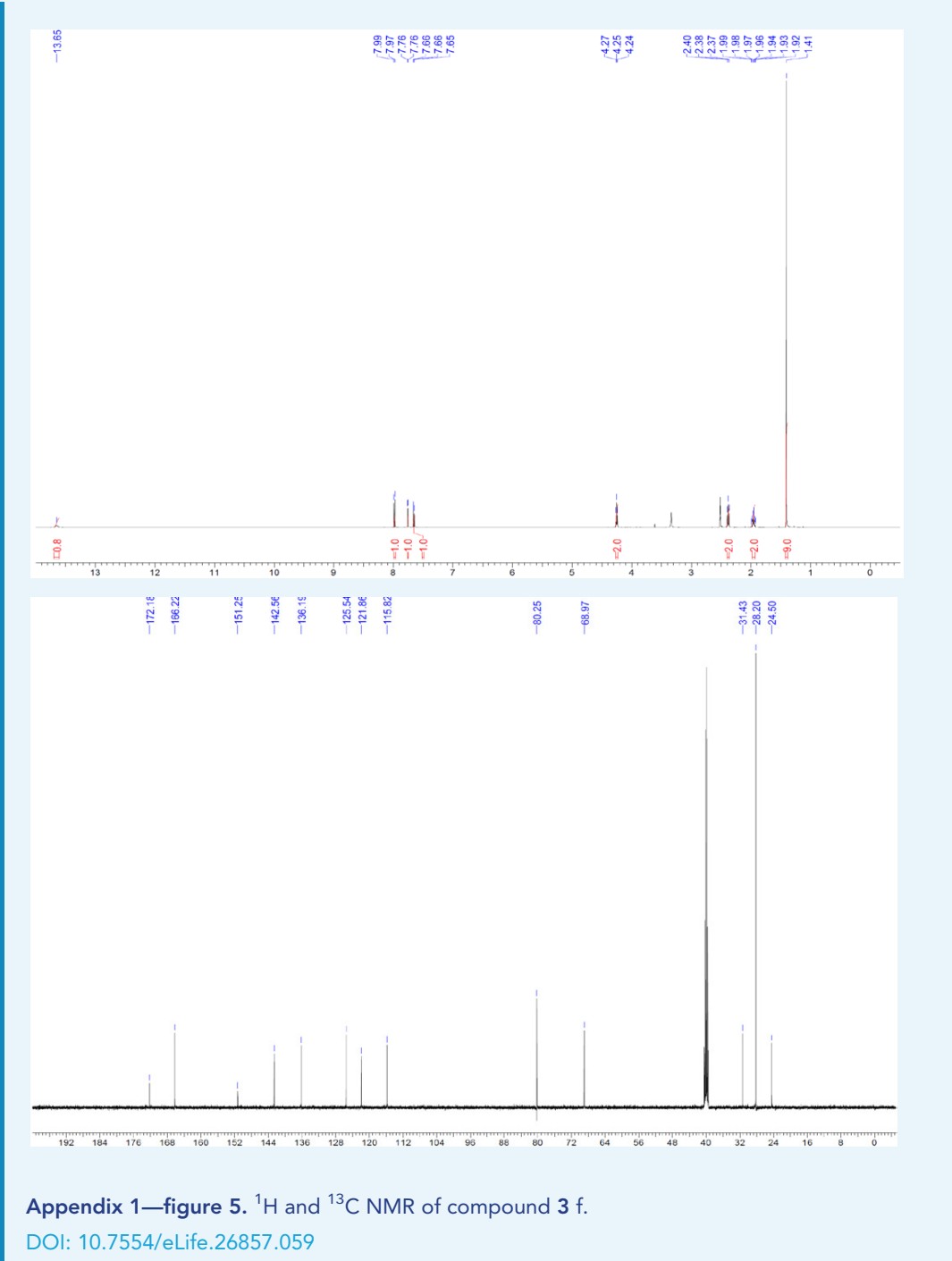

**Appendix 1—figure 5.** $^1$H and $^{13}$C NMR of compound **3** f.

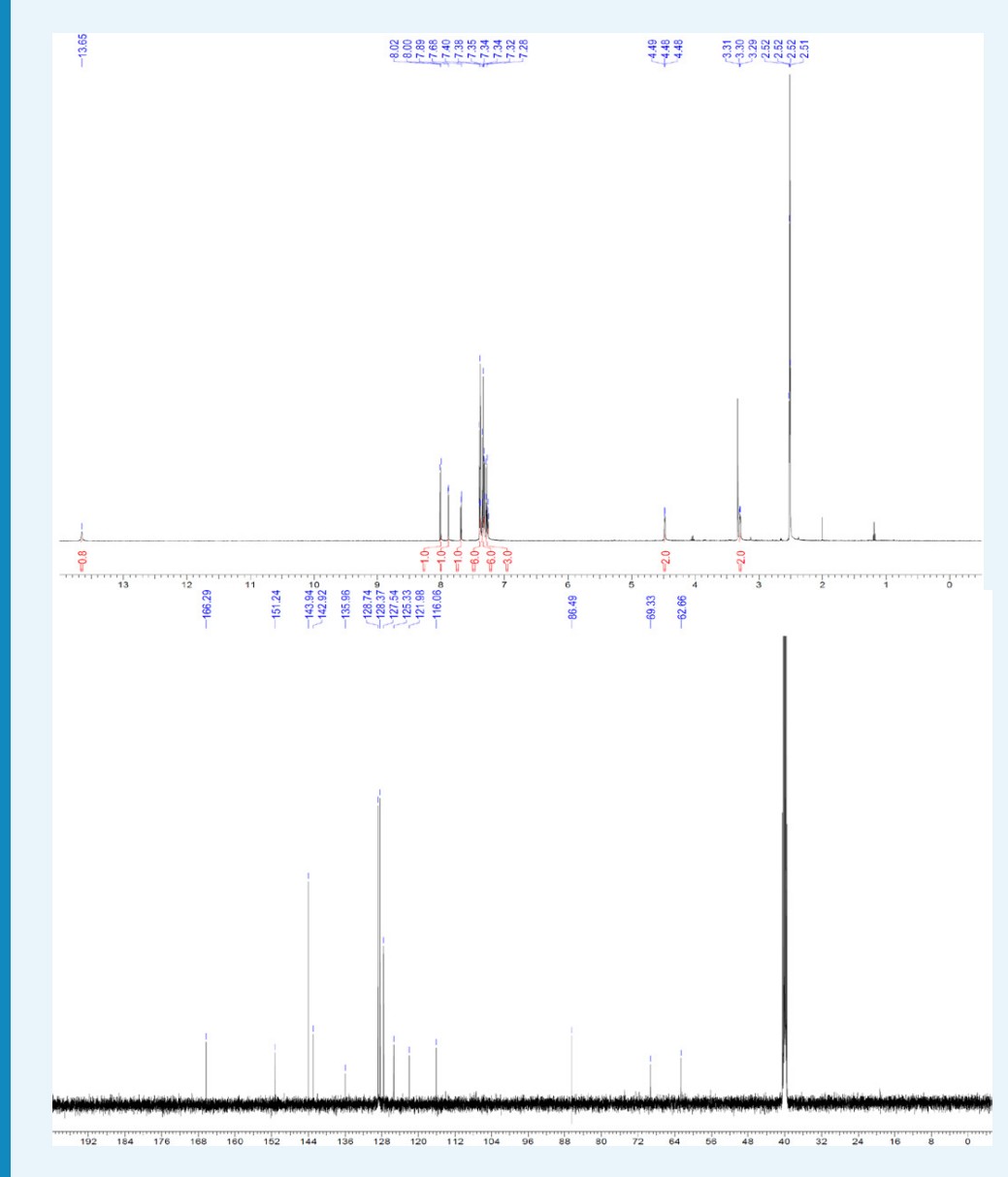

**Appendix 1—figure 6.** [1]H and [13]C NMR of compound **3** g.

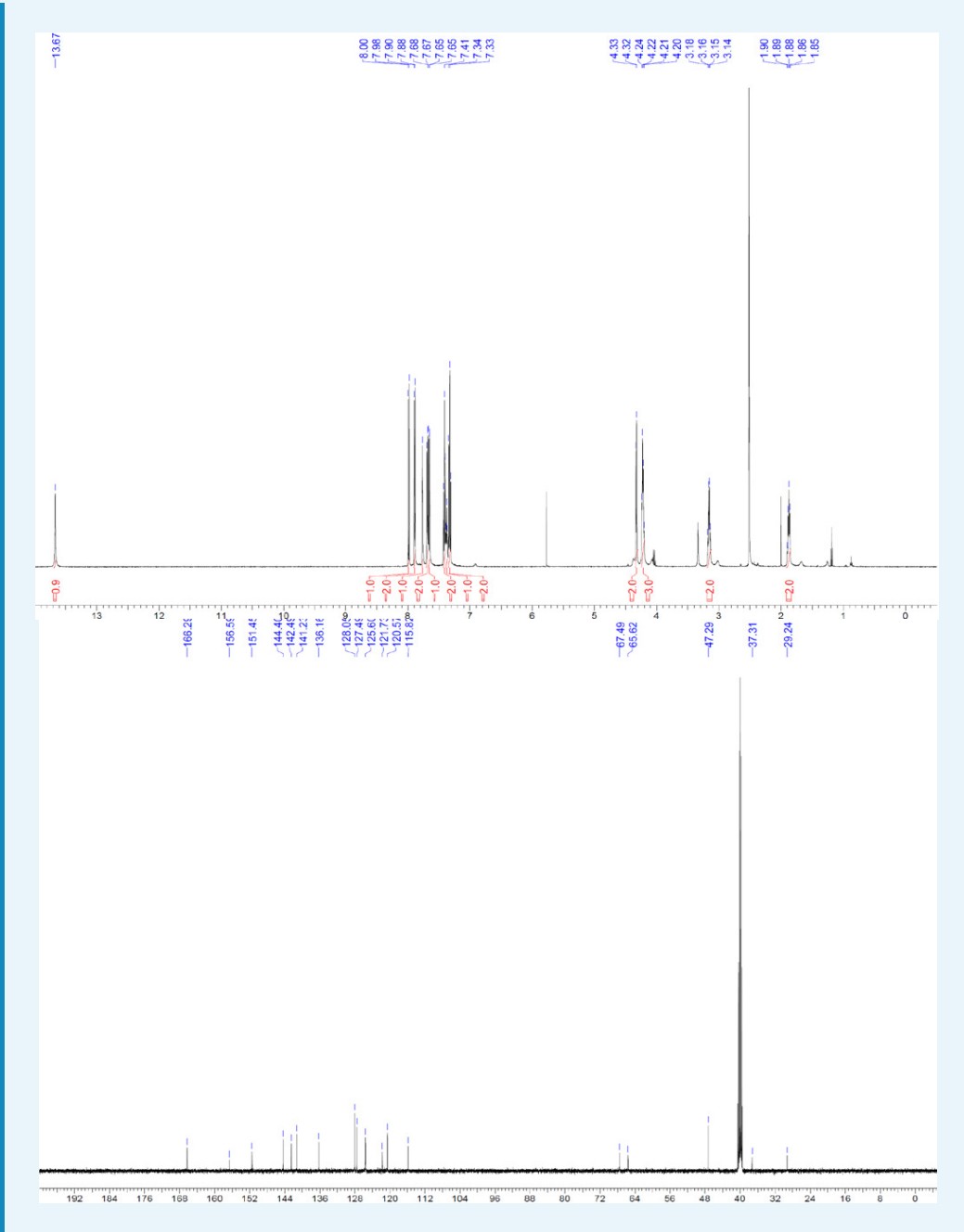

**Appendix 1—figure 7.** [1]H and [13]C NMR of compound **3i**.

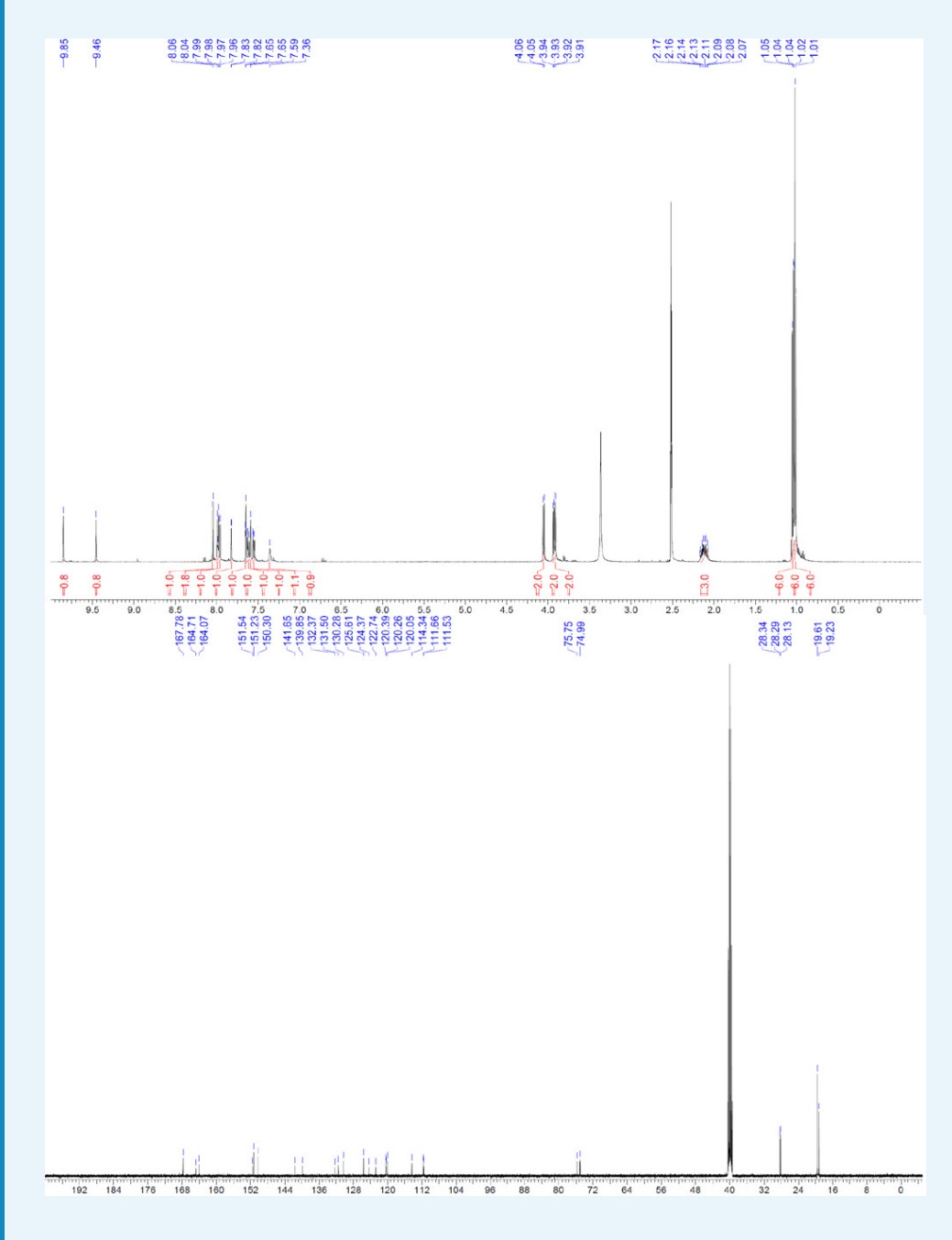

**Appendix 1—figure 8.** $^1$H and $^{13}$C NMR of **ERX-5**.

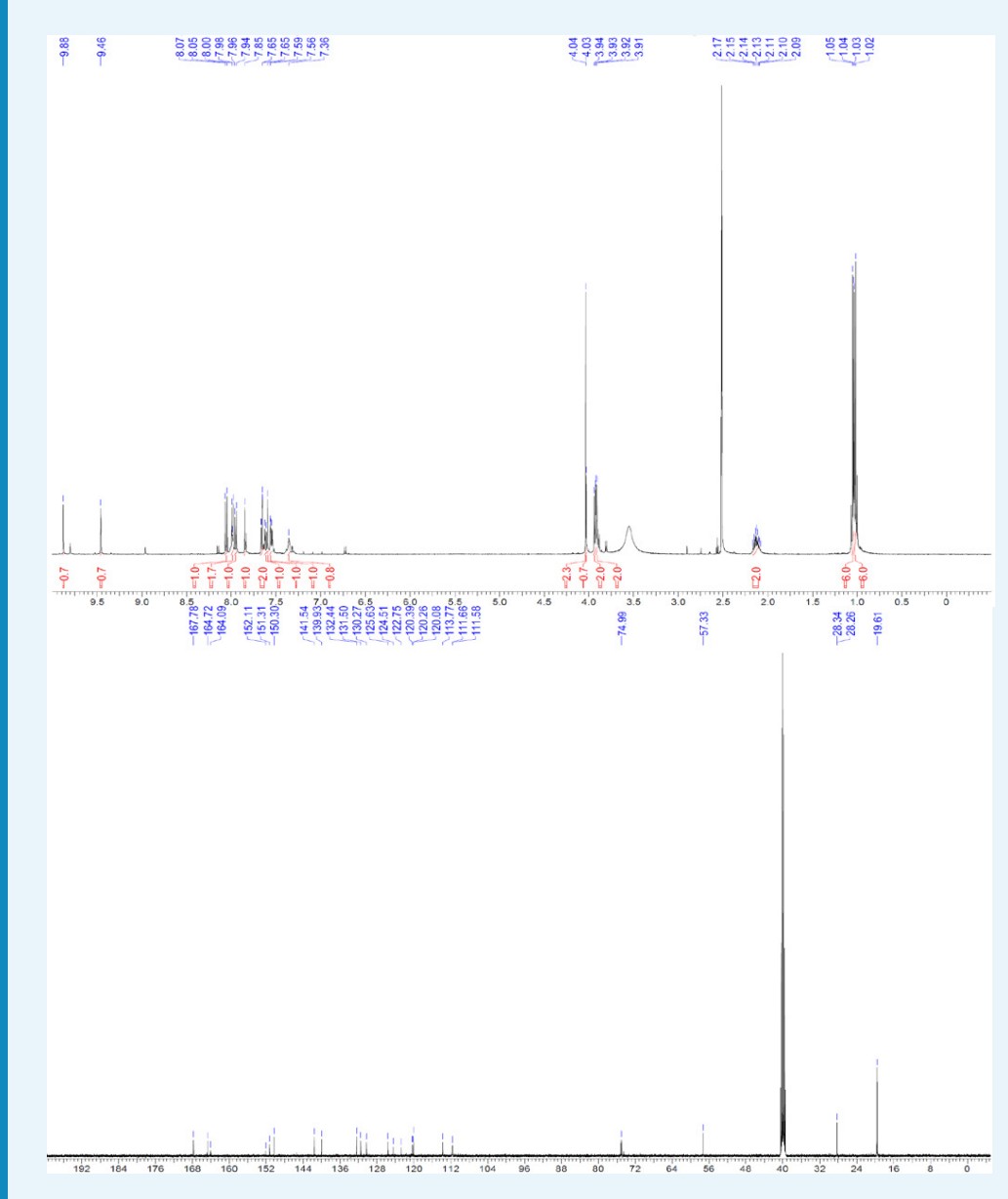

**Appendix 1—figure 9.** [1]H and [13]C NMR of **ERX-7**.

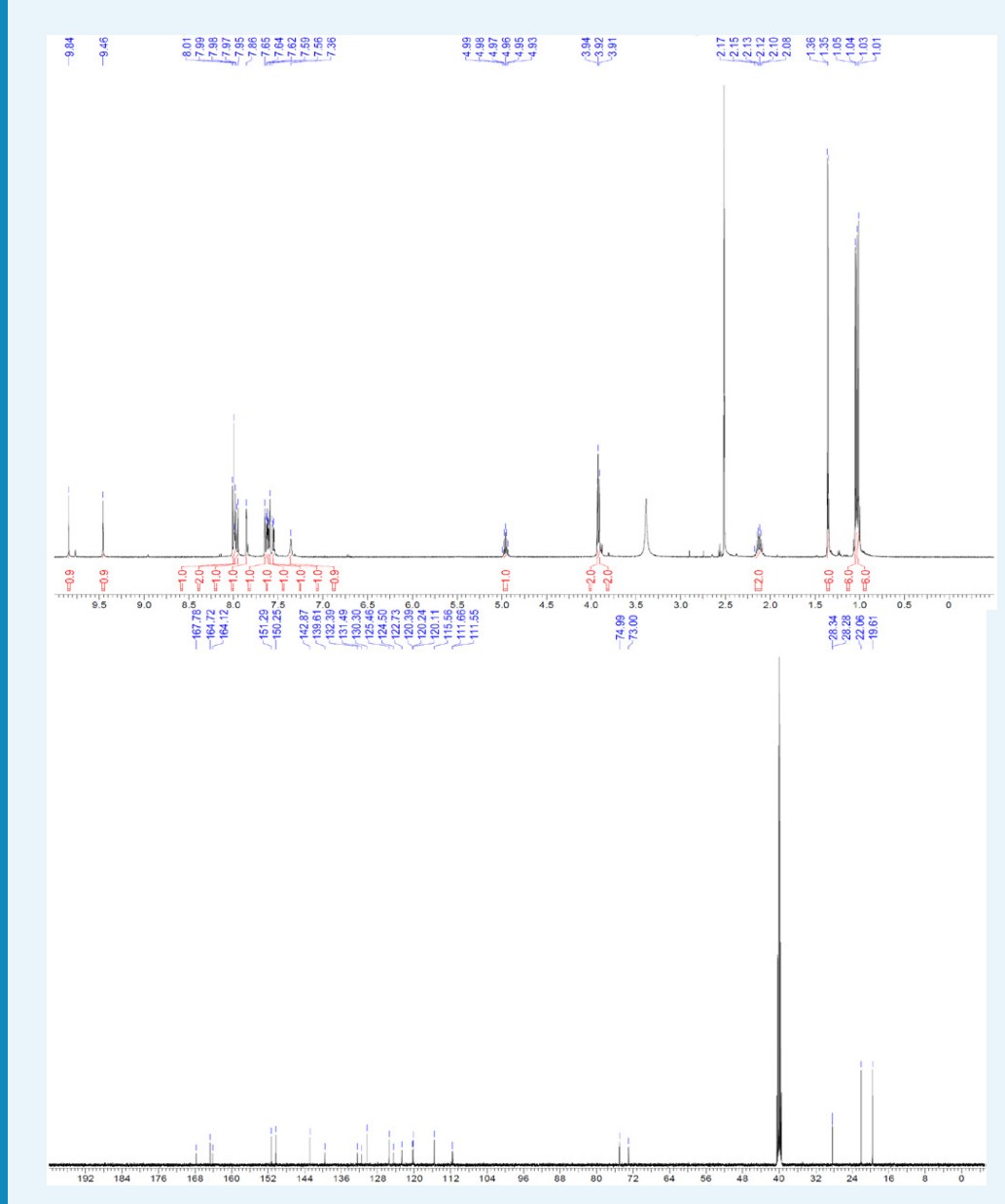

**Appendix 1—figure 10.** $^{1}$H and $^{13}$C NMR of **ERX-8**.

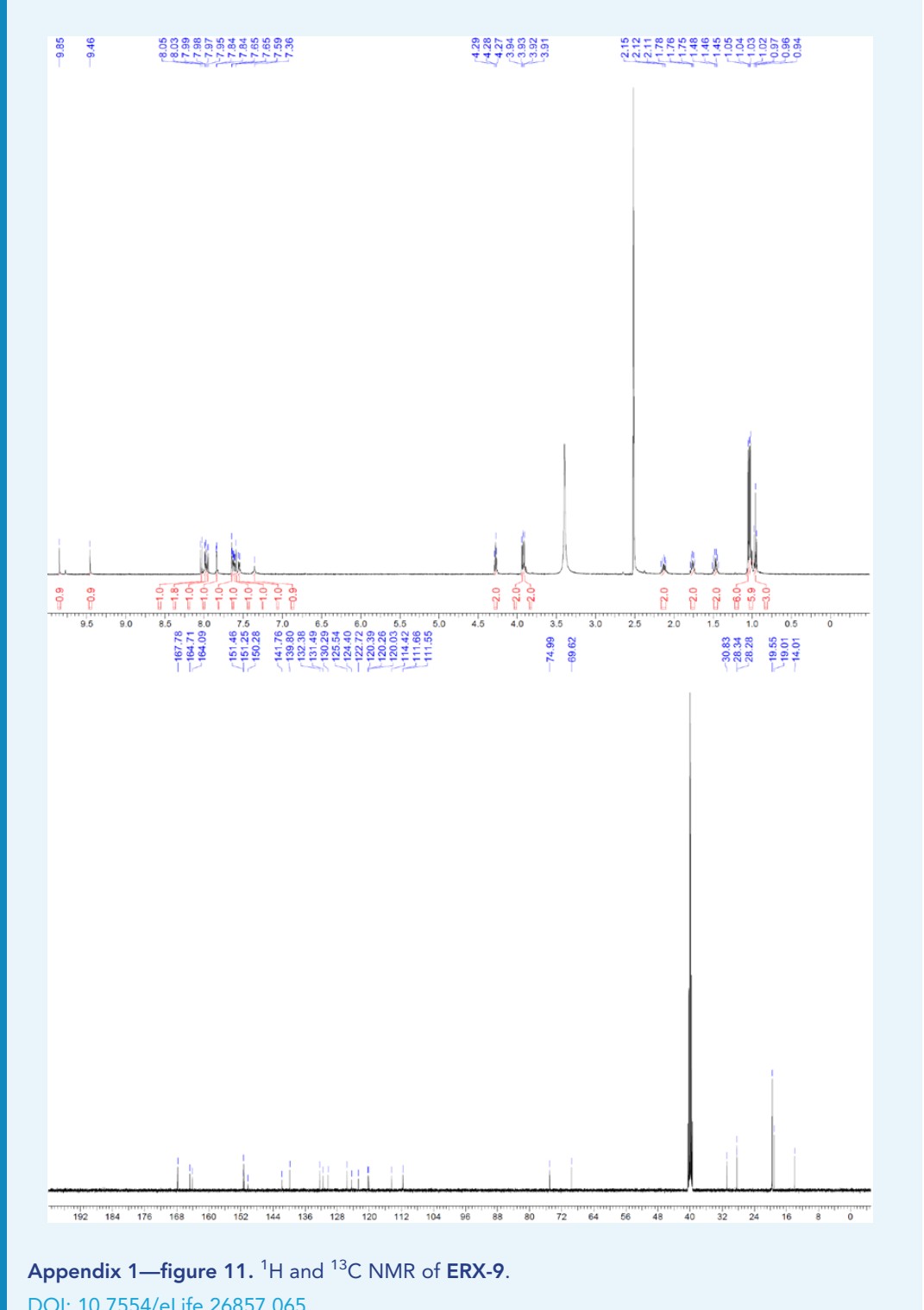

**Appendix 1—figure 11.** [1]H and [13]C NMR of **ERX-9**.

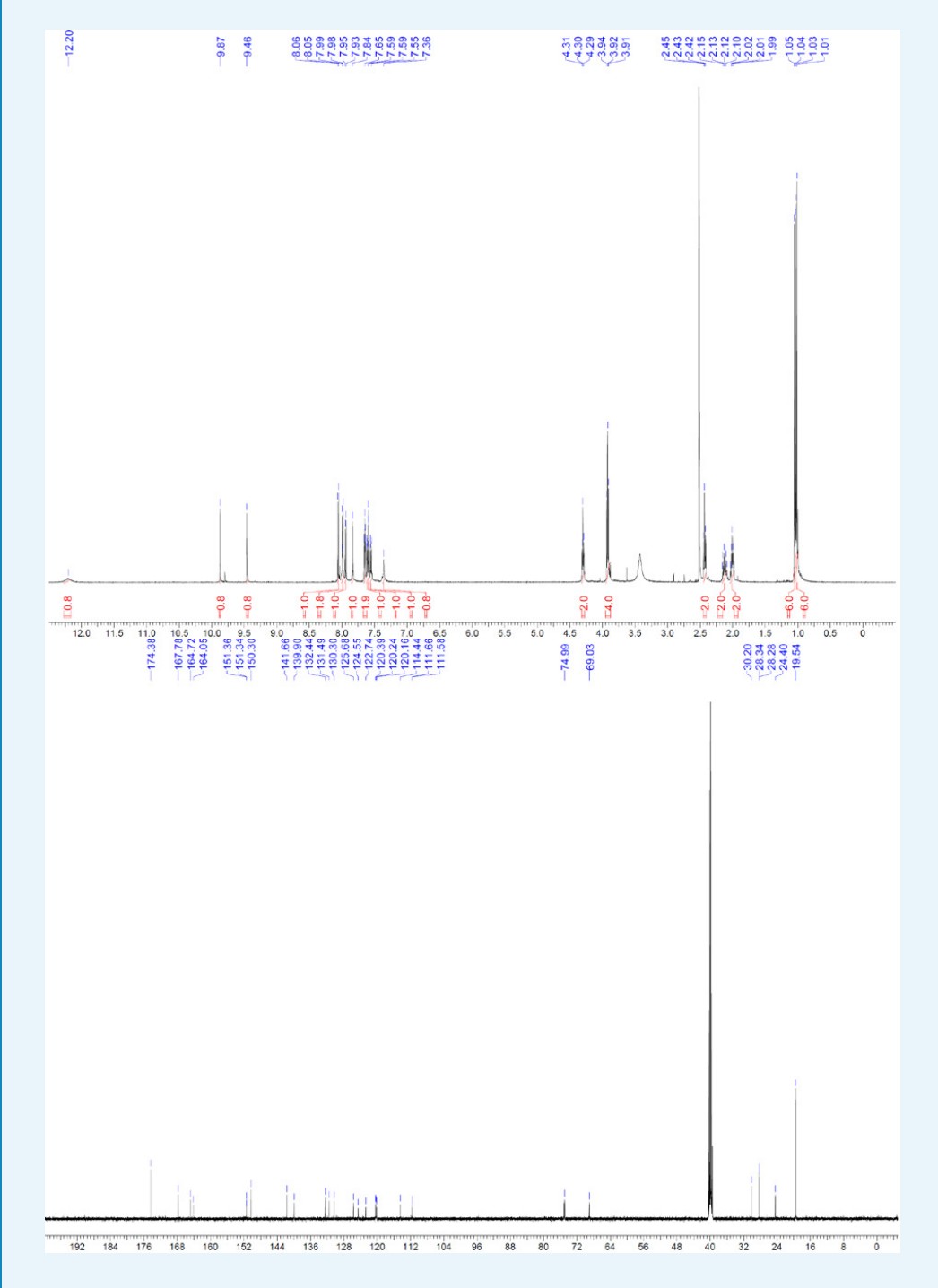

**Appendix 1—figure 12.** [1]H and [13]C NMR of **ERX-10**.

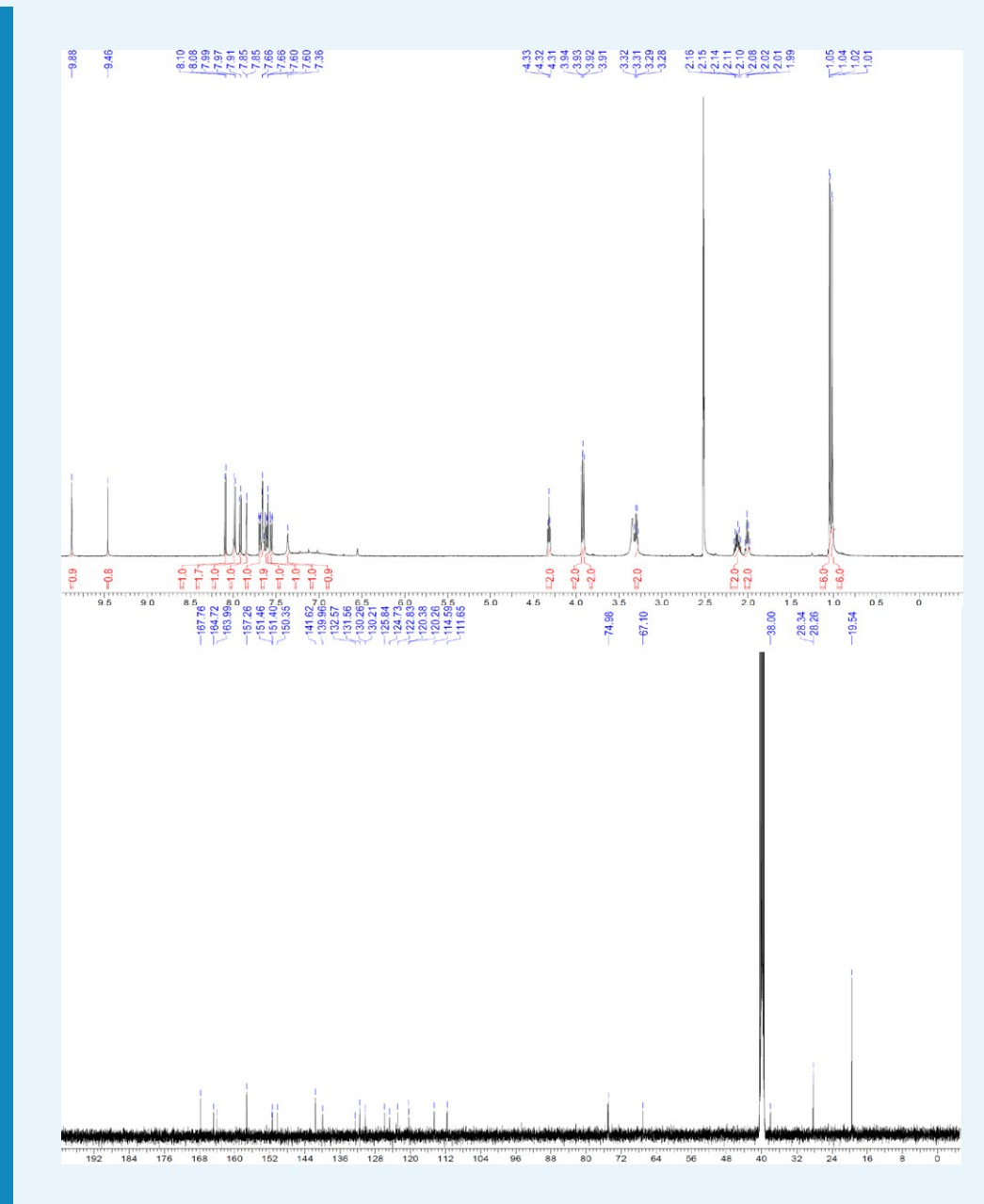

**Appendix 1—figure 13.** $^1$H and $^{13}$C NMR of **ERX-12**.

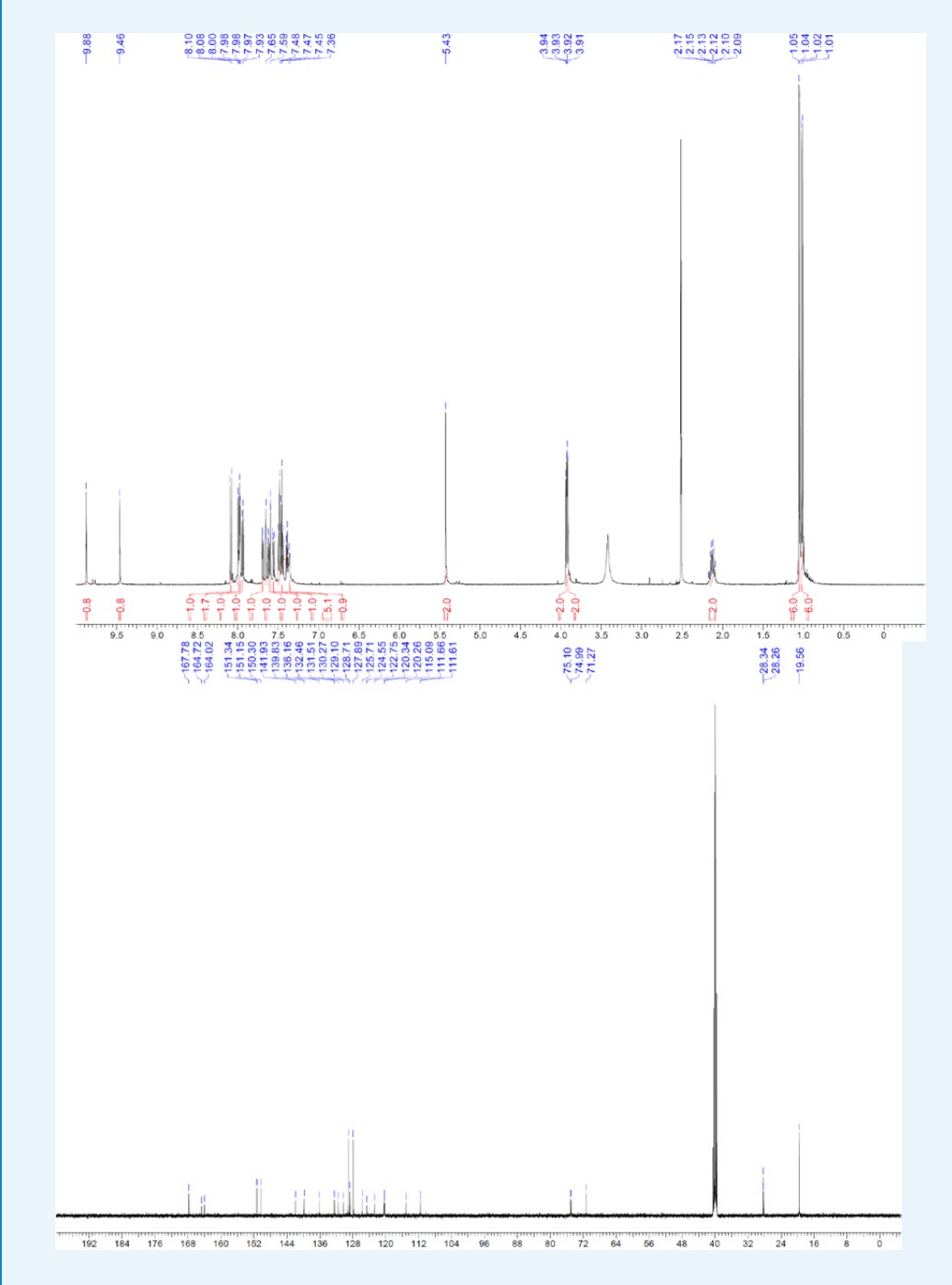

**Appendix 1—figure 14.** [1]H and [13]C NMR of **ERX-13**.

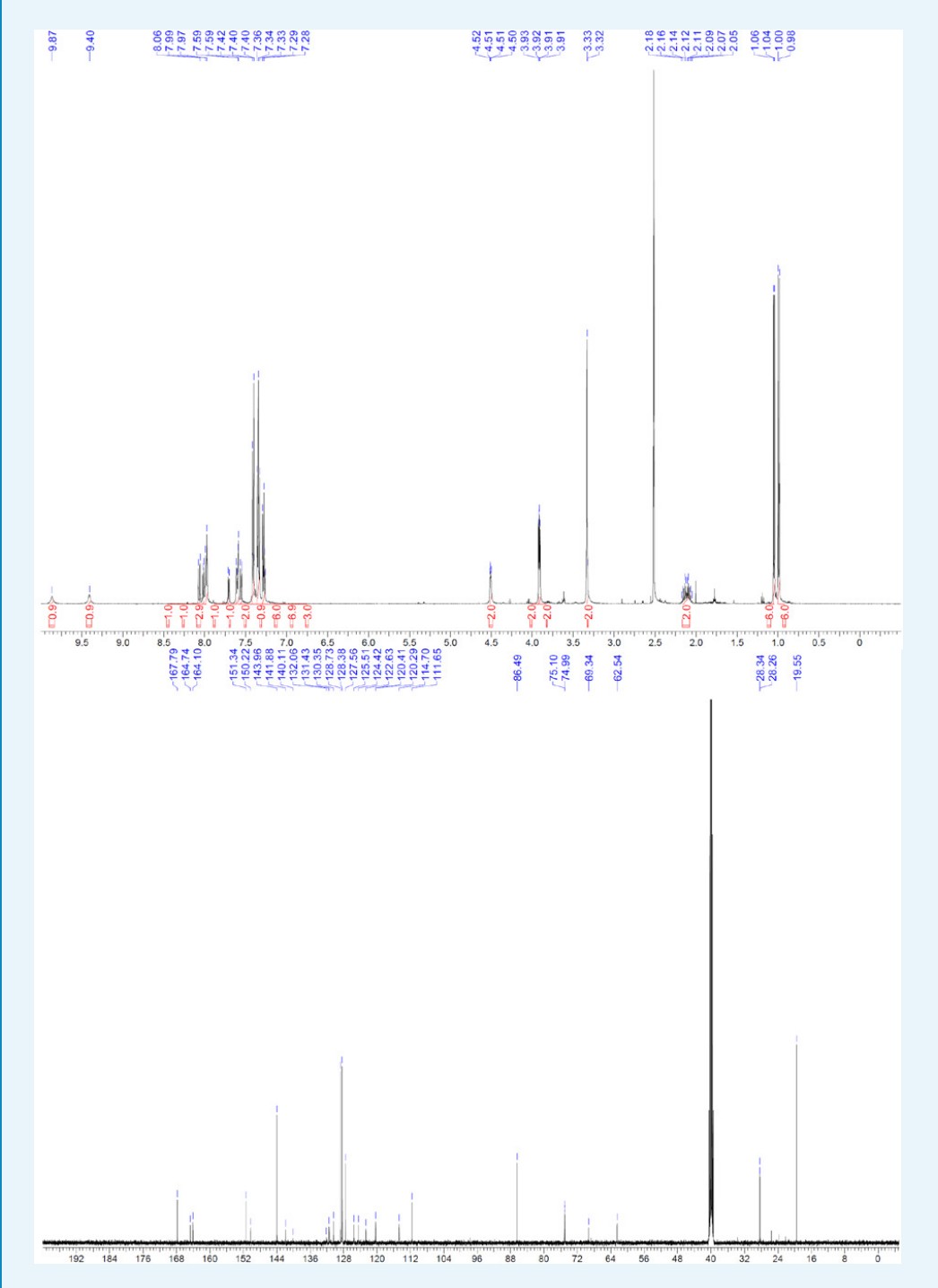

**Appendix 1—figure 15.** $^1$H and $^{13}$C NMR of compound **9**.

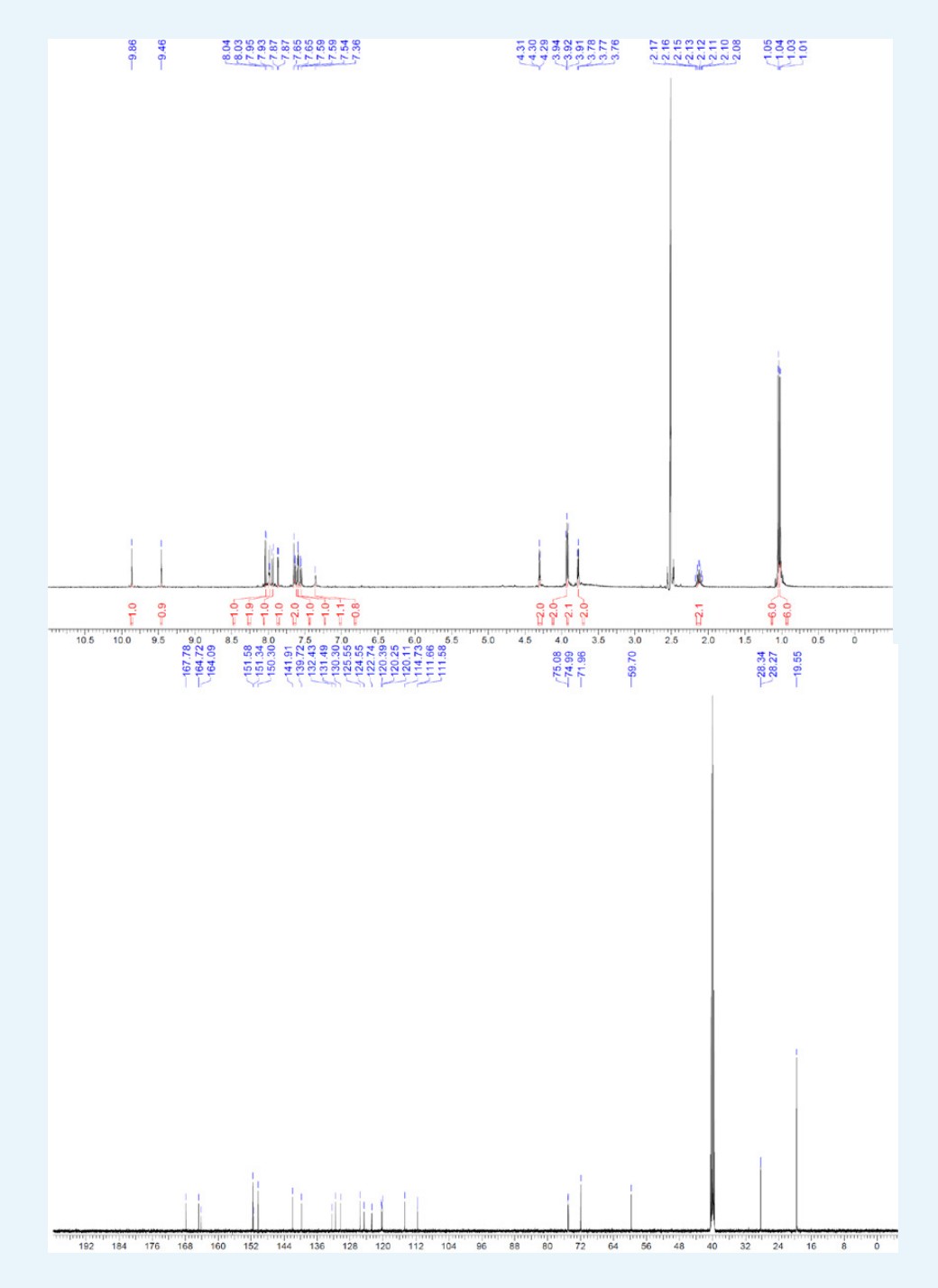

**Appendix 1—figure 16.** $^1$H and $^{13}$C NMR of **ERX-11**.

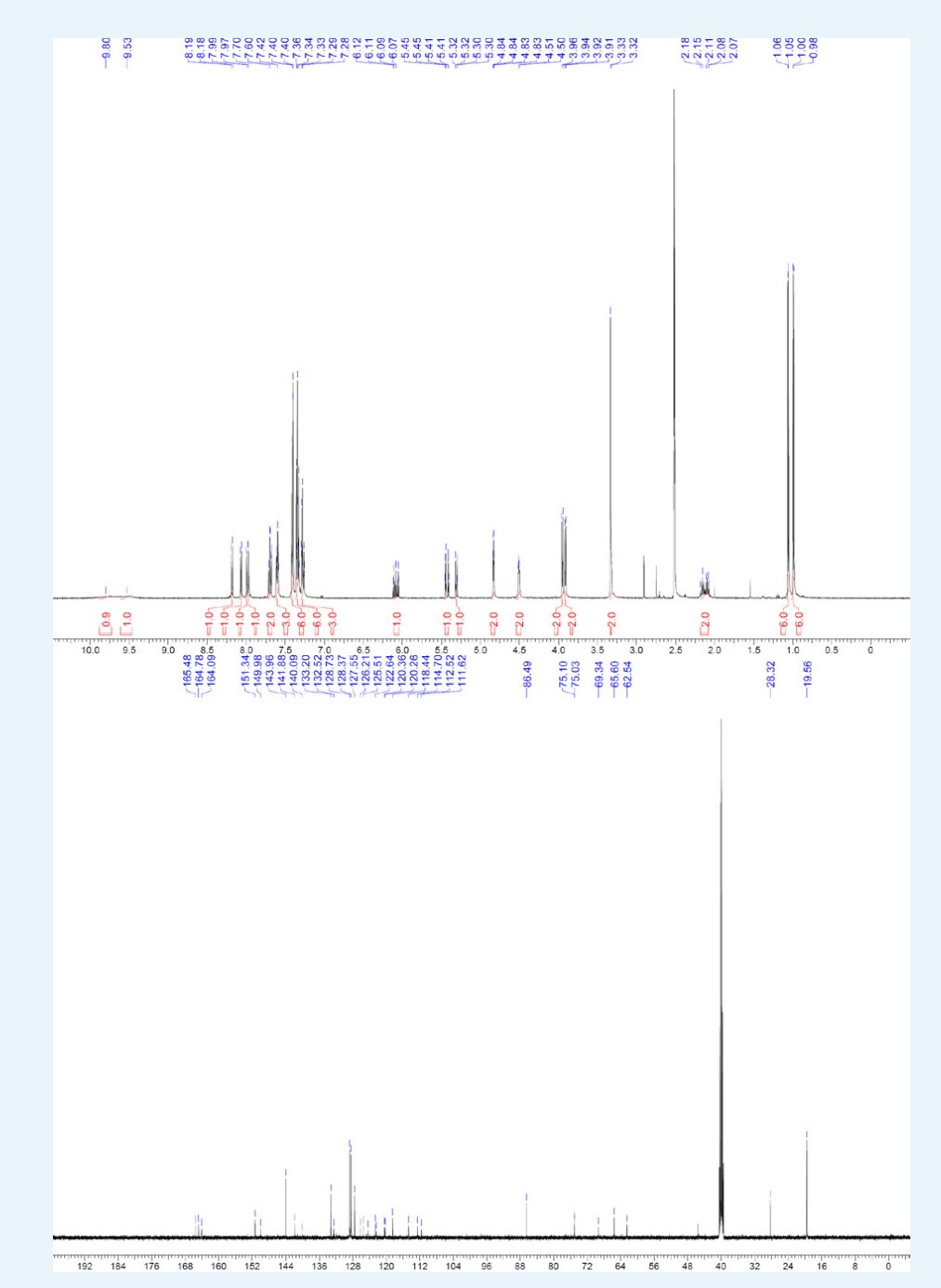

**Appendix 1—figure 17.** [1]H and [13]C NMR of compound **11**.

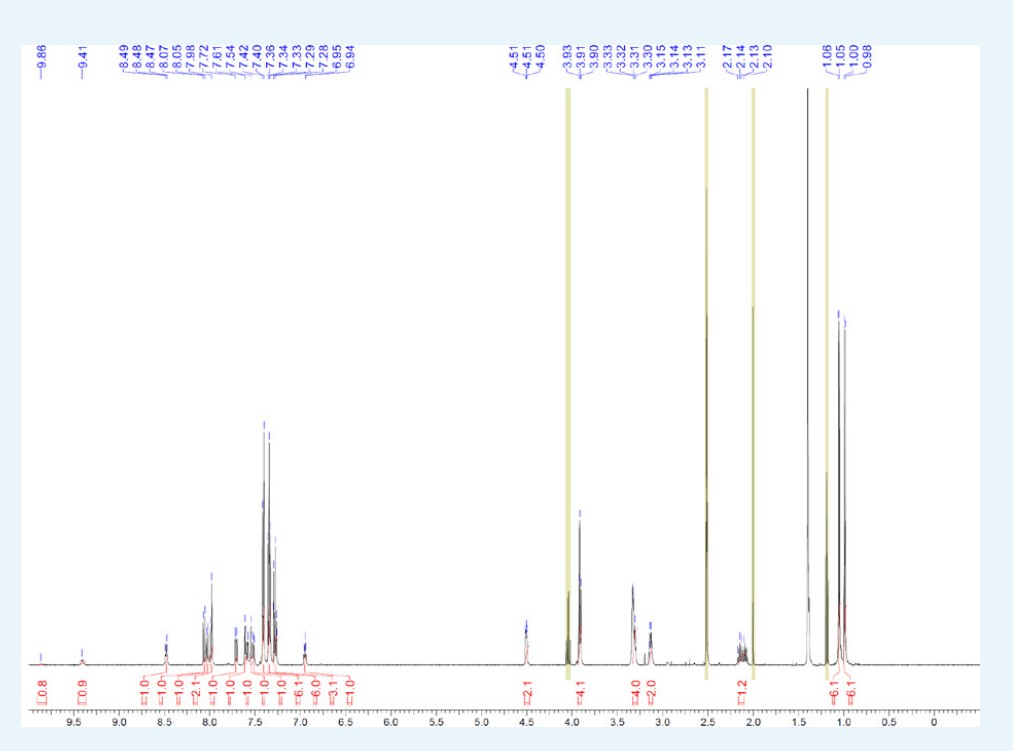

**Appendix 1—figure 18.** [1]H NMR of compound **12**.

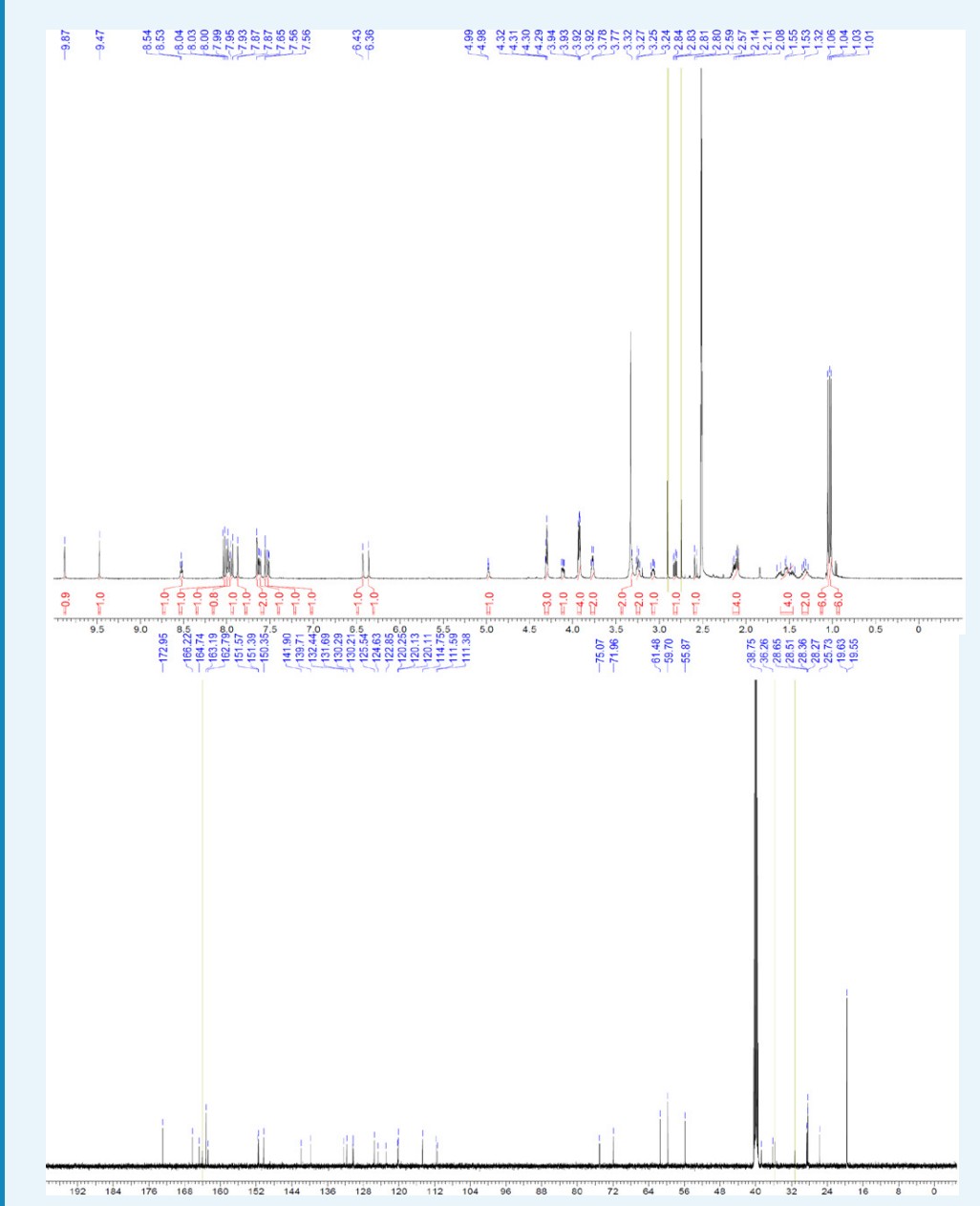

**Appendix 1—figure 19.** [1]H and [13]C NMR of compound **ERX-11-biotin**.

(a) HPLC chromatogram of SRC1-LXXLL peptide

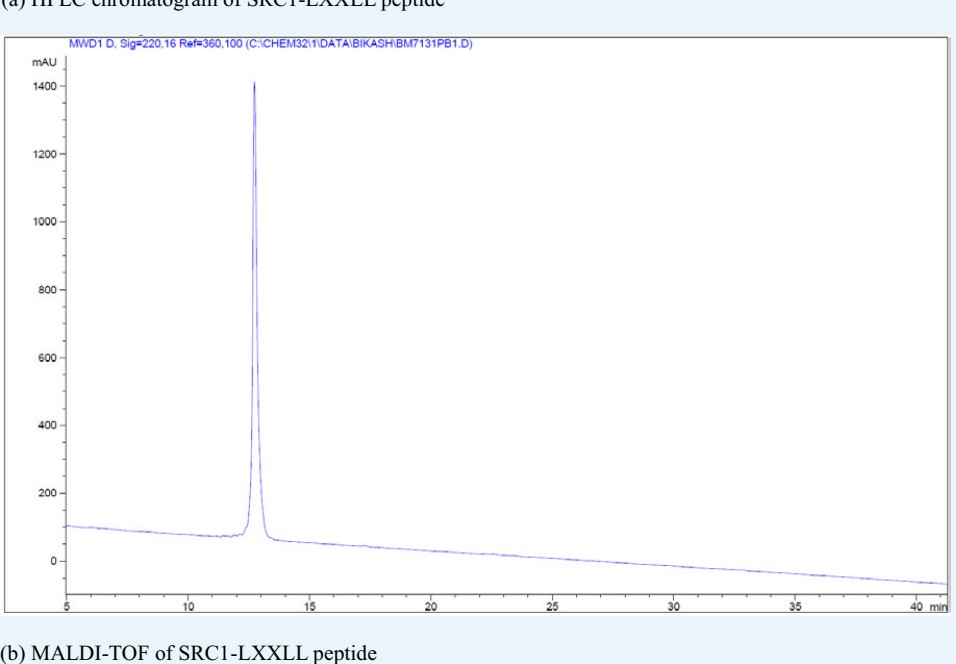

(b) MALDI-TOF of SRC1-LXXLL peptide

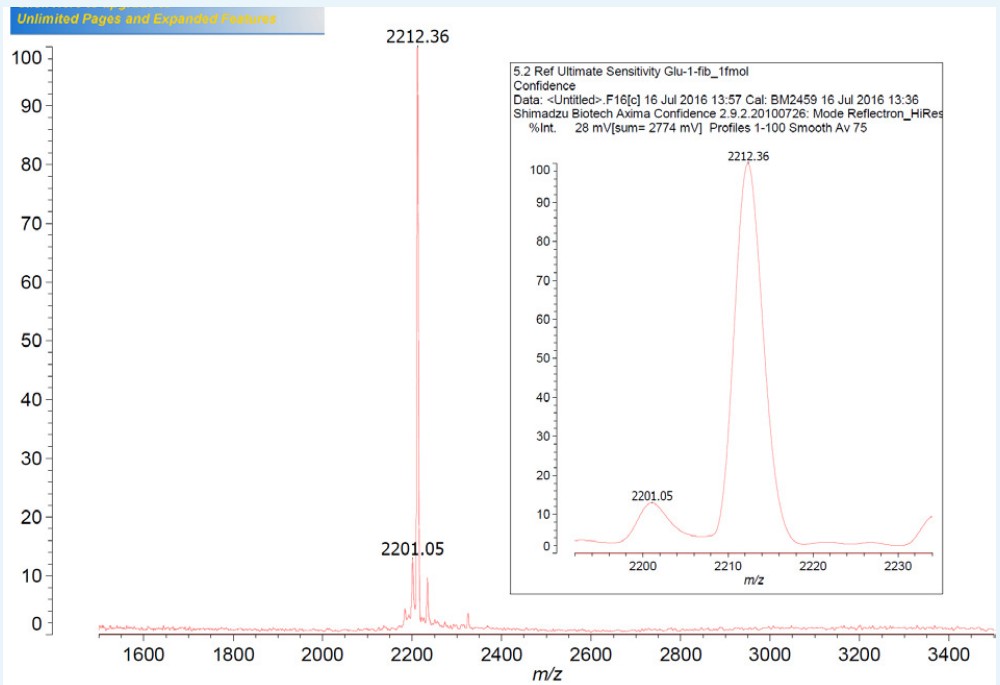

**Appendix 1—figure 20.** Characterization of SRC1-LXXLL peptide. (**a**) HPLC chromatogram of SRC1-LXXLL peptide. (**b**) MALDI-TOF of SRC1-LXXLL peptide.

(a) HPLC chromatogram of SRC2-LXXLL peptide

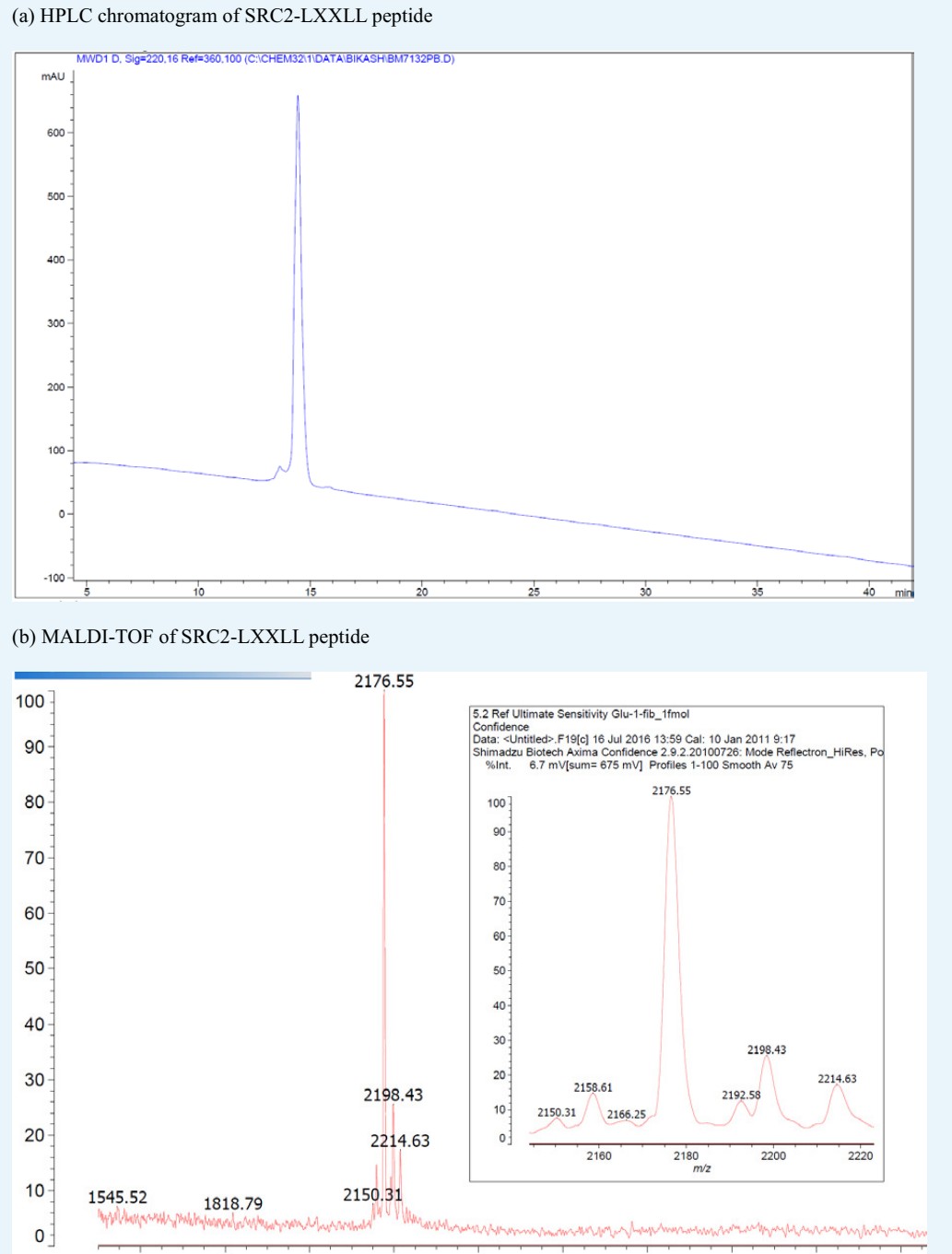

(b) MALDI-TOF of SRC2-LXXLL peptide

**Appendix 1—figure 21.** Characterization of SRC2-LXXLL peptide. (**a**) HPLC chromatogram of SRC2-LXXLL peptide. (**b**) MALDI-TOF of SRC2-LXXLL peptide.

(a) HPLC chromatogram of AIB1-LXXLL peptide

(a) MALDI-TOF of AIB1-LXXLL peptide

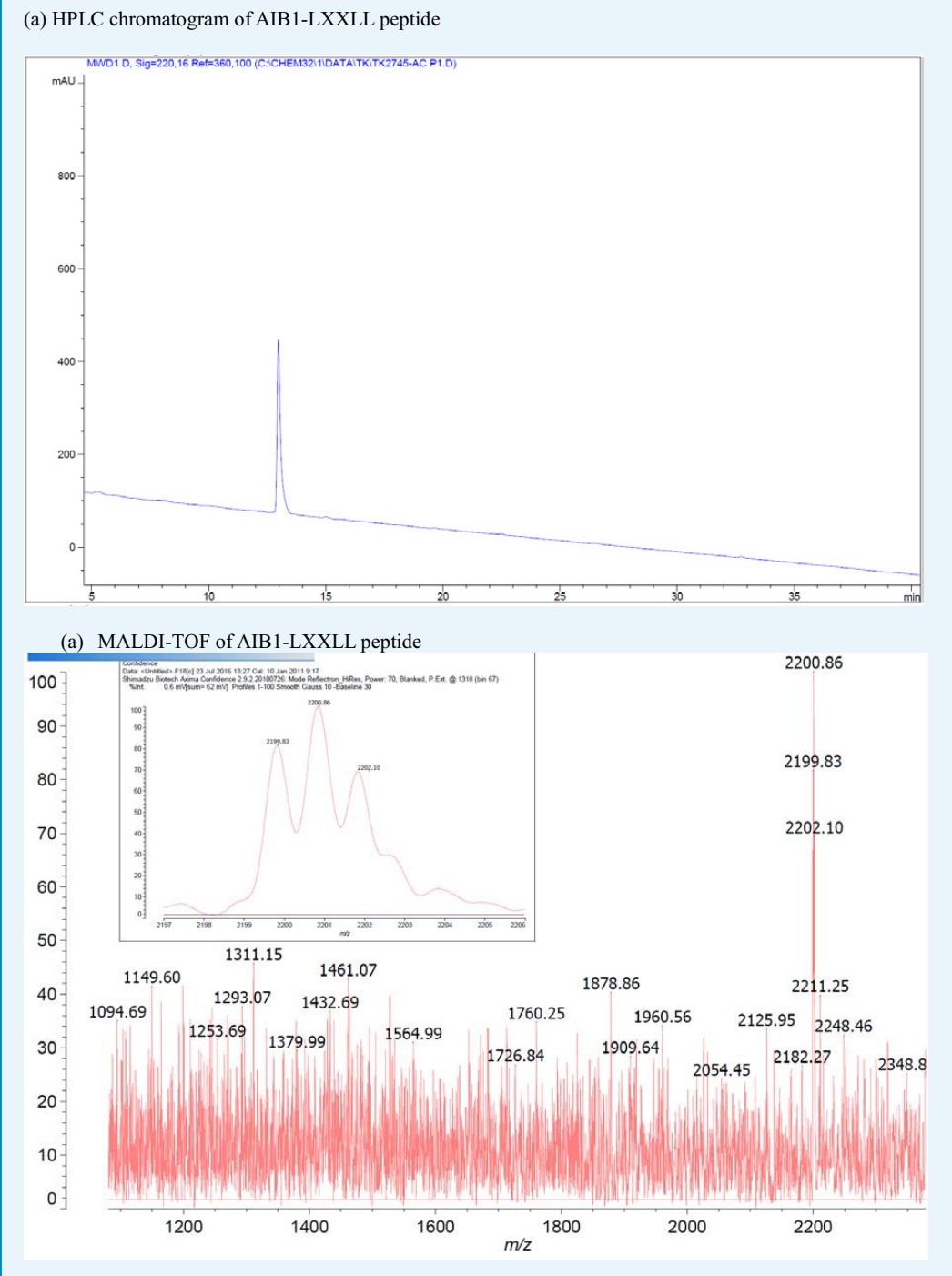

**Appendix 1—figure 22.** Characterization of AIB1-LXXLL peptide. (**a**) HPLC chromatogram of AIB1-LXXLL peptide. (**b**) MALDI-TOF of AIB1-LXXLL peptide.

(a) HPLC chromatogram of PELP1-LXXLL peptide

(b) MALDI-TOF of PELP1-LXXLL peptide

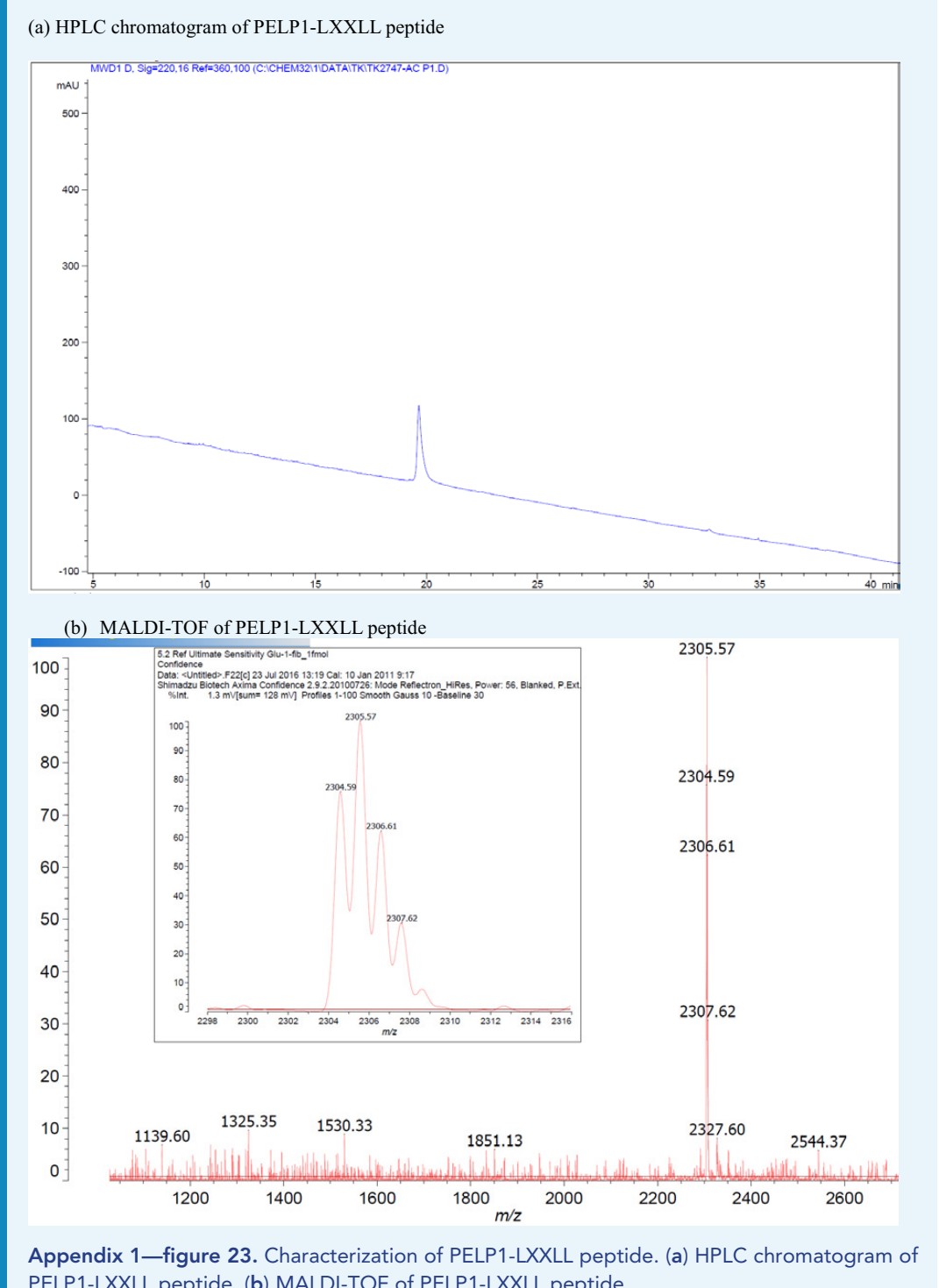

**Appendix 1—figure 23.** Characterization of PELP1-LXXLL peptide. (**a**) HPLC chromatogram of PELP1-LXXLL peptide. (**b**) MALDI-TOF of PELP1-LXXLL peptide.

