## [Decision Letter]

Thank you for submitting your article "Coregulator Binding Inhibitors effectively target Estrogen receptor positive Breast Cancers" for consideration by *eLife*. Your article has been reviewed by three peer reviewers, and the evaluation has been overseen by a Reviewing Editor and Kevin Struhl as the Senior Editor. The following individual involved in review of your submission has agreed to reveal his identity: Kendall W Nettles (Reviewer #1).

The reviewers have discussed the reviews with one another and the Reviewing Editor has drafted this decision to help you prepare a revised submission.

As you will see below, the reviewers have a number of concerns that require considerable work and explanation.

The estrogen receptor-alpha (ESR1) continues to be an important target in luminal breast cancer. Whereas the currently used endocrine therapies (e.g. aromatase inhibitors, tamoxifen and more recently fulvestrant) have had considerable success, the effectiveness of these therapies is limited by de novo and acquired resistance, and in the case of fulvestrant, suboptimal pharmaceutical properties. These newly described findings strongly support a novel first in class therapeutic for breast cancer, and a design strategy that can be readily applied to other nuclear receptors. However, the reviewers agree that some of these findings have been misinterpreted and additional information is required to support some of the key claims.

Specific comments:

1) The focus in the manuscript is on blocking the active conformation of the receptor, and especially the constitutively active mutants found in metastatic treatment resistant disease, but it Ignores the role of corepressors that are required for efficacy of tamoxifen (see PMID:12482846, 12145334, 17283072). It also ignores that their compound binds better to an antagonist conformation of the receptor, upon deletion of helix 12. There is a literature on CBI's as coactivator binding inhibitors. This group is now calling them coregulator binding inhibitors, which is confusing. The authors are encouraged to propose an alternative terminology. Perhaps coregulator binding modulator? Such a change would highlight that these compounds are distinct from previous CBIs in binding with higher affinity to an antagonist conformation surface in addition to the AF2 surface, which only exists with the agonist conformer, and that ECBI-11 uniquely activates apoptotic genes.

2) Crystal structures have shown that the LBD can adopt three major conformations, with helix 12 docked to form one side of the AF2 coactivator binding surface (diethylstilbesterol, 3ERD.pdb); 2) docked into the coregulator binding surface, blocking both coactivator and corepressor binding (tamoxifen, 3ERT.pdb); 3) or in solution (fulvestrant analog, 1HJ1.pdb), allowing binding of a corepressor peptide to a longer groove than seen in the AF2 surface (2JFA.pdb). Another important consideration is that the ligands can also directly contact corepressors (visualized in PMID 17283072, 2JFA.pdb). The identification of binding sites and discussion of the mechanism needs further clarification.

3) If ECBI-11 blocks corepressor binding one would expect AF1 activity. The coactivator binding data in Figure 2—figure supplement 2 suggests some residual binding with ECBI-11, similar to tamoxifen, so it will be important to determine the degree of AF1 activity of the compound, compared to tamoxifen. ECBI vs tamoxifen should be tested with an ERE-luc reporter assay in HepG2 cells, which show high tamoxifen agonist activity, or some other cell line with tamoxifen agonist activity.

4) In Figure 1, even though ER+ cell lines are more sensitive to ECBI-11 than ER- cell lines, the maximum reduction of viability varies dramatically. The authors should show a graph displaying the% maximum inhibition, and can they also include data comparing activity of ECBI-11 in these ER+ lines to fulvestrant or tamoxifen.

5) Figure 1: This model ignores the fact that the compound binds more strongly to an antagonist conformer, when helix 12 is deleted. Please model ECBI-11 to the LBD in the absence of helix 12, with tamoxifen versus SERD bound LBD. Use pdb files 3ERT and 5ACC and delete residues 531+. Please also describe whether the compound is interacting with helix 12 in the current model. Please make another figure panel showing side-chains and identify key molecular interactions, which could be supplemental.

6) The data in Figure 1 are confusing. Firstly, a control to evaluate the interaction of ECBI-11 with apo-ESR1 is missing. However, more puzzling is the fact that GDC-810 and AZD9494, both of which disrupt the AF-2 packet of ESR1, do not block the interaction of ECBI-11 with ESR-1. Tamoxifen does this, as expected. The authors note this discrepancy and suggest that maybe the activity of tamoxifen relates to a second tamoxifen binding site ESR1, as suggested several years ago by Elwood Jensen. This may be the case but if so it puts a significant hole in the mechanistic model that ECBI-11 is working through the canonical AF-2 pocket. Resolution of this issue is needed in light of the authors' proposed model.

7) Figure 2. The analysis is entirely focused on which coregulators are dismissed, and ignores the potential coregulators required for induction of apoptosis or transcriptional repression. Please provide additional analyses of the ECBI-recruited coregulators. This is one area where focusing on "binding inhibitor" is misrepresenting the results, and in doing so underselling the significance. By treating it as a binding inhibitor, the data reveals only a partial inhibition. By focusing on coregulator binding modulation, the authors can point out the novel ensemble of proteins that are driving new biology such as efficacy in treatment resistance models.

8) Figure 2. This is a key experiment that addresses whether tamoxifen competition derives from its positioning of helix 12, its side chain interaction with ECPI-11 or whether tamoxifen displaces ECBI-11 through a putative second binding site, overlapping with the corepressor binding site (PMID: 16782818). These data suggest that it is one of the latter two explanations. This needs to be further clarified using either transfected or recombinant protein. Tamoxifen binds in the pocket with single digit nanomolar affinity, and to the second site with "greatly reduced" affinity. The authors should perform a dose curve to determine which is more likely.

9) The Discussion needs revision to reflect the fact that the compound binds more tightly to an antagonist conformer, but also the agonist AF2 surface. This is a feature that is significantly different from previous CBIs, and likely contributes to the improved efficacy. The authors should include a more balanced Discussion that is less focused on inhibition of binding. There appear to be several contributions to the unique mechanism of action, including altered coregulator binding, inhibition of dimerization and DNA binding, induction of apoptotic genes, and a reduction in ESR1 expression.

10) It is not entirely convincing that the observed anti-tumor activities are on-target. The central premise of the work is that the ESR1 coregulator binding pocket that forms upon binding 17-β estradiol (E2) allows the interaction of specific LXXLL motifs within coregulators to engage the receptor and that this interaction can be targeted with peptidomimetics. In support of this mechanism the authors show that their lead compound ECBI-11 inhibits E2-dependent transcriptional activity, ESR1-coregulator binding, and the growth of ESR1 expressing cell lines. However, there are several pieces of inconsistent data that the authors themselves acknowledge (somewhat) that they cannot reconcile with this simple model. The authors should modify the messaging of the manuscript to present a new, potentially very important molecule (ECBI-11), whose activities are partially ESR1-dependent.

11) Does ECBI-11 inhibit the activity of an ERα-VP16 chimera? This receptor derivative does not require AF-2 as the VP16 activator overrides the activity of AF1 and AF-2. If the drug does indeed inhibit this chimera then the primary mechanistic hypothesis is unlikely to be correct. This, or a similar experiment that is specifically designed to "disprove" the authors mechanistic hypothesis, is needed.

12) Whereas it is interesting that the activity of ECBI-11 was restricted to ESR1 positive cells it cannot be concluded that the inhibition observed results from the inhibition of ESR1. Not all of the cells tested require E2/ESR1 for growth. Maybe it's just being a luminal breast cancer that defines responsively.

13) It is not clear why ZR75 cells were used for the in vivo studies when MCF-7 cell derived tumors are the gold standard. Did ECBI-11 not work in MCF-7 cells in vivo? There is data showing efficacy in the MCF-7 LTLT model although it's not clear how these cells were derived and a positive control (fulvestrant) was not included. Thus, although ECBI-11 works in this model, the role of ESR1 is not clear.

14) The data generated in the D2A1 model are problematic. The authors present this as a model of ESR-1 positive luminal cancer. However, a literature search revealed that it is used as a model of TNBC! Is the subline the authors using different from that used by others? Have they shown that ESR1 is expressed in these cells/tumors? This data is only of value (with respect to implicating ESR1) if they show that the tumors can be inhibited by more standard drugs (tamoxifen and/or fulvestrant).

15) The authors note that ECBI-11 treatment induces apoptosis in cells in which fulvestrant does not. Given that fulvestrant eliminates ESR1, expression it is unclear why ECBI-11 can induce this activity unless the apoptosis is due to an off-target activity.

16) The authors should include additional controls to demonstrate the specificity of ECBI-11 in cells.

a) In Figure 2 authors should include western analysis of more proteins that are immuno-precipitated by ER (e.g. SRC's, MED, TIF1, p300).

b) A negative control (a protein whose binding to ER is not reduced by ECBI-11 e.g. TIF1) should be included in their proximity ligation assays shown in Figure 2 and Figure 2—figure supplement 2. A similar negative control needs to be included in Figure 4.

17) If the authors want to examine effects of ECBI-11 on AR binding (Figure 4—figure supplement 1), why not include an AR agonist to stimulate AR activity? E2 is not an AR agonist and will not induce AR activity. In the absence of this control it is difficult to conclude that ECBI-11 does not modulate AR activity. This experiment should also include an AR antagonist to demonstrate that disruption of AR binding can be detected in their assay.

---

## [Author Response]

*Specific comments:*

*1) The focus in the manuscript is on blocking the active conformation of the receptor, and especially the constitutively active mutants found in metastatic treatment resistant disease, but it Ignores the role of corepressors that are required for efficacy of tamoxifen (see PMID:12482846, 12145334, 17283072). It also ignores that their compound binds better to an antagonist conformation of the receptor, upon deletion of helix 12. There is a literature on CBI's as coactivator binding inhibitors. This group is now calling them coregulator binding inhibitors, which is confusing. The authors are encouraged to propose an alternative terminology. Perhaps coregulator binding modulator? Such a change would highlight that these compounds are distinct from previous CBIs in binding with higher affinity to an antagonist conformation surface in addition to the AF2 surface, which only exists with the agonist conformer, and that ECBI-11 uniquely activates apoptotic genes.*

We agree with reviewer’s suggestions. Our mechanistic studies do indeed confirm that the activity of our compound is due to multiple mechanisms of action,including its abilityto bind to ER,modulate coregulator binding to ER, inhibit ER dimerization and induce apoptotic genes. To reflect its multiple mechanisms of action, we will use the nomenclature ERX, for Estrogen Receptor coregulator binding modulators where X refers to multiple mode of actions of ERX-11 on ER functions. We have modified the figures and the manuscript to reflect the new terminology.

*2) Crystal structures have shown that the LBD can adopt three major conformations, with helix 12 docked to form one side of the AF2 coactivator binding surface (diethylstilbesterol, 3ERD.pdb); 2) docked into the coregulator binding surface, blocking both coactivator and corepressor binding (tamoxifen, 3ERT.pdb); 3) or in solution (fulvestrant analog, 1HJ1.pdb), allowing binding of a corepressor peptide to a longer groove than seen in the AF2 surface (2JFA.pdb). Another important consideration is that the ligands can also directly contact corepressors (visualized in PMID 17283072, 2JFA.pdb). The identification of binding sites and discussion of the mechanism needs further clarification.*

As suggested by the reviewer, we have now performed docking simulations of ERX-11 on four crystal structures corresponding to 3ERD, 3ERT, 1HJ1, and 2JFA (Figure 2—figure supplement 6).

In 3ERD, binding of the agonist leads to a rearrangement of helix 12 and forms a hydrophobic cleft (i.e., AF2 binding site) that is surrounded by helix 3, 4, 5, and 12: we found that ERX-11 makes hydrophobic contact with the AF2 site with its two isobutyl side chain groups (Figure 2—figure supplement 6). In addition, the hydroxyl group of ERX-11 interacts with a residue near AF2 domain. These data indicate that ERX-11 interacts with ER LBD differently than the agonist.

The crystal structure of 3ERT shows that 4-hydroxytamoxifen (antagonist) induces conformational change and makes helix 12 occupy the AF2 binding site, blocking both coactivator and corepressor binding. This makes ERX-11 change its binding site to a nearby pocket formed by helix 5, 11, and 12, as shown in the Figure 2—figure supplement 6. These data show that ERX-11 could only interact with ER in the presence of tamoxifen, through an alternate binding site. These data could explain why the interaction between ERX-11 and purified ER could be blocked by tamoxifen. However, within the cell, tamoxifen cannot block ERX-11 binding to ER:, suggesting that the secondary binding site of ERX-11 on ER may be stabilized by coregulators.

Docking simulation of ERX-11 on human ERα with affinity tagged corepressor peptide (Figure 2—figure supplement 6) (2JFA.pdb) or rat ERβ crystal structure with ICI boundFigure 2—figure supplement 6 (1HJ1.pdb) showed that ECBI-11 can still bind to the AF2 domain, in a similar manner as it does when the ligand is bound. These data support our biochemical findings that ICI does not block ERX-11 interaction with ER.

*3) If ECBI-11 blocks corepressor binding one would expect AF1 activity. The coactivator binding data in Figure 2—figure supplement 2 suggests some residual binding with ECBI-11, similar to tamoxifen, so it will be important to determine the degree of AF1 activity of the compound, compared to tamoxifen. ECBI vs tamoxifen should be tested with an ERE-luc reporter assay in HepG2 cells, which show high tamoxifen agonist activity, or some other cell line with tamoxifen agonist activity.*

We have now tested whether ERX-11 has any residual activity via AF1 domain using endometrial Ishikawa cell line, which exhibit agonist activity via AF1. As expected, tamoxifen treatment promoted agonist activity in this model. However, in this assay we failed to detect any agonist activity of ERX-11. These results suggest that ERX-11 lacks AF1 activity. We have included this data as Figure 2—figure supplement 4.

*4) In Figure 1, even though ER+ cell lines are more sensitive to ECBI-11 than ER- cell lines, the maximum reduction of viability varies dramatically. The authors should show a graph displaying the% maximum inhibition, and can they also include data comparing activity of ECBI-11 in these ER+ lines to fulvestrant or tamoxifen.*

We have included a graph displaying the% maximum inhibition mediated by ERX-11 (1μM) in a number ER+ and ER- cell lines as a waterfall graph in Figure 1—figure supplement 1. We have also included data comparing the activity of ERX-11 in ER+ cell lines to fulvestrant or tamoxifen in Figure 1—figure supplement 1.

*5) Figure 1: This model ignores the fact that the compound binds more strongly to an antagonist conformer, when helix 12 is deleted. Please model ECBI-11 to the LBD in the absence of helix 12, with tamoxifen versus SERD bound LBD. Use pdb files 3ERT and 5ACC and delete residues 531+. Please also describe whether the compound is interacting with helix 12 in the current model. Please make another figure panel showing side-chains and identify key molecular interactions, which could be supplemental.*

We have created a model showing the side chains and interactions of ERX-11 with residues in the ER protein using 1L2I.pdb structure (Figure 2—figure supplement 6).

We also have carried out additional docking experiment without helix 12 with the crystal structures of 3ERT.pdb and 5ACC.pdb as suggested by the reviewer. The deletion of the helix 12 destroys the AF2 binding site and this makes ERX-11 look for another binding site. It is interesting to note that ERX-11 was found to bind to the tamoxifen binding site when the helix 12 was deleted in the crystal structure of 3ERT.pdb (Figure 2—figure supplement 6). The superimposition of tamoxifen (red) and ERX-11 (green) clearly shows the overlap of their binding sites (Figure 2—figure supplement 6). This may explain our experimental results showing the competition of tamoxifen with ERX-11 on ER-▲12. On the other hand, the deletion of the helix 12 of the crystal structure of 5ACC.pdb still allows ERX-11 to bind to the AF2 domain (Figure 2—figure supplement 6), which does not overlap with the binding site of SERD in the crystal structure (Figure 2—figure supplement 6 2—figure supplement C). This also confirms our experimental results of the inability of the SERDS to compete with ERX-11 binding to ER-▲12.

*6) The data in Figure 1 are confusing. Firstly, a control to evaluate the interaction of ECBI-11 with apo-ESR1 is missing. However, more puzzling is the fact that GDC-810 and AZD9494, both of which disrupt the AF-2 packet of ESR1, do not block the interaction of ECBI-11 with ESR-1. Tamoxifen does this, as expected. The authors note this discrepancy and suggest that maybe the activity of tamoxifen relates to a second tamoxifen binding site ESR1, as suggested several years ago by Elwood Jensen. This may be the case but if so it puts a significant hole in the mechanistic model that ECBI-11 is working through the canonical AF-2 pocket. Resolution of this issue is needed in light of the authors' proposed model.*

We have now repeated this experiment and included apo-ER control. The results showed that ERX-11 has weak interaction with apo-ER and addition of E2 significantly increased ERX-11 ability to interact with ER. Further, addition of tamoxifen interfered with ERX-11 interaction with ER, while SERDs only slightly reduced ERX-11 binding. New data replaced old Figure 1. Further, using crystal structures, we have refined our models, that explain why ERX-11 interaction with ER could be affected by tamoxifen but not by SERD. We thank the reviewers for their suggestions to model the interaction using existing structures, as it potentially explains our findings(Figure 2—figure supplement 6,Figure 2—figure supplement 7).

*7) Figure 2. The analysis is entirely focused on which coregulators are dismissed, and ignores the potential coregulators required for induction of apoptosis or transcriptional repression. Please provide additional analyses of the ECBI-recruited coregulators. This is one area where focusing on "binding inhibitor" is misrepresenting the results, and in doing so underselling the significance. By treating it as a binding inhibitor, the data reveals only a partial inhibition. By focusing on coregulator binding modulation, the authors can point out the novel ensemble of proteins that are driving new biology such as efficacy in treatment resistance models.*

Based on the reviewer’s suggestions, we have now reanalyzed the ERX-11-recruited binding proteins (Figure 2). Pathways analysis in terms of either biological processes or molecular functions revealed that ERX-11 binding proteins were involved in the activation of multiple pathways leading to transcriptional regulation (Figure 2—figure supplement 1). Based on these studies, and the finding that our compound mediates apoptosis, we have revised the terminology of our compounds to ERX to reflect its biological activity.

*8) Figure 2. This is a key experiment that addresses whether tamoxifen competition derives from its positioning of helix 12, its side chain interaction with ECPI-11 or whether tamoxifen displaces ECBI-11 through a putative second binding site, overlapping with the corepressor binding site (PMID: 16782818). These data suggest that it is one of the latter two explanations. This needs to be further clarified using either transfected or recombinant protein. Tamoxifen binds in the pocket with single digit nanomolar affinity, and to the second site with "greatly reduced" affinity. The authors should perform a dose curve to determine which is more likely.*

We have added a dose response experiment that suggests that ERX-11 binding to a putative seconding binding site on ER in the presence of tamoxifen. Increasing concentrations of tamoxifen fail to dislodge ERX-11 from ER (Figure 2—figure supplement 5). However, for the ER▲12 mutant, increasing concentrations of tamoxifen is only able to dislodge ERX-11 from ER at higher concentrations (Figure 2—figure supplement 5), suggesting that ERX-11 interaction with the primary binding site of ER is through the second binding site of tamoxifen with greatly reduced affinity. We have also added a model to explain the potential interaction between ER and ERX-11 or between ER▲12 mutant and ERX-11 in the absence or presence of agonist, SERDs or tamoxifen (Figure 2—figure supplement 7). Importantly, these data indicate that tamoxifen displaces ERX-11 from its binding site.

*9) The Discussion needs revision to reflect the fact that the compound binds more tightly to an antagonist conformer, but also the agonist AF2 surface. This is a feature that is significantly different from previous CBIs, and likely contributes to the improved efficacy. The authors should include a more balanced Discussion that is less focused on inhibition of binding. There appear to be several contributions to the unique mechanism of action, including altered coregulator binding, inhibition of dimerization and DNA binding, induction of apoptotic genes, and a reduction in ESR1 expression.*

We agree with reviewer’s suggestion. We have now modified the Discussion and renamed our compound as ERX-11 to reflect several contributions to its unique mode of action.

*10) It is not entirely convincing that the observed anti-tumor activities are on-target. The central premise of the work is that the ESR1 coregulator binding pocket that forms upon binding 17-β estradiol (E2) allows the interaction of specific LXXLL motifs within coregulators to engage the receptor and that this interaction can be targeted with peptidomimetics. In support of this mechanism the authors show that their lead compound ECBI-11 inhibits E2-dependent transcriptional activity, ESR1-coregulator binding, and the growth of ESR1 expressing cell lines. However, there are several pieces of inconsistent data that the authors themselves acknowledge (somewhat) that they cannot reconcile with this simple model. The authors should modify the messaging of the manuscript to present a new, potentially very important molecule (ECBI-11), whose activities are partially ESR1-dependent.*

We agree with reviewers’ suggestions. We have now modified our model and softened the discussion reflecting ERX-11 has complex mode of activity on ER and that its effects are only partially ER-dependent.

*11) Does ECBI-11 inhibit the activity of an ERα-VP16 chimera? This receptor derivative does not require AF-2 as the VP16 activator overrides the activity of AF1 and AF-2. If the drug does indeed inhibit this chimera then the primary mechanistic hypothesis is unlikely to be correct. This, or a similar experiment that is specifically designed to "disprove" the authors mechanistic hypothesis, is needed.*

We have now conducted suggested experiment usingERα-VP16 chimera receptor that does not require AF-2 as the VP16 activator overrides the activity of AF1 and AF-2. In these reporter-based assays, ERX-11 failed to reduce the ERE-luc reporter activity driven by ERα-VP16 chimera. In this assay, we have used tamoxifen and ICI as controls (Figure 2—figure supplement 4). As expected tamoxifen that signals via AF2, and AF1 also did not affected the ERα-VP16 chimera reporter activity, while ICI that degrades ER significantly reduced the reporter activity.

Collectively, these results confirm that the ERX-11 block signaling specifically driven by AF2 domain.

*12) Whereas it is interesting that the activity of ECBI-11 was restricted to ESR1 positive cells it cannot be concluded that the inhibition observed results from the inhibition of ESR1. Not all of the cells tested require E2/ESR1 for growth. Maybe it's just being a luminal breast cancer that defines responsively.*

As suggested by the reviewer, it is possible that ERX-11 target will be luminal breast cancer cells. However, our mechanistic studies support that ER as the primary target for this ERX-11 activity. To further support the importance of ER in ERX-11 activity, we have used restoration model cells where ER was introduced into ER-negative breast cancer model MDA-MB-231. MTT assays revealed that introduction of WT ER into MDA-MB-231 cells, restored ERX-11 growth inhibitory activity in non-responsive MDA-MB-231 cells. Similarly, introduction of the ER▲12 mutant into these cells restored responsiveness to ERX-11. These results further underscore the importance of ER in ERX-11 mode of action(Figure 2—figure supplement 5).

*13) It is not clear why ZR75 cells were used for the in vivo studies when MCF-7 cell derived tumors are the gold standard. Did ECBI-11 not work in MCF-7 cells in vivo? There is data showing efficacy in the MCF-7 LTLT model although it's not clear how these cells were derived and a positive control (fulvestrant) was not included. Thus, although ECBI-11 works in this model, the role of ESR1 is not clear.*

We have used two different models (MCF-7 and ZR-75) for in vivo studies to avoid any artifacts due to genetic background. Further, ERX-11 showed efficacy in both MCF-7 based xenografts tested in this study (Figure 5, MCF-7-PELP1, and 6E, MCF-7-LTLT). MCF-7-LTLT cells were widely used cells we received from Dr. Angela Brodie lab, exhibit-resistance to letrozole. MCF-7-LTLT cells were developed by continuous exposure of letrozole over a long period of time. We have now conduced an in vivo experiment showing the efficacy of fulvestrant on this xenograft. Further, we have also treated these tumors with ERX-11 as comparison with fulvestrant. Results showed that fulvestrant was able to reduce the growth of MCF-7-LTLT tumors. Further, ERX-11 exhibited similar potency in this model as fulvestrant. New data was included as Figure 6—figure supplement 1.

*14) The data generated in the D2A1 model are problematic. The authors present this as a model of ESR-1 positive luminal cancer. However, a literature search revealed that it is used as a model of TNBC! Is the subline the authors using different from that used by others? Have they shown that ESR1 is expressed in these cells/tumors? This data is only of value (with respect to implicating ESR1) if they show that the tumors can be inhibited by more standard drugs (tamoxifen and/or fulvestrant).*

D2A1 cells were initially characterized by Dr. Tekmal (coauthor of this manuscript) for hormonal therapy response (PMID 9310256). In D2A1 model cells, cellular gene int-5/aromatase in BALB/c mammary alveolar hyperplastic nodule (D2 HAN/D2 tumor cells) is activated as a result of mouse mammary tumor virus integration within the 3' untranslated region of the aromatase gene. Thus, these models also have ability to synthesize local estrogen via aromatase induction. Further, this model expresses estrogen receptor (ER) and represents a model of intra-tumoral estrogen driven mammary cancer. D2A1 cells are shown to be useful model for evaluating the effects of aromatase inhibitors and antiestrogens. To address the reviewers’ concerns, we have now conducted western blot analysis of D2A1 cells. Results showed that D2A1 cells express ER. Mammary gland lysates as well as murine ER positive cells E0771 were used as positive controls. Tamoxifen was able to reduce the growth of D2A1 cells. Further, IHC analysis confirmed that D2A1 tumors express ER and ERX-11 treatment substantially reduced ER expression (Figure 5—figure supplement 1).

*15) The authors note that ECBI-11 treatment induces apoptosis in cells in which fulvestrant does not. Given that fulvestrant eliminates ESR1, expression it is unclear why ECBI-11 can induce this activity unless the apoptosis is due to an off-target activity.*

We consistently observed activation of apoptosis by ERX-11 but failed to observe any apoptosis by tamoxifen or ICI in our assays. We believe that activation of apoptosis is not due to elimination of ER rather due to unique mechanism of action of ERX-11. Specifically, we predict that changes in the ER signaling due to alterations in coregulator binding to ER. Accordingly, our RNAseq data showed that alterations in genes that contribute activation of apoptosis. However, further studies are needed to clearly identify the mechanisms by which ERX-11 promotes apoptosis.

*16) The authors should include additional controls to demonstrate the specificity of ECBI-11 in cells.*

*a) In Figure 2 authors should include western analysis of more proteins that are immuno-precipitated by ER (e.g. SRC's, MED, TIF1, p300).*

*b) A negative control (a protein whose binding to ER is not reduced by ECBI-11 e.g. TIF1) should be included in their proximity ligation assays shown in Figure 2 and Figure 2—figure supplement 2. A similar negative control needs to be included in Figure 4.*

We have modified our figures to reflect the IP western blot analysis of additional proteins affected by ERX-11 as well as proteins that are not. We have also included a negative control for the PLA in Figure 2. We have also added negative control quantification in Figure 2—figure supplement 4.

*17) If the authors want to examine effects of ECBI-11 on AR binding (Figure 4—figure supplement 1), why not include an AR agonist to stimulate AR activity? E2 is not an AR agonist and will not induce AR activity. In the absence of this control it is difficult to conclude that ECBI-11 does not modulate AR activity. This experiment should also include an AR antagonist to demonstrate that disruption of AR binding can be detected in their assay.*

We have now conducted requested experiment with control AR agonist. Results showed ERX-11 has no activity on AR binding induced by DHT.